



# The consolidated European synthesis of CO₂ emissions and removals for EU27 and UK: 1990-2020

Matthew J. McGrath[1], Ana Maria Roxana Petrescu[2], Philippe Peylin[1], Robbie M. Andrew[3], Bradley Matthews[4], Frank Dentener[5], Juraj Balkovič[6], Vladislav Bastrikov[7], Meike Becker[8,9], Gregoire Broquet[1], Philippe Ciais[1], Audrey Fortems[1], Raphael Ganzenmüller[10], Giacomo Grassi[5], Ian Harris[11], Matthew Jones[12], Juergen Knauer[13], Matthias Kuhnert[14], Guillaume Monteil[15], Saqr Munassar[16], Paul I. Palmer[17], Glen P. Peters[3], Chunjing Qiu[1], Mart-Jan Schelhaas[18], Oksana Tarasova[19], Matteo Vizzarri[5], Karina Winkler[18,20], Gianpaolo Balsamo[21], Antoine Berchet[1], Peter Briggs[13], Patrick Brockmann[1], Frédéric Chevallier[1], Giulia Conchedda[22], Monica Crippa[5], Stijn Dellaert[23], Hugo A. C. Denier van der Gon[23], Sara Filipek[18], Pierre Friedlingstein[24], Richard Fuchs[20], Michael Gauss[25], Christoph Gerbig[16], Diego Guizzardi[5], Dirk Günther[26], Richard A. Houghton[27], Greet Janssens-Maenhout[5], Ronny Lauerwald[28], Bas Lerink[18], Ingrid T. Luijkx[18], Géraud Moulas[29], Marilena Muntean[5], Gert-Jan Nabuurs[18], Aurélie Paquirissamy[1], Lucia Perugini[30], Wouter Peters[18], Roberto Pilli[31], Julia Pongratz[10,32], Pierre Regnier[33], Marko Scholze[15], Yusuf Serengil[34], Pete Smith[14], Efisio Solazzo[5], Rona L. Thompson[35], Francesco N. Tubiello[22], Timo Vesala[36,37], Sophia Walther[16]

[1]Laboratoire des Sciences du Climat et de l'Environnement, CEA CNRS UVSQ UPSACLAY Orme des Merisiers, Gif-sur-Yvette, France
[2]Department of Earth Sciences, Vrije Universiteit Amsterdam, 1081HV, Amsterdam, the Netherlands
[3]CICERO Center for International Climate Research, Oslo, Norway
[4]Environment Agency Austria, Spittelauer Lände 5 1090, Vienna, Austria
[5]European Commission, Joint Research Centre, Via E. Fermi, 2749, TP 26/A, 21027, Ispra, Italy
[6]International Institute for Applied Systems Analysis (IIASA), 2361 Laxenburg, Austria
[7]Science Partners, 75010 Paris, France
[8]Geophysical Institute, University of Bergen, Bergen, Norway
[9]Bjerknes Centre for Climate Research, Bergen, Norway
[10]Department of Geography, Ludwig-Maximilians-Universität München, Luisenstraße 37, 80333 München, Germany
[11]National Centre for Atmospheric Science (NCAS), University of East Anglia, Norwich, United Kingdom; and Climatic Research Unit, School of Environmental Sciences, University of East Anglia, Norwich, United Kingdom
[12]Tyndall Centre for Climate Change Research, School of Environmental Sciences, University of East Anglia, Norwich Research Park,Norwich NR4 7TJ, United Kingdom
[13]Hawkesbury Institute for the Environment, Western Sydney University, Locked Bag 1797, Penrith, NSW 2751, Australia
[14]Institute of Biological and Environmental Sciences, University of Aberdeen, 23 St Machar Drive, Aberdeen, AB24 3UU, UK
[15]Dept. of Physical Geography and Ecosystem Science, Lund University
[16]Max Planck Institute for Biogeochemistry, Hans-Knöll-Strasse 10, 07745 Jena, Germany
[17]School of GeoSciences, University of Edinburgh, Edinburgh, UK
[18]Wageningen Environmental Research, Wageningen University and Research (WUR), Wageningen, 6708PB, the Netherlands
[19]Science and Innovation Department, World Meteorological Organization (WMO), Geneva, Switzerland
[20]Land Use Change & Climate Research Group, IMK-IFU, Karlsruhe Institute of Technology (KIT), Karlsruhe, Germany
[21]European Centre for Medium-Range Weather Forecasts (ECMWF), Reading, RG2 9AX, UK
[22]FAO, Statistics Division, Via Terme di Caracalla, Rome 00153, Italy
[23]Department of Climate, Air and Sustainability, TNO, Princetonlaan 6, 3584 CB Utrecht, the Netherlands
[24]College of Engineering, Mathematics and Physical Sciences, University of Exeter, Exeter EX4 4QF, UK
[25]Norwegian Meteorological Institute, Oslo, Norway
[26]Umweltbundesamt (UBA), 14193 Berlin, Germany
[27]Woodwell Climate Research Center, Falmouth, Massachusetts, U.S.A.
[28]Université Paris-Saclay, INRAE, AgroParisTech, UMR ECOSYS, Thiverval-Grignon, France
[29]ARTTIC, 39 rue des Mathurins, 75008 Paris, France
[30]Centro Euro-Mediterraneo sui Cambiamenti Climatici (CMCC), Viterbo, Italy
[31]Scientific consultant, Padua, Italy
[32]Max Planck Institute for Meteorology, Bundesstrasse 53, 20146 Hamburg, Germany
[33]Biogeochemistry and Modeling of the Earth System, Université Libre de Bruxelles, 1050 Bruxelles, Belgium
[34]Istanbul University, Faculty of Forestry, Department of Watershed Management, 34473 Sariyer, Istanbul, Turkey
[35]Norwegian Institute for Air Research (NILU), Kjeller, Norway
[36]University of Helsinki, Institute for Atmospheric and Earth System Research/Physics, Faculty of Science, 00560 Helsinki, Finland
[37]Institute for Atmospheric and Earth System Research,/Forest Sciences, Faculty of Agriculture and Forestry, University of Helsinki, Helsinki, Finland



*Correspondence to*: M.J. McGrath (matthew.mcgrath@lsce.ipsl.fr)
**Abstract**
Quantification of land surface-atmosphere fluxes of carbon dioxide ($CO_2$) fluxes and their trends and
uncertainties is essential for monitoring progress of the EU27+UK bloc as it strives to meet ambitious targets
determined by both international agreements and internal regulation.   This study provides a consolidated synthesis of
fossil sources ($CO_2$ fossil) and natural sources and sinks over land ($CO_2$ land) using bottom-up (BU) and top-down
(TD) approaches for the European Union and United Kingdom (EU27+UK), updating earlier syntheses (Petrescu et
al., 2020, 2021b). Given the wide scope of the work and the variety of approaches involved, this study aims to answer
essential questions identified in the previous syntheses and understand the differences between datasets, particularly
for poorly characterized fluxes from managed ecosystems. The work integrates updated emission inventory data,
process-based model results, data-driven sectoral model results, and inverse modeling estimates, extending the
previous period 1990-2018 to the year 2020 to the extent possible. BU and TD products are compared with European
National Greenhouse Gas Inventories (NGHGIs) reported by Parties including the year 2019 under the United Nations
Framework Convention on Climate Change (UNFCCC). The uncertainties of the EU27+UK NGHGI were evaluated
using the standard deviation reported by the EU Member States following the guidelines of the Intergovernmental
Panel on Climate Change (IPCC) and harmonized by gap-filling procedures. Variation in estimates produced with
other methods, such as atmospheric inversion models (TD) or spatially disaggregated inventory datasets (BU),
originate from within-model uncertainty related to parameterization as well as structural differences between models.
By comparing NGHGIs with other approaches, key sources of differences between estimates arise primarily in
activities.  System boundaries and emission categories create differences in $CO_2$ fossil datasets, while different land
use definitions for reporting emissions from Land Use, Land Use Change and Forestry (LULUCF) activities result in
differences for $CO_2$ land.  The latter has important consequences for atmospheric inversions, leading to inversions
reporting stronger sinks in vegetation and soils than are reported by the NGHGI.
**For $CO_2$ fossil emissions**, after harmonizing estimates based on common activities and selecting the most
recent year available for all datasets, the UNFCCC NGHGI for the EU27+UK accounts for $3392 \pm 49$ Tg $CO_2$ yr$^{-1}$
($926 \pm 13$ Tg C yr$^{-1}$), while eight other BU sources report a mean value of 3340 [3238,3401] [25th,75th percentile] Tg
$CO_2$ yr$^{-1}$ (948 [937,961] Tg C yr$^{-1}$). The sole top-down inversion of fossil emissions currently available accounts for
3800 Tg $CO_2$ yr$^{-1}$ (1038 Tg C yr$^{-1}$), a value close to that of the NGHGI, but for which uncertainty estimates are not
yet available. **For the net $CO_2$ land fluxes**, during the most recent five-year period including the NGHGI estimates,
the NGHGI accounted for $-91 \pm 32$ Tg C yr$^{-1}$ while six other BU approaches reported a mean sink of -62 [-117,-49]
Tg C yr$^{-1}$ and a 15-member ensemble of dynamic global vegetation models (DGVMs) reported -69 [-152,-5] Tg C yr$^{-}$
$^{1}$. The five-year mean of three TD regional ensembles combined with one non-ensemble inversion of -73 Tg C yr$^{-1}$
has a slightly smaller spread (0th-100th percentile of [-135,45] Tg C yr$^{-1}$), and was calculated after removing land-
atmosphere $CO_2$ fluxes caused by lateral transport of carbon (crops, wood trade and inland waters) resulting in
increased agreement with the the NGHGI and bottom-up approaches. Results at the sub-sector level (Forestland,
Cropland, Grassland) show generally good agreement between the NGHGI and sub-sector-specific models, but results
for a DGVM are mixed.  Overall, for both CO2 fossil and net CO2 land fluxes, we find current independent approaches





are consistent with the NGHGI at the scale of the EU27+UK. We conclude that $CO_2$ emissions from fossil sources
have decreased over the past 30 years in the EU27+UK, while large uncertainties on net uptake of $CO_2$ by the land
surface prevent trend identification. In addition, a gap on the order of 1000 Tg C $yr^{-1}$ between $CO_2$ fossil emissions
and net $CO_2$ uptake by the land exists regardless of the type of approach (NGHGI, TD, BU), falling well outside all
available estimates of uncertainties. However, uncertainties in top-down approaches to estimate $CO_2$ fossil emissions
remain uncharacterized and are likely substantial. The data used to plot the figures are available at
https://doi.org/10.5281/zenodo.7365863.
**1. Introduction**

Atmospheric concentrations of greenhouse gasses (GHGs) reflect a balance between emissions from both

human activities and natural sources, and removals by the terrestrial biosphere, oceans, and atmospheric oxidation.
Increasing levels of GHG in the atmosphere due to human activities have been the major driver of climate change
since the pre-industrial period (IPCC, 2021). In 2020, GHG mole fractions reached record highs, with globally
averaged mole fractions of 413.2 parts per million (ppm) for carbon dioxide ($CO_2$), representing 149% of the pre-
industrial level (WMO, 2021). The rise in $CO_2$ concentrations in recent decades is caused primarily by $CO_2$ emissions
from fossil sources. Globally, fossil emissions in 2020 (excluding the cement carbonation sink) totalled $9.5 \pm 0.5$ Gt
C $yr^{-1}$ ($34.8 \pm 1.8$ Gt $CO_2$ $yr^{-1}$), with expectations to rise in 2021 as the world recovered from the first year of the
Covid-19 pandemic (Friedlingstein et al., 2022). In contrast, global net $CO_2$ emissions from land use and land use
change (LULUC, primarily deforestation) estimated from bookkeeping models and dynamic global vegetation models
(DGVMs) were estimated to have a small decreasing trend over the past two decades, albeit with low confidence
(Friedlingstein et al., 2022). This decrease, however, is almost an order of magnitude less than the growth in fossil
emissions over the same period, and therefore the total fossil and net LULUC flux has still increased.

As all countries in the EU27+UK are Annex I Parties[1] to the United Nations Framework Convention on

Climate Change (UNFCCC), they prepare and report national GHG emission inventories (NGHGIs) on an annual
basis. These inventories contain annual timeseries of each country's GHG emissions from the 1990 base year[2] until
two years before the year of reporting and were originally set to track progress towards their reduction targets under
the Kyoto Protocol (UNFCCC, 1997). Annex I NGHGIs are reported according to the Decision 24/CP.19 of the
UNFCCC Conference of the Parties (COP) which states that the national inventories *shall* be compiled using the
methodologies provided in the *IPCC Guidelines for National Greenhouse Gas Inventories* (IPCC, 2006). The 2006
IPCC Guidelines provide methodological guidance for estimating emissions for well-defined sectors using national
activity and available emission factors. Decision trees indicate the appropriate level of methodological sophistication

---

[1] Annex I Parties include the industrialized countries that were members of the OECD (Organization for Economic Co-operation and Development) in 1992 plus countries with economies in transition (the EIT Parties), including the Russian Federation, the Baltic States, and several central and eastern European states (UNFCCC, https://unfccc.int/parties-observers, last access: February 2022).

[2] For most Annex I Parties, the historical base year is 1990. However, parties included in Annex I with an economy in transition during the early 1990s (EIT Parties) were allowed to choose one year up to a few years before 1990 as reference because of a non-representative collapse during the breakup of the Soviet Union. For the EU27+UK, this includes Bulgaria (1988), Hungary (1985–1987), Poland (1988), Romania (1989), and Slovenia (1986).





(*Tiered methods*) based on the absolute contribution of the sector to the national GHG balance and the country's
national circumstances (availability and resolution of national activity data and emission factors). Generally, Tier 1
methods are based on global or regional default emission factors that can be used with aggregated activity data, while
Tier 2 methods rely on country-specific factors and/or activity data at a higher category resolution. Tier 3 methods are
based on more detailed process-level modeling or in some cases facility-level emission observations. Annex I Parties
are furthermore required to estimate and report uncertainties in emissions (95 % confidence interval) following the
2006 IPCC guidelines using, as a minimum requirement, the Gaussian error propagation method (approach 1). Annex
I Parties are furthermore encouraged to use Monte-Carlo methods (approach 2) or a hybrid approach.  Additional
information on the NGHGIs can be found in Appendix A1.

In addition to the NGHGIs, other research groups and international institutions produce independent
estimates of national GHG emissions with two approaches: atmospheric inversions (top-down, TD) and GHG
inventories based on the same principle as NGHGIs but using slightly different methods (tiers), activity data, and/or
emissions factors (bottom-up, BU).  The current work has a strong focus on the EU27, and therefore sits within the
context of recent legislation passed by the European Parliament concerning commitments for the LULUCF sector to
achieve the objectives of the Paris Agreement and the reduction target for the Union (EU, 2018a and the proposed
amendments, EU, 2021a).  This legislation requires that, "Member States shall ensure that their accounts and other
data provided under this Regulation are accurate, complete, consistent, comparable and transparent".  The TD and BU
methods discussed below include the most up-to-date publicly available spatially explicit information, which can help
provide a quality check and increase public confidence in NGHGIs.

The work presented in this paper covers dozens of distinct datasets and models, in addition to the individual
country submissions to the UNFCCC of the EU Member States and the UK. As Annex I Parties, the NGHGIs of the
EU Member States and the UK are consistent with the general guidance laid out in IPCC (2006) yet still differ in
specific approaches, models, and parameters, in addition to definitional differences in the underlying system
boundaries and activity datasets.  A comprehensive investigation of detailed differences between all datasets is beyond
the scope of this paper, though systematic analyses have been previously made for specific sectors (e.g. AFOLU[3] -
Petrescu et al., 2020; previous synthesis to this work - Petrescu et al., 2021b; FAOSTAT versus UNFCCC NGHGIs -
Tubiello et al., 2021, Grassi et al., 2022a; UNFCCC versus bookkeeping models - Grassi et al, 2022b; and  UNFCCC
versus inversions - Deng et al., 2021) and by the Global Carbon Project $CO_2$ syntheses (e.g., Friedlingstein et al.,
2022).  Every year (time "$t$") the Global Carbon Project (GCP) in its Global Carbon Budget (GCB) quantifies large-
scale $CO_2$ budgets up to the previous year ("$t$-1"), bringing in information from global to large latitude bands, including
various observation-based flux estimates from BU and TD approaches (Friedlingstein et al., 2022).  The current
manuscript, given the focus on a single region ("Europe") with extensive data coverage, dives into more detail than
the GCB, including sector-specific models related to LULUCF (e.g., Forest land, Grassland, Cropland) and making
heavy use of the EU27+UK NGHGI in an effort to build mutual trust in the various approaches.  Compared to Petrescu
et al. (2021b), the current work updates datasets, methods, and uncertainties.

---

[3] We refer here to AFOLU as defined by the IPCC AR5: Agriculture, Forestry and Other Land Use.

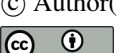



BU observation-based approaches used in the GCB rely heavily on statistical data combined with Tier 1 and
Tier 2 approaches. In the current work, focusing on a region that is well-covered with data and models (EU27+UK),
BU also refers to Tier 3 process-based models (see Sect. 2). At regional and country scales, systematic and regular
comparison of these observation-based $CO_2$ flux estimates with reported fluxes under the UNFCCC is more difficult.
Continuing our previous efforts within the European project VERIFY (VERIFY, 2022), the current study compares
observation-based flux estimates of BU versus TD approaches and compares them with NGHGI for the EU27-UK
bloc and five sub-regions. VERIFY also provides, as a first attempt, similar comparisons for all European countries
(VERIFY Synthesis Plots, 2022). The methodological and scientific challenges to compare these different estimates
have been partly investigated before (Pongratz et al., 2021, Grassi et al., 2018a, for LULUCF; Andrew, 2020, for
fossil sectors) but such comparisons were not done in a systematic and comprehensive way, including both fossil and
land-based $CO_2$ fluxes, before Petrescu et al. (2021b).
As Petrescu et al. (2021b) is the most comprehensive comparison of the NGHGI and research datasets
(including both TD and BU approaches) for the EU27+UK to date, the focus of the current paper is on improvement
of estimates in the most recent version in comparison with the previous one, including changes in the uncertainty
estimates and identification of the knowledge gaps and added value for policy making. Official NGHGI emissions are
compared with research datasets, including necessary harmonization of the latter on total emissions to ensure
consistency. Differences and inconsistencies between emission estimates were analyzed, and recommendations were
made towards future evaluation of NGHGI data. It is important to remember that, while NGHGIs include uncertainty
estimates, the "uncertainty analysis should be seen, first and foremost, as a means to help prioritize national efforts to
reduce the uncertainty of inventories in the future, and guide decisions on methodological choice" (Volume 1, Chapter
3, IPCC, 2006) and were therefore not developed to enable comparisons between countries or other datasets.  In
addition, individual spatially disaggregated research emission datasets often lack quantification of uncertainty. Here,
we focus on the mean value and various percentiles (0th, 25th, 75th, 100th) of different research products of the same
type to get a first estimate of uncertainty (see Sect. 2). Not all models/inventories provided an update for v2021, and,
therefore, for the non-updated datasets the previously published timeseries are shown.
**2.  $CO_2$ data sources and estimation approaches**
The $CO_2$ emissions and removals in the EU27+UK estimated by inversions and anthropogenic emission
inventories resolved at the source category level were analyzed. At the time of this work, data of $CO_2$ fossil emissions
and $CO_2$ land[4] emissions and removals (Tables 1 and 2) covered the period from 1990 to 2020, with some of the data
only available for shorter time periods. Since then, some datasets have been updated to include 2021, but not all, and

---

[4] The IPCC Good Practice Guidance (GPG) for Land Use, Land Use Change and Forestry (IPCC, 2003) describes a uniform structure for reporting emissions and removals of greenhouse gasses. This format for reporting can be seen as "land based": all land in the country must be identified as having remained in one of six classes since a previous survey, or as having changed to a different (identified) class in that period. According to IPCC SRCCL: Land covers the terrestrial portion of the biosphere that comprises the natural resources (soil, near surface air, vegetation and other biota, and water) the ecological processes, topography, and human settlements and infrastructure that operate within that system". Some communities prefer "biogenic" to describe these fluxes, while others find this confusing as fluxes from unmanaged forests, for example, are "biogenic" but not included in inventories reported to the UNFCCC.  As this comparison is central to our work, we decided that "land" as defined by the IPCC was a good compromise.



we made the decision to stay with the original time window for simplicity. The estimates are available both from
peer-reviewed literature and from new research results from the VERIFY project. BU results are compared to NGHGI
reported in 2021 (which contain the timeseries for 1990-2019). Data sources are summarized in Tables 1 and 2 with
the detailed description of all products provided in Appendices A1-A2. In Appendix A, the harmonized methodology
for calculation of uncertainties submitted by Member States to the UNFCCC in their National Inventory Reports
(NIRs) is explained. This includes the same 95 % confidence interval as is typically reported, but involved an
extensive gap-filling to cover more categories and more years than available in Petrescu et al. (2021b), which limited
uncertainty estimation to a single year.
BU anthropogenic $CO_2$ fossil estimates include global inventory datasets such as the Emissions Database for
Global Atmospheric Research (EDGAR v6.0.), Statistical Review of World Energy by BP, the Carbon Dioxide
Information Analysis Center (CDIAC), the Global Carbon Project (GCP), the Energy Information Administration's
(EIA) "International" dataset, and the International Energy Agency (IEA) (see Table 1). These datasets are all
described in detail by Andrew (2020). $CO_2$ land emission estimates are derived from BU biogeochemical models (e.g.
DGVMs, bookkeeping models, see Table 2). TD approaches include both high spatial resolution regional inversions
(CarboScopeReg, EUROCOM (Monteil et al., 2020), inversions based on the CIF-CHIMERE system (Berchet et al.,
2021) and LUMIA) and coarser spatial resolution global inversions (GCP 2021: Friedlingstein et al., 2022). Most of
the inversions were carried out for $CO_2$ land emissions, with only a single inversion for $CO_2$ fossil emissions (CIF-
CHIMERE). Note that CIF-CHIMERE provides estimates for both $CO_2$ land and $CO_2$ fossil from separate simulations.
These estimates are described in Sect. 2.3.
The sign of the fluxes is defined from an atmospheric perspective: positive values represent a net source to
the atmosphere and negative values a net removal from the atmosphere. As an overview of potential uncertainty
sources, Table B1 presents the use of emission factor data (EF), activity data (AD), and, whenever available,
uncertainty methods used for all $CO_2$ land data sources in this study, in addition to more details on each model in
Appendices A. The referenced data used for the figures' replicability purposes are available for download (McGrath
et. al, 2022). Upon request, the codes necessary to plot the figures in the same style and layout can be provided. The
focus is on EU27+UK emissions. In the VERIFY project, an additional web tool was developed which allows for the
selection and display of all plots shown in this paper, not only for the EU Member States and UK but for a total of 79
countries and groups of countries in Europe (Table A1, Appendix A). The data is free and can be accessed upon
registration (VERIFY Synthesis Plots, 2022).
For the sake of harmonization, we report the mean values of all ensembles. For small sample sizes (e.g., the
regional inversions of CSR with four members), the literature does not give a clear indication on whether the mean or
the median is preferred; a preference for one or the other depends on what one wishes to demonstrate. In particular,
the median downplays the skewness of the data (outliers). We have taken efforts to exclude outliers from the datasets
used to construct ensembles, and consequently the datasets which remain should be randomly distributed. For this
reason, we display the mean for all ensembles. As the number of datasets in some ensembles is small (less than five),
we display the minimum and maximum annual values for every year (i.e., the 0th/100th percentiles) to give an idea
of the spread. For ensembles with more than ten members (i.e., TRENDY), we show the mean and the 0th/100th



percentiles along with the 25th/75th percentiles in the figures. This combination demonstrates "more likely" and
"possible" behavior; as only one ensemble has both bars, displaying them does not overwhelm the reader much more
than the standard graphs, and we find the added information to be worth the trade-off. In the text, we report the mean
and 0/100th percentiles for small ensembles and mean along with the 25th/75th for larger ensembles.
The current work extends Petrescu et al. (2021b) by updating the included datasets (both increasing the
number of years covered and in some cases updating the model versions), adding datasets, and highlighting changes
in terms of mean annual emissions and trends. For clarity, the data from Petrescu et al. (2021b) is labeled as v2019,
while the latest results are labeled v2021.

**2.1. CO$_2$ anthropogenic emissions from NGHGI**

The UNFCCC NGHGI (2021) estimates for the period 1990 to year $t$-2 (2019), collected for the EU27 and
UK, are the basis for this dataset. For historical reasons, a few EU countries provide data for a different base year than
1990[5], yet it should be noted that regardless of the base year all countries of the EU27+UK bloc are obliged to report
estimates for the period 1990 to year $t$-2. The Annex I Parties to the UNFCCC are required to report annual GHG
inventories that include a NIR, with qualitative information on data and methods and a Common Reporting Format
(CRF) set of tables that provide quantitative information on GHG emission by category. This annually updated dataset
includes anthropogenic emissions and removals. For the land-based sector, the land management proxy is used as a
way to report only anthropogenic fluxes (Grassi et al., 2018a, 2021). This proxy allows Member States to report all
fluxes coming from land designed as "managed" without trying to disentangle their natural and anthropogenic origins.
Figure B1 shows the annual NGHGI (2021) anthropogenic CO$_2$ timeseries disaggregated by sector in order to provide
context.

*2.2. CO$_2$ fossil emissions*

CO$_2$ fossil emissions occur when fossil carbon compounds are broken down via combustion or other non-
combustive industrial processes. Most of these fossil compounds are in the form of fossil fuels, such as coal, oil, and
natural gas. Another source category of fossil CO$_2$ emissions is fossil carbonates, such as calcium carbonate and
magnesium carbonate, which are used in industrial processes. Because CO$_2$ fossil emissions are largely connected
with energy, which is a closely tracked commodity group of high economic importance, there is a wealth of underlying
data that can be used for estimating emissions. However, differences in collection, treatment, interpretation and
inclusion of various factors – such as carbon contents and fractions of the fuel's carbon that is oxidized – lead to
methodological differences (Appendix A) resulting in differences of emissions between datasets (Andrew, 2020).
Atmospheric inversions for emissions of fossil CO$_2$ are not as established as their bottom-up counterparts (Brophy et

---

[5] For most Annex I Parties, the historical base year is 1990. However, parties included in Annex I with an economy in transition during the early
1990s (EIT Parties) were allowed to choose one year up to a few years before 1990 as reference because of a non-representative collapse during the
breakup of the Soviet Union (e.g., Bulgaria, 1988, Hungary, 1985–1987, Poland, 1988, Romania, 1989, and Slovenia, 1986).



al., 2019). The main reason is that the types of atmospheric monitoring instruments suitable for fossil $CO_2$ atmospheric inversions have not yet been widely deployed (Ciais et al., 2015). One of the rare inversions is presented below.

In this analysis, the inventory-based bottom-up $CO_2$ fossil emissions estimates are separated and presented per fuel type and reported for the last year when all data products are available (2017). This updates Andrew (2020) and Petrescu et al. (2021b) which both report the year 2014. In order to provide a quasi-independent estimate of fossil emissions assimilating satellite observations of the atmosphere, the CIF-CHIMERE model was used to produce a fossil fuel $CO_2$ emission estimate for the year 2017. CIF-CHIMERE is a coupling between the variational mode of the Community Inversion Framework (CIF) platform developed in the VERIFY project (Berchet et al., 2021), the CHIMERE chemistry transport model (Menut et al., 2013) and the adjoint of this model (Fortems-Cheiney et al., 2021a). To overcome the lack of $CO_2$ observation networks suitable for the monitoring of fossil fuel $CO_2$ emissions at national scale, this inversion is based on the assimilation of satellite $NO_2$ data, as $NO_2$ is co-emitted with $CO_2$ during fossil fuel combustion. Recent top-down inversions of anthropogenic $CO_2$ emissions from Europe indicate that uncertainties using satellite measurements of co-emitted $NO_2$ are much lower than for co-emitted CO when deriving fossil $CO_2$ emissions (Konovalov et al., 2016). Therefore, results shown below only incorporate $NO_2$ and not CO observations. While the spatial and temporal coverage of the $NO_2$ observations is large, there are many factors that determine the ratio of $NO_2$ to $CO_2$ emissions. Therefore, the influence of using $NO_2$ observations in determining fossil $CO_2$ emissions is subject to uncertainties which have not been characterized appropriately yet in the framework of VERIFY. Here, this conversion relies heavily on the emission ratios per country, month and large sector of activity from the TNO-GHGco-v3 inventory (Dellaert et al., 2021), which has been partly developed in VERIFY, and which is based on the most recent UNECE-CLRTAP[6] and UNFCCC official country reporting respectively for air pollutants and greenhouse gasses. The detailed descriptions of each of the data products are found in Appendix A1.

*Table 1: Data sources for the anthropogenic $CO_2$ fossil emissions included in this study, all updated from Petrescu et al. (2021b):*

---

[6] UNECE Convention on Long-Range Transboundary Air Pollution. https://unece.org/environment-policy/air



| Anthropogenic CO$_2$ fossil | | | |
|---|---|---|---|
| **Data/model name** | **Contact / lab** | **Species / Period** | **Reference/Metadata** |
| **UNFCCC NGHGI (2021)** | UNFCCC | Anthropogenic fossil CO$_2$ 1990-2019 | IPCC (2006) <br> UNFCCC NIRs/CRFs <br> https://unfccc.int/ghg-inventories-annex-i-parties/2021 <br> (UNFCCC, 2021a, 2021b) |
| **Compilation of multiple CO$_2$ fossil emission data sources (Andrew 2020) EDGAR v6.0, BP, EIA, CDIAC, IEA, GCP, CEDS, PRIMAP** | CICERO | CO$_2$ fossil country totals and split by fuel type 1990-2018 (or last available year) | EDGAR v6.0 <br> https://edgar.jrc.ec.europa.eu/ <br> BP 2021 report (BP, 2021) <br> EIA <br> https://www.eia.gov/beta/international/data/browser/views/partials/sources.html <br> CDIAC <br> https://energy.appstate.edu/CDIAC (Gilfillan and Marland, 2021) <br> IEA : www.iea.org <br> CEDS <br> https://github.com/JGCRI/CEDS (O'Rourke et al., 2021) <br> GCP <br> (Friedlingstein et al., 2022) <br> PRIMAP-hist (Gütschow et al., 2021) <br> https://doi.org/10.5281/zenodo.4479171 |
| **Fossil fuel CO$_2$ inversions** | LSCE | Inverse fossil fuel CO$_2$ emissions 2005-2020 | Fortems-Cheiney et al. (2021) <br> Fortems-Cheiney and Broquet (2021) |

## 2.3. CO$_2$ land fluxes

Data products from BU and TD CO$_2$ land fluxes including CO$_2$ emissions and removals from land use, land use change, and forestry (LULUCF) activities are summarized in Table 2. All models and approaches produce an estimate of the net carbon flux from the land surface including uptake through photosynthesis and emission through respiration and/or disturbances. The details may vary significantly between approaches, however. Attempts are made where possible to harmonize input data and compare results which roughly correspond to similar categories included in the NGHGI. Further details are described throughout the rest of this article. As with CO$_2$ fossil fluxes, the primary distinctions are between the NGHGI, other bottom-up approaches, and top-down approaches. The situation becomes





more complicated for CO₂ land fluxes due to the inclusion of approaches which only address a single land use class
(e.g., Forest land).
For the analysis at category level, the CO₂ net emissions from the LULUCF sector that are primarily
considered in this synthesis are from three land use classes[7] (Forest land, Cropland, and Grassland), each split into a
land class remaining in the same land class[8] or a land class converted to another class. The NGHGIs are the only
results discussed here which make use of this transition period, but the distinction is important so as to inform which
NGHGI categories to use in the comparison. Wetlands, Settlements, Other land, and Harvested wood products (HWP)
categories are included in the discussion on total LULUCF activities in Sect. 3.3.1, 3.3.3 and 3.3.4. Not all the classes
reported to the UNFCCC are present in FAOSTAT or other models. Some models are sector-specific (e.g., Forest
land) while other models include a larger subset of the six UNFCCC classes (e.g., DGVMs which simulate Forest
land, Grassland, and Cropland). The notations FL, CL and GL are used to indicate total emissions and removals from
the respective Forest land, Cropland and Grassland land use categories (i.e. the remaining + conversions to these
classes). The notations "FL-FL", "CL-CL" and "GL-GL" are used to indicate emissions and removals from respective
forest, cropland and grassland areas which have remained in the same class from year to year, or in the case of NGHGI
lands that have not undergone conversion within the aforementioned transition period (e.g. *t*-20) .
The results from sector-specific models reporting carbon fluxes for FL-FL (EFISCEN-Space and CBM), CL
and GL (EPIC-IIASA and ECOSSE) are presented separately from the models and datasets including multiple land
use categories and simulating land use changes: FAOSTAT (version 2021), the DGVM ensemble TRENDY v10
(Friedlingstein et al., 2022; Le Quéré et al., 2009), the ORCHIDEE and CABLE-POP DGVMs forced by high
resolution meteorological data as part of the VERIFY project, and the two bookkeeping approaches of H&N
(Houghton & Nassikas, 2017) and BLUE (Hansis et al., 2015). BLUE includes two simulations with different land-
use forcing, one made for the VERIFY H2020 project and one for the GCP 2021 (Friedlingstein et al., 2022). For CL
and GL both the EPIC-IIASA and ECOSSE sector-specific models reported updates, although ECOSSE only updated
results for GL. Process included in all the products are summarized in Appendix A2 and Table B2.
The two updated inverse model ensembles presented are the GCP2021 for the period 2010-2020
(Friedlingstein et al., 2022) and EUROCOM for the period 2009-2018 (Monteil et al., 2020; Thompson et al., 2020).
The GCP inversions are global and include CarbonTracker Europe (CTE: van der Laan-Luijkx et al., 2017), CAMS
(Chevallier et al., 2005), the Jena CarboScope (Rödenbeck, 2005), NIESMON-CO₂ (Niwa et al., 2017), CMS-Flux
(Liu et al., 2021) and UoE (Feng et al., 2016). The EUROCOM inversions are regional, with a domain limited to
Europe and higher spatial resolution atmospheric transport models, with four inversions covering the entire period
2009-2018 as analyzed in Thompson et al. (2020). All inversions provide Net Ecosystem Exchange (NEE) fluxes.
These inversions make use of more than 30 atmospheric observing stations within Europe, including flask data and

---

[7] According to 2006 IPCC guidelines the LULUCF sector includes six management classes (Forest land, Cropland, Grassland, Wetlands, Settlements and Other land)

[8] According to 2006 IPCC guidelines, land converted to a new category should be reported in a "conversion" category for *N* years and then moved to a "remaining" category, unless a further change occurs. Converted land refers to CO₂ emissions from conversions to and from all six classes that occurred in the previous *N* years. By default, *N* is equal to 20, although the guidelines recognize that longer times may be necessary in temperate and boreal environments for the dead biomass and soil carbon pools to reach the new equilibrium. Member States have the freedom to select a length of time appropriate to their own circumstances.



continuous observations and work at typically higher spatial resolution than the global inversion models (Table 2).
The prior anthropogenic emissions provided for all regional inversions reported here (i.e., EUROCOM, EUROCOM
drought 2018, VERIFY CSR, VERIFY CIF-CHIMERE, and VERIFY LUMIA) are all based on EDGAR v4.3, BP
statistics, and TNO datasets by generating spatial and temporal distributions through the COFFEE approach
(Steinbach et al., 2011).  Small differences exist between exact versions used by the different groups.  The prior
anthropogenic emissions for the GCP global inversions, GridFEDv2021 and v2022, are also based on EDGARv4.3.2
(Janssens-Maenhout et al., 2019).  Overall, differences in the prior anthropogenic emissions are not expected to explain
the large differences seen between the different regional biogenic inversions nor between the regional and global
biogenic inversions, but efforts should be continued to harmonize them to the greatest extent possible in future
intercomparisons.

Additional inversions for Europe from three regional scale inversion systems are analyzed. Two of these
systems are part of the EUROCOM ensemble, but new runs were carried out for the VERIFY project. The
CarboScopeRegional (CSR) inversion system has performed additional runs for VERIFY for the years 2006-2020
with multiple ensemble members differing by biogenic prior fluxes and assimilated observations. The results are
plotted separately to illustrate two points: 1) that the CSR simulations for VERIFY are not identical to those submitted
to EUROCOM (VERIFY runs from CSR included several sites that started shortly before the end of the EURCOM
inversion period), and 2) the CSR model was used in four distinct runs in VERIFY.  Note that the ensemble members
differ from previous years (the spatial correlation length is kept constant this year, while more prior fluxes are used).
By presenting CSR separate from the EUROCOM results, one can get an idea of the uncertainty due to various model
parameters in one inversion system with one single transport model. The LUMIA inversion system submitted four
simulation results to the VERIFY project, based on the 2018 Drought Task Force project (labeled here as EUROCOM,
Thompson et al., 2020).  The primary difference is that the years 2019-2020 were added based on boundary conditions
using TM5 and ERA5 meteorological data. The four different variants include one reference simulation and three
simulations which change spatial correlation lengths, the number of observation sites, and the magnitude of
uncertainties in the boundary conditions. As one of the variants is only available for 2019-2020 (changing the
uncertainties in the boundary conditions), this variant was dropped from the results and only the remaining three
simulations are presented, covering the period 2006-2020.

An inversion of the NEE over 2005-2020 from the CIF-CHIMERE variational inversion system is also
analyzed. The configuration of this inversion is close to that of the PYVAR-CHIMERE NEE inversions in the
EUROCOM ensembles and follows the general principles of Broquet et al. (2013). However, it uses distinct inputs,
which play a critical role in the inversion, such as a more recent ORCHIDEE simulation as prior estimate of the NEE
and a more recent CAMS global inversion to impose the regional $CO_2$ boundary conditions.

*Table 2: Data sources for the land $CO_2$ emissions included in this study.  Details are found in Appendix A2.*



| NGHGI net $CO_2$ land flux | | | | |
|---|---|---|---|---|
| **Data source** | **Contact / lab** | **Variables** <br> **Period (timestep)** <br> **Resolution** | **References** | **Status compared to Petrescu et al. (2021b)** |
| **UNFCCC NGHGI (2021)** | Member State inventory agencies <br><br> Annual uncertainty gap-filling for total LULUCF by Environment Agency Austria (EAA). | LULUCF Net $CO_2$ emissions/removals. 1990-2019 (1Y) Country-level | IPCC (2006) <br><br> UNFCCC CRFs <br><br> https://unfccc.int/process-and-meetings/transparency-and-reporting/reporting-and-review-under-the-convention/greenhouse-gas-inventories-annex-i-parties/national-inventory-submissions-2019 | Updated |
| **Inventory and model estimates of net $CO_2$ land flux** | | | | |
| **ORCHIDEE** | LSCE | $CO_2$ fluxes from all ecosystems reported as Net Biome Productivity (NBP). 1990-2020 (3H) 0.125º x 0.125º | Ducoudré et al. (1993) <br> Viovy et al. (1996) <br> Polcher et al. (1998) <br> Krinner et al. (2005) | Updated |
| **CABLE-POP** | Western Sydney University | $CO_2$ fluxes (NBP). Model includes N cycling. 1990-2020 (1M) 0.125º x 0.125º | Haverd et al. (2018) | New |
| **TRENDY v10** | MetOffice UK | $CO_2$ fluxes (NBP) 15 models (all except ISAM) 1990-2020 (3H-1M) 0.125º x 0.125º | Friedlingstein et al. (2022; Table 4) | Updated |





| | | | | |
|---|---|---|---|---|
| **CO₂ emissions from inland waters** | ULB | Average C fluxes from rivers, lakes and reservoirs, with lateral C transfer from soils. 1990-2018 (-) 0.1º x 0.1º | Lauerwald et al. (2015) Hastie et al. (2019) Raymond et al. (2013) | Not updated |
| **CBM** | EC-JRC | CO₂ fluxes (NBP) as historical 2000-2015 and extrapolation for 2017-2020 (1Y) Country-level | Kurz et al. (2009) Pilli et al. (2022) | Updated |
| **ECOSSE** | UNIABDN | CO₂ fluxes (NBP) from croplands and grassland ecosystems. Crops: 1990-2020 (1Y) Grass: 1990-2018 (1Y) 0.125º x 0.125º | Bradbury et al. (1993) Coleman (1996) Jenkinson (1977, 1987) Smith et al. (1996, 2010a,b) | Updates only for croplands |
| **EFISCEN-Space** | WUR | CO₂ fluxes (NBP): single average value for 5 year periods, replicated on a yearly time axis. 0.125º x 0.125º | Verkerk et al. (2016) Schelhaas et al. (2017, 2020) Nabuurs et al. (2018) | Updates for 15 countries |
| **EPIC-IIASA** | IIASA | CO₂ fluxes (NBP) from cropland 1991-2020 (1M) 0.125º x 0.125º | Balkovič et al. (2013, 2018, 2020) Izaurralde et al. (2006) Williams et al. (1990) | Updated for croplands, new estimates for grasslands |
| **BLUE (VERIFY) and BLUE (GCP)** | LMU Munich | CO₂ fluxes from land use change. VERIFY: 1990-2019 (1Y) GCP: 1990-2020 (1Y) 0.25º x 0.25º | Hansis et al. (2015) Ganzenmüller et al. (2022) - VERIFY Friedlingstein et al. (2022) - GCP2021 | Updated |
| **H&N** | Woodwell Climate Research Center | CO₂ fluxes from land use change. 1990-2020 (1Y) Country-level | Houghton and Nassikas (2017) | Updated |
| **FAO** | FAOSTAT | CO₂ emissions / removal from LULUCF processes. 1990-2020 (1Y) Country-level | FAO (2021) Federici et al. (2015) Tubiello et al. (2021) | Updated |
| **CO₂ atmospheric inversion estimates** | | | | |



| CSR inversions for VERIFY | MPI -Jena | Total $CO_2$ inverse flux (NBP) 2006-2020 (3H) 0.5° x 0.5° | Kountouris et al. (2018 a,b) | Updated |
|---|---|---|---|---|
| LUMIA | Lund University (INES) | Total $CO_2$ inverse flux (NBP) 2006-2020 (1W) 0.25° x 0.25° | Monteil and Scholze (2021) | New |
| CIF-CHIMERE | LSCE | Total $CO_2$ inverse flux (NBP) 2005-2020 (3H) 0.5° x 0.5° | Berchet et al. (2021) Broquet et al. (2013) | New |
| GCP 2021 global inversions (CTE, CAMS, CarboScope, NISMON-CO₂, UoE, CMS-Flux) | GCP | Total $CO_2$ inverse flux (NBP) Six inversions 2010-2020 (various) | Friedlingstein et al. (2022) Van der Laan-Luijk et al. (2017) Chevallier et al. (2005) Rödenbeck et al. (2005) Niwa et al. (2017) Feng et al. (2016) Liu et al. (2021) | Updated |
| EUROCOM regional inversions (CSR, LUMIA, PYVAR) | LSCE, ULUND, MPI-Jena, NILU | Total $CO_2$ inverse flux (NBP) Three inversions 2009-2018 (3H-1M) | Monteil et al. (2020) Thompson et al. (2020) | Updated (also replaced CSR with the mean of the four runs submitted to VERIFY). FLEXINVERT and NAME are not included (Fig. A5) |


364  All of the bottom-up models in this work require external forcing datasets. In the context of the VERIFY
project (VERIFY, 2022), an effort was made to provide a single, harmonized version of several kinds of data
(meteorological, land use/land cover, and nitrogen deposition) on a high-resolution grid over Europe. These datasets
were then made available to all of the modeling groups to use in their simulations. Such a practice is common in
model intercomparison projects. However, as the models in Table 2 are not all the same type, data harmonization
presented more of a challenge in this work as not all models use the same inputs. All of the datasets described in
Appendix A2 were used by at least one modeling group in this work.
**3. Results and discussion**
**3.1. Overall NGHGI reported anthropogenic $CO_2$ fluxes**

373  In 2019, the UNFCCC NGHGI (2021) net $CO_2$ flux estimates for EU27+UK, accounted for 3.01 Gt $CO_2$
from all sectors (including LULUCF) and 3.28 Gt $CO_2$ excluding LULUCF (Fig. B1), corresponding to a net sink of



LULUCF of -0.27 ± 0.11 Gt $CO_2$. In 2019, few large economies accounted for the majority of EU27+UK emissions,
with Germany, UK, Italy and France representing 53 % of the total $CO_2$ emissions (excluding LULUCF). For the
LULUCF sector, the countries reporting the largest $CO_2$ sinks in 2019 were Italy, Spain, Sweden, and France
accounting for 56 % of the overall EU27+UK sink. Only a few countries (Czech Republic, The Netherlands, Ireland
and Denmark) reported a net LULUCF source in 2019. Some countries, like Portugal, report sources in some years
due to wildfires, with sinks in other years. The NGHGI shows minimal inter-annual variability (largely due to
methodology), and consequently the 2019 values are indicative of longer-term averages, showing a constant trend
between 2017-2019.

$CO_2$ fossil emissions reported by Member States are dominated by the energy sector (energy combustion and

fugitives) representing 92 % of the total EU27 + UK $CO_2$ emissions (excluding LULUCF) or 3.02 Gt $CO_2$ $yr^{-1}$ in 2019.
The Industrial Process and Product Use (IPPU) sector contributes 7.6 % or 0.2 Gt $CO_2$ $yr^{-1}$. $CO_2$ emissions reported
as part of the agriculture sector cover only liming and urea application, UNFCCC categories 3G and 3H[9] respectively.
Together with waste, in 2019 the emissions from agriculture represent 0.4 % of the total UNFCCC $CO_2$ emissions in
the EU27+UK.

An overview of all $CO_2$ fossil and land datasets in this work (Fig. 1) leads to a series of conclusions: 1)

Regardless of the method used (NGHGI, bottom-up models, top-down models), the timeseries of annual fluxes from
fossil $CO_2$ emissions rest almost one order of magnitude higher than removals from $CO_2$ uptake/removal by the land
surface and well outside uncertainty estimates; 2) Uncertainties are much larger in the LULUCF estimates than in the
fossil CO2 estimates (both for total LULUCF and for individual components of FL, CL, and GL); 3) Interannual
variability (IAV) is much more present in non-NGHGI LULUCF datasets than in NGHGI LULUCF datasets or any
of the fossil datasets.

The overall message that fossil $CO_2$ emissions exceed the land sink (Fig. 1a-c) is the same as found in the

Global Carbon Budget (Friedlingstein et al., 2022), although the difference is larger in the EU27+UK. Contrary to
the GCB, however, fossil $CO_2$ emissions in the EU27+UK have decreased over the past three decades. Again, this
finding is supported by the NGHGI, bottom-up models, and a single atmospheric inversion. Similarly, carbon uptake
by the land surface has remained more or less stable over the past three decades, with the vast majority of that occurring
in forests. While the latter conclusion is clear in the NGHGI (Fig. 1d), very large spreads among bottom-up sectorial
models lead to more uncertainty (bottom-center).

The difference in uncertainty between the estimates of fossil $CO_2$ emissions and $CO_2$ uptake/removal by the

land surface is also striking. Eight bottom-up models produce a 25-75 % percentile which is almost invisible on the
scale of the graph (center-top, gray shading). On the other hand, four models estimating Grassland emissions/removals
produce an error bar that covers the bottom part of the graph and masks any apparent trend (bottom-center, light green
shading). A similar conclusion can be drawn from top-down estimates of LULUCF fluxes (top-right, blue shading).
Additional work on reducing the uncertainty of LULUCF fluxes in the EU27+UK is highly welcome.

---

[9] 3G and 3H refer to UNFCCC category activities, as reported by the standardized common reporting format (CRF) tables, which contain $CO_2$ emissions from agricultural activities: liming and urea applications.

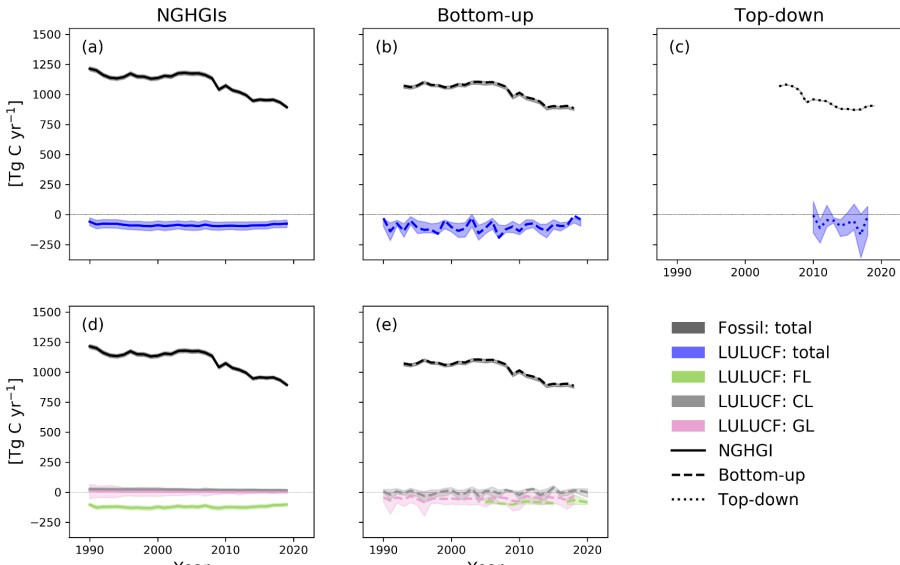


*Figure 1: A synthesis of all the $CO_2$ net fluxes shown in the work for the EU27+UK. The estimates are divided by*

*approach: NGHGI estimates (panels a, d); bottom-up methods (b, e); and top-down methods (c). Panels (d) and (e)*

*include a breakdown of the LULUCF flux into three of the dominant components: FL, GL, and CL. Such a breakdown*

*is not provided for NHGHI CO2 fossil as partitioning of bottom-up $CO_2$ fossil datasets corresponding to UNFCCC*

*NGHGI categories is not currently available. The NGHGI UNFCCC uncertainty is calculated for submission year*

*2021 as the relative error of the NGHGI value, computed with the 95 % confidence interval method gap-filled and*

*provided for every year of the timeseries, except for FL, GL, and CL which are taken directly from the EU NIR (2021).*

*Shaded areas for the other estimates represent the 0th-100th percentiles for groups with fewer than seven members,*

*and the 25th-75th percentile for groups with seven or more members. Ensembles (e.g., TRENDY v10) are included*

*in the above only their mean values, to avoid more heavily weighting the ensembles compared to the other datasets.*

Several caveats remain with this overall synthesis. First, the timeseries were combined rather naively in Fig. 1

by taking the mean of annual timeseries for each dataset discussed below. This leads to, for example, the 15-member

TRENDY ensemble being given identical weight as the ORCHIDEE high-resolution simulation over Europe. This

was done to weigh more heavily the regional approaches under the assumption that higher resolution simulations and

more region-specific input data will lead to more accurate results. While the latter assumption appears reasonable,

the first assumption can be disputed. Second, only a single top-down result for fossil $CO_2$ emissions is currently

available, preventing an estimate of the uncertainty for this approach. Third, sector models were combined

disregarding distinctions between those models estimating "Remain" and "Total" fluxes. These points are discussed



in more detail in the following sections.  However, addressing these points is highly unlikely to alter the overall
conclusions in this section.

**3.2. CO$_2$ fossil emissions**

The inventory-based fossil CO$_2$ estimates from nine data sources (and some subsets) are presented as timeseries
(1990-last available year) based on Andrew (2020) with the objective to explore differences between datasets and
visualize trends (Fig. 2).  Because the emissions source coverage (also called the "system boundary") of datasets
varies, comparing total emissions from these datasets is not a like-for-like comparison. Therefore, some harmonization
of system boundaries prior to comparison is needed. This harmonization relies on specifying the system boundary of
each dataset and, where possible, removing emission sources to produce a near-common system boundary.  For
example, if IEA doesn't include any carbonates, then carbonates are removed from all emissions datasets that report
these separately.  UNFCCC (CRFs) Energy+IPPU, CDIAC, CEDS, PRIMAP, and GCP include Energy sector plus
all fossil fuels in IPPU; EIA, EDGAR and BP include some fossil fuels in IPPU, while EIA and BP include bunker
fuels as well. UNFCCC CRFs include Energy total and Energy combustion.  Further details on how data sets are
harmonized are provided by Andrew (2020).  Because of differing levels of detail provided by datasets, it isn't possible
to do this perfectly, but the approximate harmonization gives something closer to a like-for-like comparison, with the
legend in Fig. 2 indicating the most significant remaining differences.  The pre-harmonization curves are shown in
Appendix A (Fig. A1) for reference.


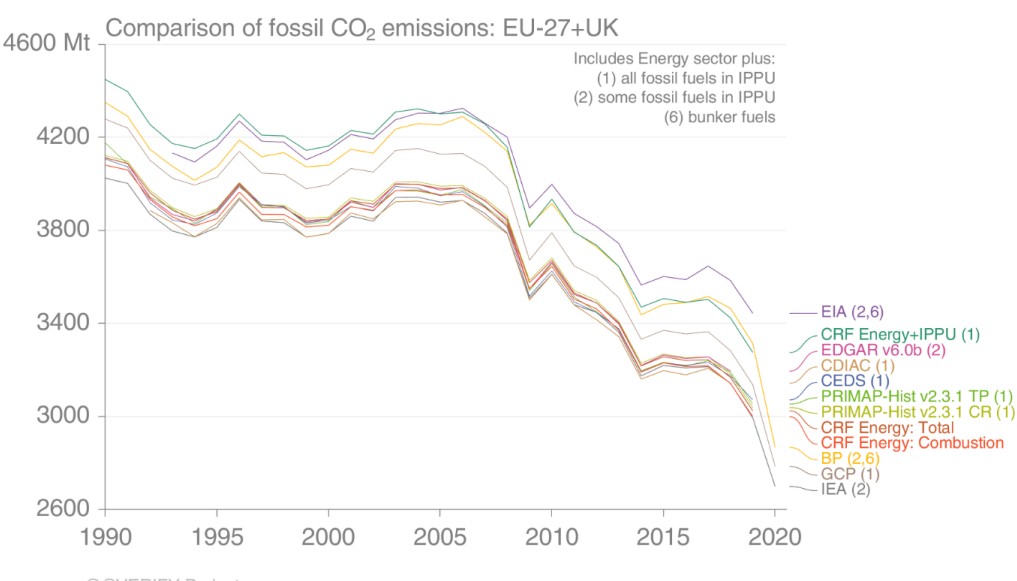






*Figure 2: Comparison of EU27+UK fossil $CO_2$ emissions from multiple inventory datasets with system boundaries*
*harmonized as much as possible. Harmonization is limited by the disaggregated information presented by each*
*dataset. CDIAC does not report emissions prior to 1992 for former-Soviet Union countries. CRF: UNFCCC NGHGI*
*from the Common Reporting Format tables. The pre-harmonization figure is shown in Fig. A1.*
Given the remaining differences in system boundaries after harmonization, most datasets agree well
(Andrew, 2020). In response to inconsistencies identified in this work, the EIA recently corrected some double-
counting of emissions from liquid fuels and has revised its estimates of total emissions down about 10 % for the
EU27+UK (pers. comm., US Energy Information Agency, February 2022). For comparison, applying a similar
harmonization procedure to the UNFCCC NGHGI and retaining only Fuel combustion (1A), Fugitive emissions (1B),
Chemical industry (2B), Metal industry (2C), Non-energy products from fuels and solvent use (2D), and Other (2H)
results in emissions of $3392 \pm 49$ Tg $CO_2$ yr$^{-1}$ ($926 \pm 13$ Tg C yr$^{-1}$) for the year 2017, where the uncertainty was
propagated through quadrature using the gap-filled uncertainties described in this work and taking the total sector
uncertainty if the category uncertainty was not available. This mean value falls within the 25th-75th percentiles of
the eight other harmonized BU sources ([3238,3401] Tg $CO_2$ yr$^{-1}$).
The sole available inversion for $CO_2$ fossil fluxes is produced by the CIF-CHIMERE model, shown in Fig.
1c and Fig. B3 (for a single year). The inversion yields plausible and consistent fossil emission estimates compared
to nine bottom-up estimates from BU datasets with global coverage (Fig. 1b,c,B3). Uncertainties of CIF-CHIMERE
inversion estimate have not yet been quantified, however they are likely largely driven by large uncertainties in the
input data. The satellite observations of $NO_2$ have large uncertainties, which partly explains the small departure from
the prior fluxes during the optimization. Emission ratios between $NO_2$ and $CO_2$ are also uncertain (those from the prior
are currently used). The atmospheric residence time of $NO_2$ is another major source of uncertainty. The inversion
reports total fossil $CO_2$ emissions calculated from NOx combustion emissions. However, in principle, the derivation
of $CO_2$ emissions from the NOx inversions should be restricted to fossil fuel $CO_2$ emissions based on the fossil fuel
$CO_2$/NOx ratio from the TNO, as there is a better-established relationship between $CO_2$ and NOx from combustion of
fossil fuels. Future inversions co-assimilating $CO_2$ data will make a clearer distinction in the processing of fossil-fuel
and other anthropogenic emissions. Finally, it's important to note that the inversion results are not fully independent
of the bottom-up methods, as the prior estimates are based on TNO gridded products. However, part of the lack of
departure from the prior can also be attributed to the general consistency between the prior and the observations, which
raise optimistic perspectives for the co-assimilation of co-emitted species with the data from future $CO_2$ networks
dedicated to anthropogenic emissions.



**3.3. $CO_2$ land fluxes**
This section updates the benchmark data collection of $CO_2$ emissions and removals from the LULUCF sector
in EU27+UK previously published in Petrescu et al. (2020) and Petrescu et al. (2021b), expanding on the scope of
those works by adding additional datasets and years. The countries analyzed in this study use country-specific activity





data and emissions factors for the most important land use categories and pools (EU NIR 2022, UK NIR 2022).
However, several gaps still exist, mainly in non-forest lands and non-biomass pools (e.g., EU NIR, 2022). In addition,
since NGHGIs largely rely on periodic forest inventories (carried out every five to ten years) for the most important
land use (Forest land), the net $CO_2$ LULUCF flux often does not capture the most recent changes, nor the full
interannual variability.
While the net LULUCF $CO_2$ flux was relatively stable from 1990 to 2016, staying mostly between -300 to -
350 Mt $CO_2$/yr, in the past three years the sink has weakened to around -250 Mt $CO_2$/yr in 2020 (black dotted line in
Fig. B2, Appendix B; Abad-Viñas, pers. comm, 2022). This weakening occurred mostly in Forest land, due to a
combination of increased natural disturbances, forest aging and increased wood demand (Nabuurs et al., 2013; EU
NIR, 2022). Natural disturbances, including fires (especially in the southern Mediterranean), windthrows, droughts
and insect infestations (especially in central and northern European countries), have increased in recent years (e.g.,
Seidl et al., 2014) which explains most of the interannual variability of NGHGIs.  Forest aging affects the net sink
both through the forest growth (net increment) - which tends to level off or decline after a certain age - and the harvest,
because a greater area of forest reaches forest maturity (Grassi et al., 2018b).  Although the exact increase in total
harvest in Europe in recent years is still subject to debate (Ceccherini et al., 2020; Palahi et al. et al., 2021), demand
for fuelwood at least has increased (Camia et al., 2021).
Carbon uptake as seen by the atmosphere may occur on either managed or unmanaged land, and results from
processes such as photosynthesis, respiration, and disturbances (e.g., fire, pests, harvest).  As discussed by Petrescu et
al. (2020), the fluxes reported in NGHGIs relate to emissions and removals from direct LULUCF activities (clearing
of vegetation for agricultural purposes, regrowth after agricultural abandonment, wood harvesting and recovery after
harvest and management) but also indirect $CO_2$ fluxes due to processes such as responses to environmental drivers on
managed land. Additional $CO_2$ fluxes occur on unmanaged land, but these fluxes are very small in Europe. According
to Table 4.1 in the EU27 and UK NGHGIs (2021) CRF, almost all land (~95 %) in the EU27+UK is considered
managed.  France and Greece report some unmanaged forest areas (1.1 % and 16.6 %, respectively). Hungary and
Malta report unmanaged Grassland areas of 33 % and 100 %, respectively, and Nordic and Baltic countries plus
Ireland, Slovakia and Romania report sometimes quite large (up to 100 %) unmanaged wetland areas.
The indirect $CO_2$ fluxes on managed and unmanaged land due to changing climate, increasing atmospheric
carbon dioxide concentrations, and nitrogen deposition, are part of the (natural) land sink in the definition used in
IPCC Assessment Reports and the Global Carbon Project's annual global carbon budget (Friedlingstein et al., 2022),
while the direct LULUCF fluxes are termed "net land-use change flux", as discussed by Grassi et al. (2018a, 2021,
2022a), Petrescu et al. (2020, 2021b) and Pongratz et al. (2021). Results should thus be interpreted with caution due
to these definitional differences, but as most of the land in Europe is managed and the indirect effects are small, the
definitional differences should be modest compared to other sources of uncertainty (Petrescu et al., 2020). Other
relatively recent studies have already analyzed the European land carbon budget using GHG budgets from fluxes,
inventories and inversions (Luyssaert et al., 2012) and from forest inventories (Pilli et al., 2017; Nabuurs et al., 2018).

*3.3.2. LULUCF CO$_2$ fluxes from NGHGI and decadal changes*

Figure 3 shows the decadal change in CO$_2$ LULUCF flux from the UNFCCC NGHGI (2019) (upper plot)

compared with the UNFCCC NGHGI (2021) (bottom plot). The contribution of each category ("remaining" and
"conversion") to the overall reduction of CO$_2$ emissions in percentages between the three mean periods (gray columns)
are the mean values over 1990–1999, 2000–2009 and 2010–(2017) 2019. The "+" and the "−" signs represent a source
and a sink to the atmosphere. LUC(−) represents the land use conversion changes that increase the strength of the
LULUCF sink between two averages (i.e., values become more negative); LUC(+) represents the land use conversion
changes that decrease the strength of the overall LULUCF sink. Note that the categories inside LUC(−) may be sources
or may be sinks, but between the two average periods, they become more negative. The HWP pool can constitute
either a source or a sink depending on the balance between the timber input to the pool (contributes to a sink) and the
loss of carbon as products reach their end-of-life (source). The absolute contributions of each category to the total
LULUCF fluxes for 1990-2019 are given in Fig. B2 for context.
From the 1990–1999 mean to the 2000–2009 mean, the CO$_2$ LULUCF flux changed from -87.98 to -96.98 Tg C
in the 2021 NGHGI (i.e., strengthened by 10.0 %), compared with -10.7 % for the 2019 NGHGI(note that Petrescu et
al. (2021b) reported -9.6 %, which is the change relative to the 2000-2009 mean instead of the 1990-1999 mean that
we adopt here due to common usage). This indicates a slight decrease in the reported European land sink compared
to the previous estimates due to revised historical estimates. A 3.8 % growth in emissions from FL-FL and LUC(+)
(Wetlands, Settlements and Other land conversions) weakened the overall sink[10,] while the sink related to all other
categories grew by 15 % to strengthen the overall sink[11].
From the 2000-2009 mean to the 2010–2019 mean, the CO$_2$ LULUCF flux changed by +3.7 % (i.e., weakened
sink), compared with +3.4 % reported by Petrescu et al., (2021b) which denotes a slight weakening of the European
land sink compared to the previous estimate. Note the difference in time period (2010-2019 here, but 2010-2017
previously). A 9.6 % growth in emissions from FL-FL, HWP and LUC(+) (Forest land, Wetlands, and Settlements
conversions) weakened the overall sink[12], while the sink related to all other categories changed by -5.9 % and
strengthened the overall sink[13].

---

[10]Positive percentages represent sources

[11]Negative percentages represent sinks.

[12]Positive percentages represent sources

[13]Negative percentages represent sinks.

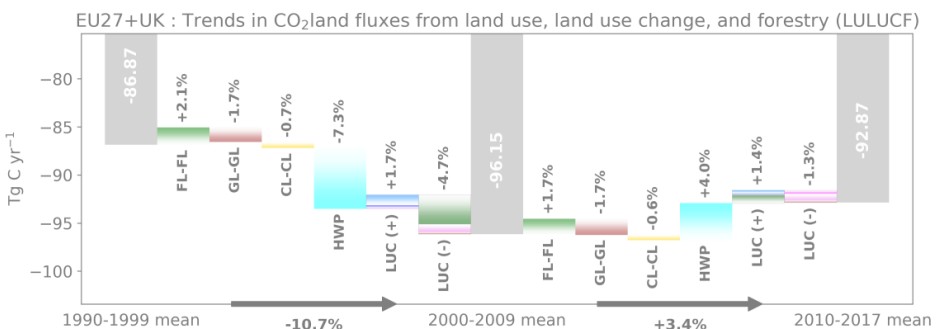

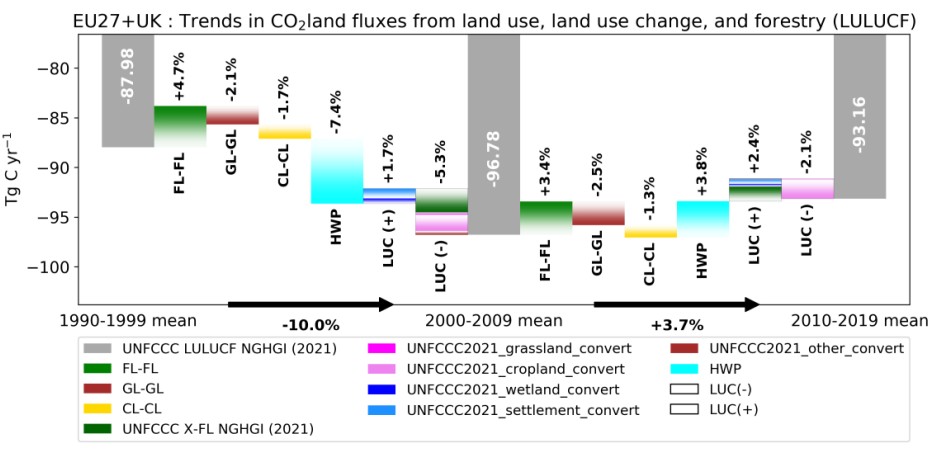


*Figure 3: The contribution of changes (%) in $CO_2$ land fluxes from various LULUCF categories to the overall change*
*in decadal mean for the EU27+UK as reported by Member States to the UNFCCC. The top plot shows the previous*
*NGHGI data from Petrescu et al. (2021b) and the bottom plot illustrates data from UNFCCC NGHGI (2021). Changes*
*in land categories converted to other land are grouped to show net gains and net losses in the same column, with the*
*bar color dictating which category each emission belongs to; note that the composition of the "LUC(+)" and*
*"LUC(−)" bars can change between time periods. Not shown are emissions from "Wetlands remaining wetlands",*
*"Settlements remaining settlements", and "Other land remaining other land" as none of the BU models used*
*distinguish these categories. The fluxes follow the atmospheric convention, where negative values represent a sink*
*while positive values represent a source. The color bars are shaded to guide the eye in the direction of the change*
*(white-to-color).*

Similar to Petrescu et al. (2021b), changes of HWP emissions remain by far the major contributor to changes in
the LULUCF sink strength, but the direction of their contribution is opposite across the two periods: from
strengthening the sink during 1990–1999 to 2000–2009 to reducing the sink in 2010-2019. However, the balance



between HWP and FL-FL is quite sensitive to the periods selected: for the difference between 1993-2001 and 2002-
2010, FL-FL contributes more (+7.3 %) than HWP (-5.2 %). EU-27+UK Member States have all implemented the
IPCC Approach B (i.e., production approach) for the HWP pool (EU NIR, 2021), which "inventories carbon in wood
products from domestically harvested wood only and does not provide a complete inventory of wood carbon in
national stocks" (Volume 4, Chapter 12, IPCC, 2006). Figure 3 suggests that carbon emissions from HWP "end-of-
life" became greater than the amount of carbon entering HWP from domestic harvest in recent decades. If the flux of
carbon into the HWP (a portion of domestic harvest) decreases, there will be a lag effect where outputs (due to wood
product end-of-life) may dominate, leading to a source from HWP. This is confirmed by a more detailed analysis of
the reported gains and losses for the bloc (see Figure A2 in the Appendix), which shows a drop in harvested wood
product gains around 2008 followed by a slower recovery compared to the pre-2008 trend. Gross losses from the
HWP pool, on the other hand, have been increasing as HWP produced pre-2008 reach their end-of-life, leading to a
weakened sink from 2009 onwards compared to during the mid-2000s.

### 3.3.3. Estimates of CO$_2$ land fluxes from bottom-up approaches
In this section we present annual total net CO$_2$ land emissions between 1990-2020 i.e., induced by both
LULUCF and natural processes (e.g. environmental changes) from class-specific models as well as from models that
simulate multiple land cover/land use classes. The definitions of the classes may differ from the IPCC definitions of
LULUCF (e.g., FL, CL, GL) where, according to IPCC 2006 guidelines, to become accountable in the NGHGI under
"remaining" categories, a land-use type must be in that class for at least N years (where N is the length of the transition
period; 20 years by default). In an effort to create the most accurate comparison as possible in terms of categories and
processes included, total Forest land (FL) has been divided up into Forest land remaining forest land (FL-FL) and
Land converted to forest land (X-FL), while only total Grassland (GL) and Cropland (CL) are reported. This is largely
due to the non-forest sector models explored here only considering net land use change, which prevents separating out
the "converted" component.

*Forest land*
Fluxes from **Forest land** which remain in this class (FL-FL) are shown in Fig. 4. These fluxes were simulated
with ecosystem models (CBM and EFISCEN-Space, described in more detail in the Appendices) and countries'
official inventory statistics reported to UNFCCC. The results show that the differences between models are systematic,
with CBM having slightly weaker sinks than EFISCEN-Space. CBM updated its historical data (1990-2015) and
presents new NBP estimates based on extrapolation of historical timeseries (see Appendix A2) for 2017-2020
(CBMsim). Both CBM and EFISCEN-Space use national forest inventory (NFI) data as the main source of input to
describe the current structure and composition of European forests. NFIs are also the main source of input data for
most countries in the EU27 for NGHGIs (EU NIR, 2021), including data for carbon stock changes in various pools as
well as the estimation of forest areas. Given that EFISCEN-Space does not cover all countries in the EU27+UK
(Austria, Bulgaria, Denmark, Hungary, Lithuania, Portugal and Slovenia are missing), the results were scaled by
1/0.74 to account for the fact that the available countries comprise around 74 % of the forest NBP for the EU27+UK,



according to previous EFISCEN results (Petrescu et al., 2021b). As noted above, EU regulations are driving Member
States to report spatially explicit NGHGIs. Unlike the original EFISCEN, EFISCEN-Space is a spatially explicit
model, in addition to being able to simulate a wider variety of stand structures, species mixtures and management
options. Note that EFISCEN-Space reports only a single mean value for forest fluxes from 2005-2020; the annually
varying value shown in Fig. 4 arises from scaling by annually varying forest areas.

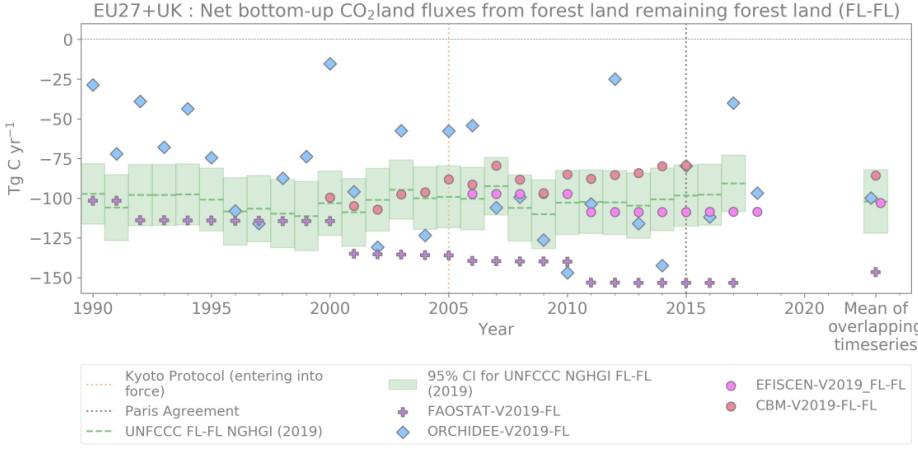

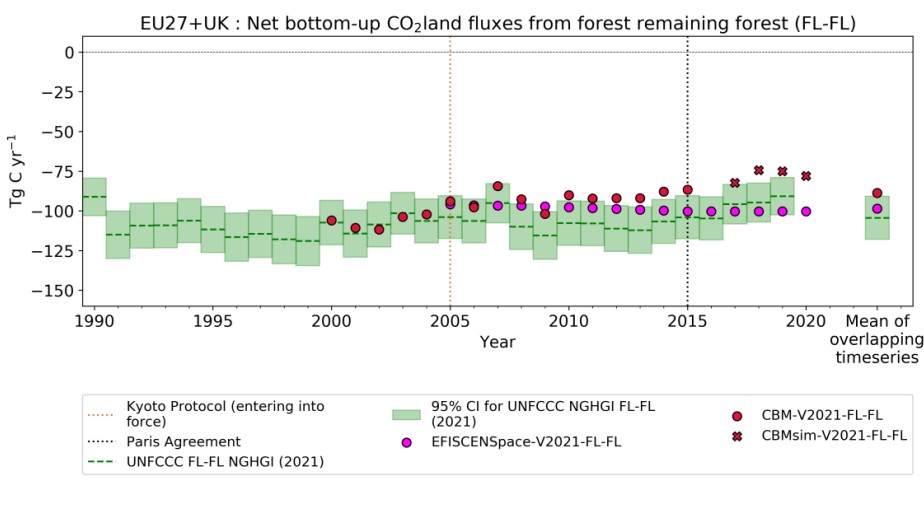


*Figure 4: Net CO$_2$ land flux from Forest land remaining forest land (FL-FL) estimates for EU27+UK CO$_2$ from the*
*Petrescu et al. (2021b) synthesis paper (top) and a comparable graph using the updated data this year (bottom).*
*Means are given for 2006-2015 (top) and 2005-2019 (bottom) on the right side of both plots. CBM FL-FL historical*
*estimates include 25 EU and UK countries (excl. Cyprus and Malta) and include new estimates for 2017-2020 (red*





The UNFCCC NGHGI uncertainty of $CO_2$ estimates for FL-FL across the EU27+UK, computed with the
error propagation method (95 % confidence interval) (IPCC, 2006), ranges between 34 % - 55 % when analyzed at
the country level for all years, as it varies as a function of the component fluxes (EU NIR, 2019). Despite contrasting
methodologies and input data for emission calculation and uncertainties in each method (Appendix A), there is
reasonable agreement on the trend in FL-FL fluxes from CBMsim and the UNFCCC NGHGI (2021) (Fig. 4).  The
magnitude of the values between EFISCEN-Space and the NGHGI (2021) also agree well, though as noted above the
EFISCEN-Space results only vary with the amount of forest area which makes the trend much flatter.  Given that all
three methods (NGHGI, CBM, and EFISCEN-Space) are heavily based on national forest inventory data, the general
agreement between the three is not surprising.
Figure 5 presents $CO_2$ land estimates for total Forest land (both remain and convert classes, "FL"). For the
total Forest land, the results were simulated with an ecosystem model (ORCHIDEE) and a global dataset (FAOSTAT)
as it is not possible for these two approaches to separate out the "remain" and "convert" land use category.  This
obstacle arises due to the use of net land use/land cover information which does not include detailed information on
the nature of the conversions.  Consequently, Fig. 5 compares them to the total Forest land from the countries' official
inventory statistics (UNFCCC NGHGI, 2021).
From 2001 and until 2010, the FAOSTAT reports an increasing sink over time, which weakens from 2011
until 2019 (Fig. 5). This is explained by a reporting inconsistency in the Romanian inventory which had not been
corrected at the time of this analysis.  Therefore, Romanian estimates for Forestland and Net forest conversion have
been removed for the whole 1990-2020 timeseries in Fig. 5. Starting in 2016, FAOSTAT estimates better match those
from the NGHGIs as FAOSTAT updated its estimates. FAOSTAT uses input data directly from country submissions
to the FAO Global Forest Resource Assessments (FRA[14]) (e.g., carbon stock change is calculated by FAO directly
from carbon stocks and area data submitted by countries).  It is important to note that these data are not always identical
to those submitted to the UNFCCC (Tubiello et al., 2021).

---

[14]The Global Forest Resource Assessment (*FRA*) is the supplementary source of Forest land data disseminated in *FAOSTAT,*
http://www.FAO.org/forestry/fra/en/



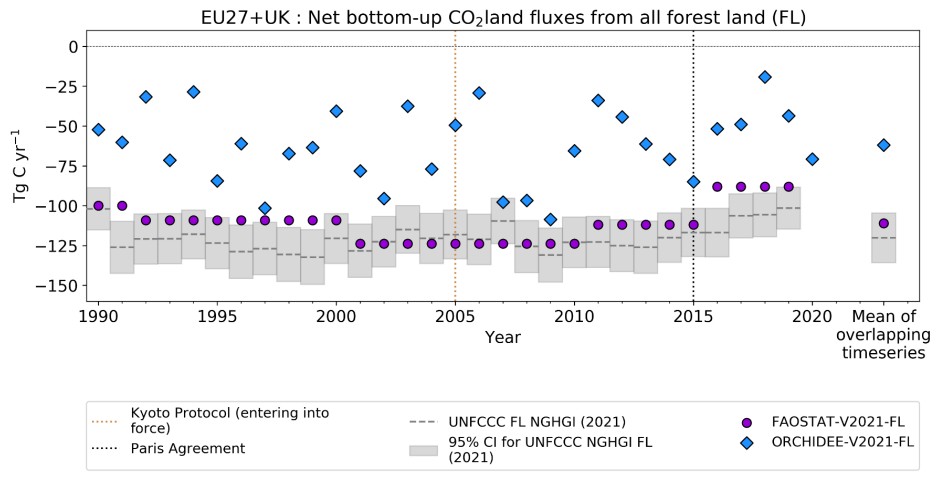

*Figure 5: Net CO₂ land flux from total Forest land estimates (FL) for EU27+UK CO₂ from the UNFCCC NGHGI (2021) submissions, the FAOSTAT data-driven inventory, and the ORCHIDEE DGVM. The relative error on the UNFCCC value represents the UNFCCC NGHGI (2021) MS-reported uncertainty with no gap-filling (EU NIR, 2021). FAOSTAT data does not include Romanian inventory estimates. The means are calculated for the 1990–2019 overlapping period. The fluxes follow the atmospheric convention, where negative values represent a sink while positive values represent a source.*

ORCHIDEE was updated to include a dynamic nitrogen cycle coupled to the carbon cycle in this work. As shown in Appendix A2, the coupled nitrogen cycle results in a stronger sink, even if identical forcing is used. ORCHIDEE shows a high inter-annual variability in carbon fluxes for forests in Fig. 5 because it incorporates meteorological data at sub-monthly timescales, while methods based on forest inventories are generally updated only every few years (e.g, five years for FRA), which results in a more climatological perspective. ORCHIDEE results indicate that climatic perturbations and extreme events (multi-month droughts, in particular) can have significant impacts on the net carbon fluxes depending on their timing in relation to the growing season. This is in line with flux tower measurements that show significant year to year variability (Ciais et al. 2005). This is also to some extent supported by dendrometer data although such data varies greatly among sites and tree species which obscures a significant net effect (Scharnweber et al., 2020). It should also be noted that dendrometer data measures carbon stored in individual trees, while the NBP reported in figures in this paper include fluxes from litter and soil respiration. The variability of the weather data affects the carbon dynamics of all components of the ecosystems (hence NBP), which, for instance, impacts on C assimilation rates, length of the growing season, dynamics of respiration rates and allocation of the carbon in the plant (cf. Fig. 1 and 2 in Reichstein et al. (2013) and Bastos et al. (2020b)).

A few reasons for differences between estimates seen in Fig. 4 and 5 can be readily identified. For this study, the ORCHIDEE model used the ESA-CCI LUH2v2 PFT distribution (a combination of the ESA-CCI land cover map



for 2015 with the historical land cover reconstruction from LUH2 (Lurton et al., 2020)), and assumes that the shrub
land cover classes are equivalent to forest. In terms of area, the original ESA-CCI product corresponding to the
EU27+UK shows shrub land equal to about 50 % of the tree area in 2015. A similar analysis using the FAOSTAT
domain Land Cover, which maps and disseminates the areas of MODIS and ESA-CCI land cover classes to the SEEA
land cover categories[15], shows that shrub-covered areas are around 20 % of that of forested areas for the
EU27+UK. The impact of classifying shrubs as "forests" on the total carbon fluxes could therefore account for a
significant percentage of the differences between ORCHIDEE and other results in Fig. 5. In addition, CBM depends
strongly on input data and related uncertainty. Historical data are retrieved from both country and EU statistics and
usually refers to forest management units rather than individual inventory plots. Finally, trends in forest carbon
strongly result from management, which are not represented in this version of ORCHIDEE but are included in CBM
and EFISCEN-Space.


*Cropland*
Cropland (CL, represented in the UNFCCC NGHGI 2021 as UNFCCC category 4B) includes net $CO_2$
emissions and removals from soil organic carbon (SOC) under "remaining" and "conversion" categories. Figure 6
shows the annual fluxes belonging to the category CL from the NGHGI for the EU27+UK along with four other
approaches: one bottom-up inventory (FAOSTAT), two sector-specific models (EPIC-IIASA, ECOSSE), and one
DGVM (ORCHIDEE). Note that the FAOSTAT value only includes the carbon flux from organic soils drained for
agriculture, while ECOSSE, EPIC-IIASA, and ORCHIDEE include biomass volatilized immediately upon harvest;
biomass left on site to decay as litter; and soil organic carbon.
The previous synthesis of Petrescu et al. (2021b) (Fig. 6, top) compared models against results for GL-GL
from the NGHGI. For the current work, we compare against the total Grassland values (GL). The reason for this is
that FAOSTAT, ECOSSE, EPIC-IIASA, and ORCHIDEE all use land use/land cover maps generated by IPCC
Approach 1, which only records the total amount of land in a category for each year; information on transitions
between categories is unknown. Therefore, it is not possible to separate out "remain" and "convert" categories.
For the common period (1990-2019), ORCHIDEE simulates a mean sink of -26 Tg C yr[-1], while ECOSSE,
EPIC-IIASA, and FAOSTAT all simulate mean sources of 21 Tg C yr[-1], 10 Tg C yr[-1] and 16 Tg C yr[-1], respectively.
With the exception of ORCHIDEE, all models are in line with the NGHGI results (mean over the same period of 22
Tg C yr[-1]). In Petrescu et al. (2021b) (Fig. 6, top) the NGHGI reported a very small but constant source over the whole
period (mean of 5.6 ± 3.5 Tg C yr[-1]) with almost no inter-annual variability by construction, while all three process-
based models simulated a sink.
The sink in ORCHIDEE must arise from the soil, as no simulated biomass in croplands remains from year to
year; carbon is assimilated into biomass growth during the growing season, after which the biomass dies, is partitioned
between litter and harvest (50 % to each), and either decays or vaporizes, respectively. In other words, no woody or
perennial crops are simulated. NGHGIs assume that all aboveground biomass of non-woody crops re-enters the

---

[15] http://www.fao.org/faostat/en/#data/LC





atmosphere at harvest. Given more favorable growing conditions due to climatic changes and $CO_2$ fertilization, this leads to more carbon entering the soil in ORCHIDEE in recent decades, which is driving the calculated CL sink observed in the model.

In the NGHGI, the reported source for the EU27+UK is mostly attributed to emissions from cropland on organic soils[16] in the northern part of Europe where $CO_2$ is emitted due to C oxidation from tillage activities and drainage of peat. The fact that FAOSTAT values are similar to the UNFCCC values points to the primary role of drained organic soils, as this is the only flux included for the FAOSTAT dataset in Fig. 6. Finland and Sweden are of particular importance, as they together account for more than half of the total area of organic soil in Europe. Organic soils are an important source of emissions when they are under management practices that disturb the organic matter stored in the soil. In general, the NGHGI emissions from these soils are reported using country-specific values when they represent an important source within the total budget of GHG emissions.

ORCHIDEE also shows a much larger year-to-year variation due to the response of vegetation and respiration fluxes to sub-daily meteorology. EPIC-IIASA and ECOSSE both operate on daily timescales (ECOSSE was updated to daily for this work, though the previous version was monthly). As both photosynthesis (e.g., Kumarathunge et al., 2019) and respiration (e.g., Yvon-Durocher et al., 2012) show non-linear dependence on temperature, the more extreme temperatures experienced by plants in ORCHIDEE will lead to a higher variation in vegetation response given the same photosynthetic model. High IAV can be seen clearly for drought impacts in ORCHIDEE where regions change from sources to sink in a single year (e.g., for 2003 and 2018 (Ciais et al., 2005; Bastos et al., 2020a)). The other two ecosystem models follow ORCHIDEE's patterns but with smaller magnitudes. FAOSTAT and NGHGIs are mostly insensitive to inter-annual variability as the estimations are mainly based on statistical data for surfaces/activities and emission factors that do not vary with changing environmental conditions.

Both ECOSSE and EPIC show a striking improvement in agreement with the NGHGI between V2019 (Fig. 6, top) and the current work (Fig. 6, bottom). For ECOSSE, this is the result of improved data, in particular around residue management. The aboveground biomass is divided into harvest (which is accounted as direct emissions) and residues (biomass that is partly removed and partly left on the field). The external tool MIAMI serves as the central model for the NPP and follows the allocation distribution of Neumann and Smith (2018). The removed residues are set to 50 % as a compromise between the wide range of residue removal rates given by Scarlat et al. (2010). Residue and yield biomass from MIAMI are provided as input into the ECOSSE simulations. Additionally, more realistic fertilizer data (Mueller et al., 2012) were used. For EPIC, the shifts in net $CO_2$ fluxes in the current EPIC results stem from the updated soil organic carbon and nitrogen module (Balkovič et al., 2020) and updates in meteorological forcing. Firstly, the updated soil module resulted in higher heterotrophic respiration across many EU regions. Besides attributing more carbon to the soil surface emissions, enhanced respiration leads to higher NPP and yields in regions with low fertilization rates as more nitrogen is released from the SOM pool. Secondly, altered solar radiation and air

---

[16]The 2006 IPCC Guidelines largely follow the definition of Histosols by the Food and Agriculture Organization (FAO), but have omitted the thickness criterion from the FAO definition to allow for often historically determined, country-specific definitions of organic soils (see Annex 3A.5, Chapter 3, Volume 4 of IPCC (2006) and Chapter 1, Section 1.2 (Note 3) of IPCC (2014)).

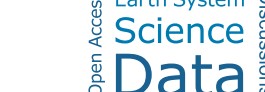

temperature data affected the full range of carbon variables in EPIC, including NPP, harvested biomass, heterotrophic
respiration, and leached carbon.

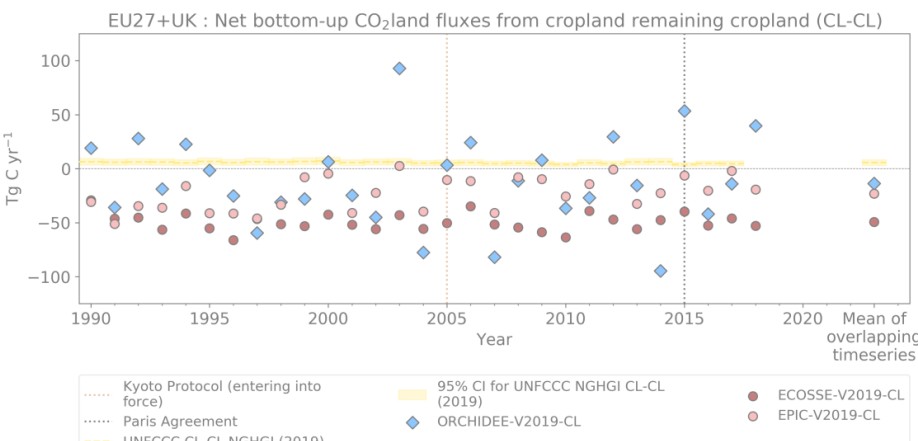

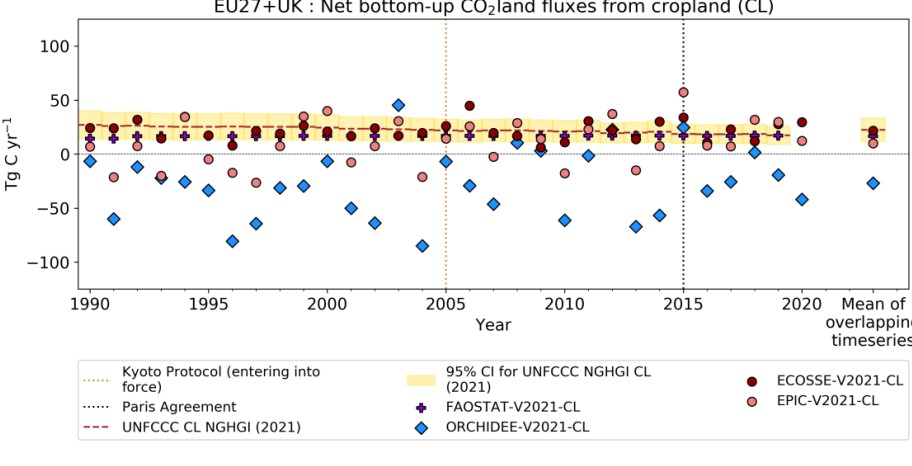


*Figure 6: Net $CO_2$ land flux from Cropland estimates for the EU27+UK from: previous data from Petrescu et al.,*
*(2021b) showing only the "remaining" fluxes (CL-CL) (top plot), and data from the UNFCCC NGHGI (2021)*
*submissions and models showing net carbon fluxes for the total Cropland (CL), with their 1990-2019 mean given on*
*the right (bottom plot). CL net carbon fluxes are estimated with three ecosystem models: ORCHIDEE, ECOSSE and*
*EPIC-IIASA, in addition to the FAOSTAT inventory. Note that the FAOSTAT value only includes the carbon flux from*
*organic soils drained for agriculture. The relative error on the UNFCCC value represents the UNFCCC NGHGI*
*(2021) MS-reported uncertainty with no gap-filling (EU NIR, 2021). The fluxes follow the atmospheric convention,*
*where negative values represent a sink while positive values represent a source.*




Finally, differences in the results between the models and the NGHGIs may arise from definitions. The
cropland definition in the IPCC includes cropping systems and agroforestry systems where vegetation falls below the
threshold used for the definition of Forest land category, consistent with the selection of national definitions (IPCC
glossary). Given that every country is allowed to select their definition of Forest land, which therefore influences the
area of Cropland and the total emissions, it is beyond the scope of this study to summarize here the criteria for the 28
countries under consideration and compare those to the methods used in determining the land use/land cover data for
the other models. However, the interested reader is referred to Tables 6.10 (forests), 6.18 (croplands), and 6.22
(grassland) in the 2022 NIR of the European Union (EEA/PUBL/2022/023).

***Grassland***
Grassland (GL, UNFCCC category 4C) includes net $CO_2$ emissions and removals from soil organic carbon
(SOC) under "remaining" and "conversion" categories. The grassland definition in the IPCC includes rangelands and
pasture land that is not considered as Cropland, as well as systems with vegetation that fall below the threshold used
in the Forest land category (same explanation as for Cropland). This category also includes all grassland from wild
lands to recreational areas as well as agricultural and silvo-pastoral systems, subdivided into managed and unmanaged,
consistent with national definitions (Petrescu et al., 2021b). For similar reasons to those expressed in the section
Cropland above, the current work (Fig. 7, bottom) compares modeled $CO_2$ flux against NGHGI results for total
Grassland (GL).
The NGHGIs of countries in the EU27+UK report emissions from managed pastures and grasslands, although
the details of what is included varies between countries (Table 6.21, EU NIR, 2021). Grasslands can be managed
through grazing or by cutting. If a grassland is used for grazing but retains the natural vegetation, it is called a
"rangeland". If the area has been replanted with vegetation specifically for animal forage, it is commonly referred to
as "pasture"[17]. Since almost all European grasslands are somehow modified by human activity and to a major extent
have been created and maintained by agricultural activities, they can be defined as "semi-natural grasslands", even if
their plant communities are natural (Silva et al., 2008).
The NGHGI reports a slightly positive net flux over 1990-2019, although with a much larger uncertainty than
for either Forest land or Cropland ($4 \pm 28$ Tg C yr$^{-1}$). While increased uncertainty compared to forest emissions is
understandable given the emphasis on collecting accurate forestry statistics due to their economic importance, the
increased uncertainty in Grassland compared to Cropland is more puzzling. Three possible explanations include: 1)
absolute Grassland emissions/removals are lower than for Cropland, which may lead to higher relative uncertainty
given the nearness to zero; 2) MS with lower uncertainties may dominate Cropland, while MS with higher
uncertainties may dominate Grassland; 3) Extensive work has been carried out on national/regional factors
representing changes in Cropland management, while less has been done on Grassland. For (3), this also may apply
to other biomass pools, as eight countries report "country specific" instead of "default" parameters for living biomass

---

[17] See, for example, https://www.epa.gov/agriculture/agricultural-pasture-rangeland-and-grazing




in Cropland versus Grassland (while only one country does the reverse; Table 6.6., EU NIR, 2021). Additional
analysis will be needed to elucidate this issue.


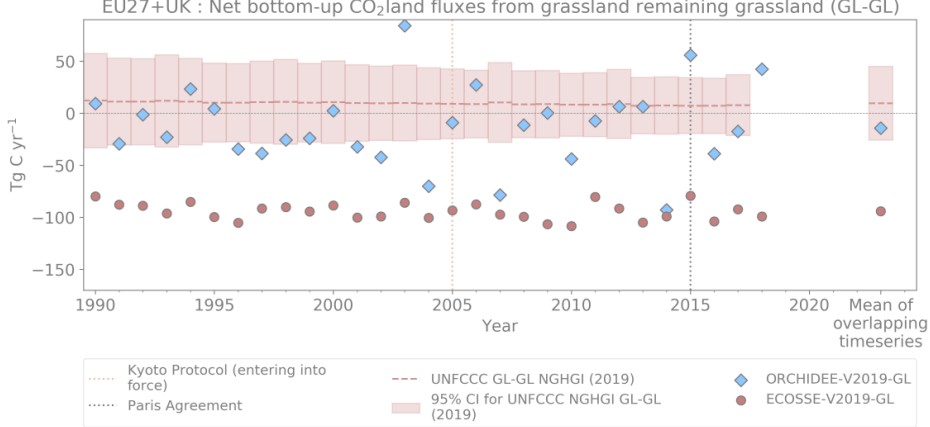

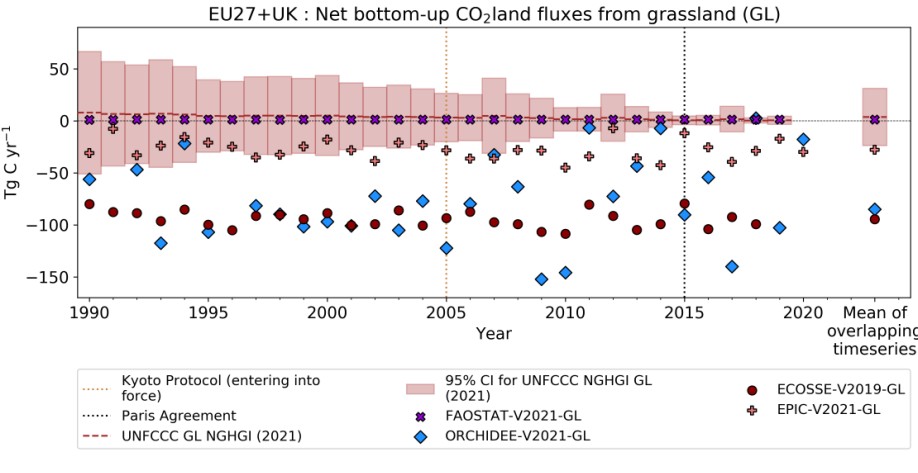


*Figure 7: Net CO₂ land flux from total Grassland (GL) estimates for EU27+UK from: previous data from Petrescu et*
*al. (2021b) (top plot), and the updated datasets considered here (bottom plot). The means shown on the right of each*
*plot are for 1990-2017 (top) and 1990-2018 (bottom). GL net carbon fluxes are estimated with the ORCHIDEE, EPIC-*
*IIASA, and ECOSSE (not updated and therefore identical to Petrescu et al., 2021b) models in addition to FAOSTAT.*
*The relative error on the UNFCCC value represents the UNFCCC NGHGI (2021) MS-reported uncertainty with no*
*gap-filling (EU NIR, 2021). The fluxes follow the atmospheric convention, where negative values represent a sink*
*while positive values represent a source.*




In addition to the NGHGI, updated results are available for ORCHIDEE (using a coupled C-N cycle) and
FAOSTAT. For the first time, EPIC-IIASA contributed estimates for Grassland fluxes using five different grassland
types and simulating carbon export due to herbivores (see Appendix A2 for more details). Both of these models
exhibit a strong sink in Grassland. For ORCHIDEE, this is likely due to the same reasons as the sink in croplands:
more suitable growing conditions due to climate change, $CO_2$ fertilization, and nitrogen deposition leading to increased
inputs into the soil which are not lost during tillage due to the lack of explicit management in the version reported
here. For EPIC-IIASA, this results from manure left on site and incorporated into the soil. A Tier 1 IPCC approach
assumes no changes in either living or dead biomass pools on grasslands; only considers organic soils which have
been drained for grazing; and only considers mineral soils which have undergone a change in management. This
greatly reduces or eliminates mechanisms which promote sinks in ORCHIDEE and EPIC-IIASA. On the other hand,
FAOSTAT reports a slight source in Grasslands, in line with the NGHGI. This is because, as is the case for Cropland,
FAOSTAT data only considers emissions from drained organic soils. As incorporation of manure in EPIC-IIASA
changes grasslands from a net source to a net sink, consideration of $CO_2$ from manure input in other inventories may
have a similar effect.

***3.3.4. Bottom-up $CO_2$ estimates from all LULUCF categories***

This section analyzes $CO_2$ emissions and sinks for the LULUCF sector, including NGHGI categories (from
Fig. 3) and a suite of different bottom-up approaches. This comparison is challenging due to differences in terms of
activities covered in the different estimates, as well as differences in terminology (see, for example, Petrescu et al.,
2020, Fig. 12). To summarize:
● FAOSTAT differs from NGHGIs for reasons recently summarized by Tubiello et al. (2021), Petrescu et al.
(2021b), and Grassi et al, (2022a), including numerically different data provided by Member States to
FAOSTAT and UNFCCC; different methods (FAOSTAT applies a Tier 1 approach globally, while Member
States reports to the UNFCCC vary from Tier 1 to Tier 3); differences between net and gross land use change
(FAOSTAT is based on net transitions, following Approach 1 as detailed by the 2006 IPCC guidelines
(Chapter 3 of Volume 4, Sect. 3.3.1)); and differences in biomass pools. For the latter, FAOSTAT only
considers living biomass pools instead of the five IPCC pools[18] reported to the UNFCCC. A preliminary
examination shows that changes in dead wood, litter, and mineral soil carbon stock are generally less than
0.1 t C/ha, which is relatively small compared to reported changes around 1.0 t C/ha in living biomass pools
(Tables 6.13, 6.14, 6.15, EU NIR, 2021). On the other hand, changes in organic soil carbon stock are
approximately the same magnitude as living biomass, which may lead to significant discrepancies between
the NGHGI and FAOSTAT for the EU27.

---

[18] According to the IPCC 2006 guidelines the reporting is done for the five LULUCF carbon pools: above-ground biomass, belowground
biomass, dead wood, litter, and soil organic matter





•    DGVMs (represented here by the TRENDY v10 ensemble, as well as the high-resolution ORCHIDEE and
CABLE-POP simulations) include the impact of $CO_2$ fertilization, climate change and land use change for
Forest land, Grassland and Cropland categories; they do not explicitly treat the Wetlands, Settlement and
Other land categories as in the NGHGIs. They account for the evolution of living biomass, dead biomass,
and soil organic carbon for all categories while for NGHGIs reporting is not mandatory for all subcategories
depending on the method Tier employed (e.g., dead organic matter in a Tier 1 method is assumed to be
constant). There is significant uncertainty associated with the DGVMs' fluxes both from i) the forcing data,
including datasets of land-use changes and the coverage of different land use change practices, ii) model
parameters, and iii) model structural uncertainty (i.e., processes not included) (Arneth et al., 2017). Similar
to FAOSTAT, DGVMs typically deal with net land use change emissions at the spatial resolution of the
model simulations (e.g., 0.5° or 1° for the TRENDY ensemble and 0.125° for the ORCHIDEE and CABLE-
POP simulations) instead of gross land use change as reported in NGHGIs. CABLE-POP is an exception to
most DGVMs and actually incorporates gross land use transitions (Haverd et al., 2018). The use of gross land
use transitions may induce significant differences with coarse resolution model simulations (e.g., the
TRENDY ensemble). In addition, DGVMs often do not distinguish between managed and unmanaged land,
while NGHGIs report results only from managed land.
•    The bookkeeping models, BLUE and H&N, calculate net emissions from land use change including
immediate emissions following land conversion, legacy emissions from slash and soil carbon decomposition
after land-use change, carbon uptake during regrowth of secondary forest after pasture and cropland
abandonment, and emissions from harvested wood products as they decay. While activities on the category
Land remaining land are generally not considered in bookkeeping models, one major exception is fluxes from
wood harvest, which are a primary source of emissions on managed forest land. In addition, bookkeeping
models do not account for fluxes arising from "indirect" anthropogenic influences such as $CO_2$ fertilization
or climate change.


Given all these differences in terms of activities, the comparison in this section should be considered as a
rough overview that highlights both important aspects of the C cycle and questions that need to be addressed in the
future. Going towards a more specific comparison of only net land-use change (LUC) fluxes would require additional
considerations. In GCP's annual global carbon budget, net LUC term is estimated by global DGVMs as the difference
between a run with and a run without land-use change (i.e., the S3 and S2 simulations from TRENDY, respectively)
and by bookkeeping models (Friedlingstein et al., 2022). Such an estimate is given in Fig. 13 in Petrescu et al. (2020)
for Forest land. However, this approach does not fully resolve the differences mentioned above. In particular,
questions remain about net vs. gross land use change, managed vs. unmanaged land, and emissions from wood
harvest. In addition, UNFCCC "convert" emissions (i.e., emissions resulting from land that has been converted from
one type to another) are reported within 20 years following conversion in the "convert" category (biomass losses are
typically reported in the year of conversion, while net changes in soil organic carbon during the entire conversion
period). FAOSTAT, DGVMs, and bookkeeping models usually only include "convert" fluxes from the year following
conversion, although bookkeeping models and DGVMs which deal with gross transitions may be able to include this
transition period more easily.

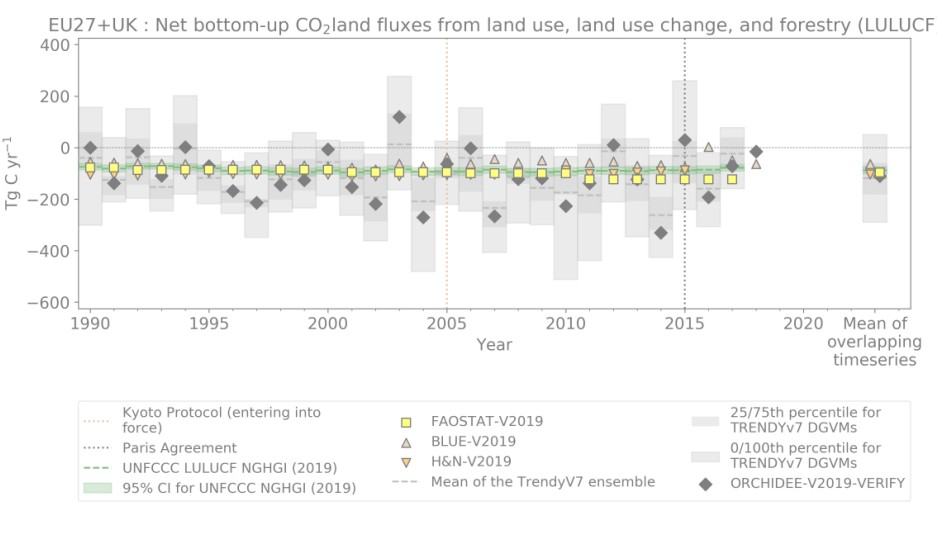

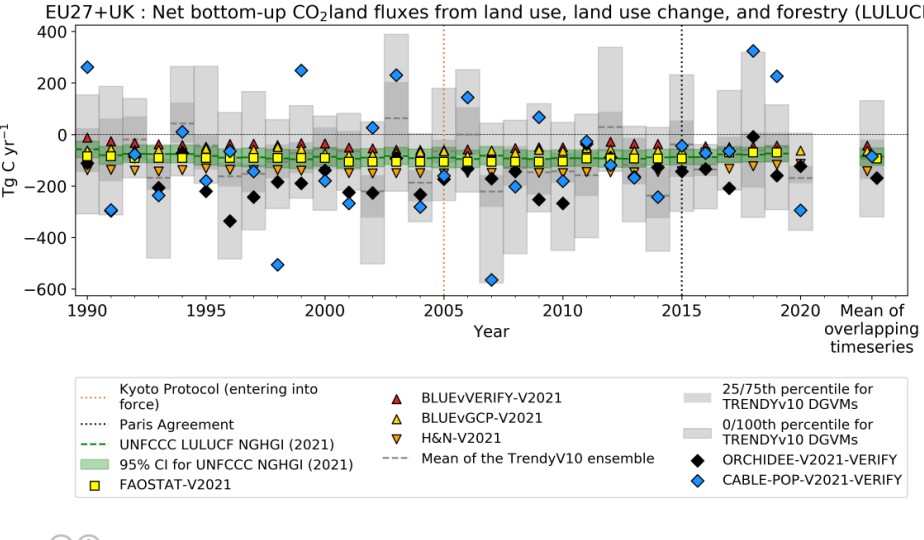


*Figure 8: Net CO$_2$ fluxes from total LULUCF activities in the EU27 + UK from previous data from Petrescu et al.*
*(2021b) (top plot) and data from seven new or updated sources (bottom plot) including: UNFCCC NGHGI (2021),*
*BLUE (vVERIFY), BLUE (vGCP2021), H&N (GCP2021), DGVMs (TRENDY v10), FAOSTAT (2021), ORCHIDEE*
*and CABLE-POP with high-spatial-resolution (0.125°) meteorological forcing (both models are also part of the*
*TRENDY ensemble at 0.5°). The gray bars represent the individual model data for the DGVMs. The UNFCCC estimate*
*includes all classes (remain and convert), as well as HWP. The relative error of the UNFCCC values represent the*



*UNFCCC NGHGI (2021) Member States reported uncertainty computed with the error propagation method (95 %*
*confidence interval), gap-filled and provided for each year of the timeseries. Biomass burning emissions are included*
*in the C stock estimates. The FAOSTAT estimate includes both Forest land remaining forest land in addition to*
*incorporating afforestation and deforestation as conversion of Forest land to other land types. The means are*
*calculated for the 1990–2019 overlapping period. The fluxes follow the atmospheric convention, where negative*
*values represent a sink while positive values represent a source.*
Figure 8 shows $CO_2$ fluxes from the NGHGI LULUCF sector compared to all other comparable bottom-up
(BU) estimates in this work: high-resolution S3 simulations for both ORCHIDEE and CABLE-POP; the median of
15 S3 simulations from the TRENDYv10 DGVM ensemble; three bookkeeping models; and FAOSTAT. As
mentioned above, taking the difference of the TRENDY S2 and S3 simulations provides an estimate of the net flux
from land use change, but inconsistencies are introduced either way, and therefore further research is needed in order
to establish which approach (S3-S2, or simply S3) leads to the most consistent comparison. For the overlapping period
1990-2019, the means of two out of the three bookkeeping models (BLUE vGCP (-61 Tg C yr$^{-1}$) and BLUE vVERIFY
(-43 Tg C yr$^{-1}$, using the Hilda+ land use forcing)) along with the mean of FAOSTAT (without Romania) (-93 Tg C
yr$^{-1}$) fall within the 95 % confidence interval of the UNFCCC NGHGI estimate of -86 ± 33 Tg C yr$^{-1}$. Only H&N rests
apart with a stronger sink (-142 Tg C yr$^{-1}$).
Bookkeeping models like BLUE and H&N do not include indirect effects on biomass growth due to factors
such as $CO_2$ fertilization, nitrogen deposition, and climate change, while NGHGIs implicitly include these impacts on
managed land through updated statistics. Recent work by Grassi et al. (2022b) demonstrates that including the sink
associated with human-induced indirect effects (as estimated by the S2 simulations from the TRENDY DGVM
ensemble) into results by bookkeeping models can largely reconcile estimates of net global LULUCF fluxes between
the NGHGIs and bookkeeping models. At the level of the EU27+UK, the inclusion of this sink results in an
overcompensation; the BMs estimate a net sink of -56.5 Tg C yr$^{-1}$ compared to the NGHGI estimate of -87.9 Tg C yr$^{-1}$
$^{-1}$, while the BMs+DGVMs results in -112 Tg C yr$^{-1}$. However, all of these estimates fall inside the NGHGI uncertainty
range in Fig. 8. This suggests that indirect effects are small in the EU27+UK.
The UNFCCC LULUCF estimates contain $CO_2$ emissions from all six land use categories and HWP,
including remaining categories and conversion to and from a category to another. The DGVMs show high interannual
variability, as demonstrated clearly by the high-resolution CABLE-POP simulation in Fig. 8. The mean values for
DGVMs across the overlapping period, on the other hand, agree fairly well with the NGHGI: -170 Tg C yr$^{-1}$, -84 Tg
C yr$^{-1}$, and -81 (min -285, max 118) Tg C yr$^{-1}$ for ORCHIDEE, CABLE-POP, and TRENDY v10, respectively,
compared to the NGHGI mean of -86 ± 33 Tg C yr$^{-1}$. Note again that ORCHIDEE and CABLE-POP are also part of
the TRENDYv10 ensemble, but the simulations included in TRENDY used a coarser meteorological forcing than the
one used within the VERIFY project (around 0.125° resolution). CABLE-POP also used a higher resolution land use
land cover change (LULCC) dataset for the results submitted to VERIFY (0.25° as opposed to 1.0°). The increased
IAV from the high-resolution CABLE-POP compared to ORCHIDEE is suspected to have been introduced through
the construction of the LULCC dataset as described in Appendix A2. Gross fluxes are, by definition, larger than net



fluxes, and consequently a method which incorporates gross fluxes (like CABLE-POP) can be expected to undergo
larger changes than a method incorporating net fluxes (like ORCHIDEE).

The differences between bookkeeping models and UNFCCC and FAOSTAT are discussed in detail
elsewhere, and focus on the inclusion of unmanaged land in bookkeeping models but not FAOSTAT and UNFCCC
methodologies (Petrescu et al., 2020; Grassi et al., 2018a, 2021). ORCHIDEE, CABLE-POP and the TRENDY v10
ensemble means show much higher inter-annual variability due to the sensitivity of the model fluxes to highly variable
meteorological forcing at sub-daily time steps which allow for much more rapid responses to changing conditions, as
already discussed in the previous sections. The incorporation of variable climate data and the fact that DGVM models
simulate explicitly climate impacts on $CO_2$ fluxes, which inventories and bookkeeping models do not, explain these
differences. A comparison including sector-specific models (e.g., ECOSSE, EFISCEN-Space, EPIC-IIASA, CBM)
where multiple model results are harmonized and aggregated to produce a "total" LULUCF flux comparable to
DGVMs and bookkeeping models would be insightful; however, such a comparison requires extensive analysis which
is beyond the scope of the current work.

### 3.3.5. Comparison of atmospheric inversions with NGHGI CO₂ estimates


Figure 9 highlights the range of estimates from global and regional atmospheric inversions (GCP2021,
EUROCOM, CSR, LUMIA, and CIF-CHIMERE; see Table 2 and Appendix A2 for more details) against bottom-up
total annual EU27+UK $CO_2$ land emissions/removals from the UNFCCC NGHGI (2021). The top panel in the figure
shows the previous results from Petrescu et al. (2021b). In these inversions, all components of the carbon cycle that
contribute to the observed atmospheric $CO_2$ gradients between stations are implicitly included as the inversions
incorporate observed atmospheric concentrations of $CO_2$. This includes processes where carbon is uptaken by
vegetation in one area and emitted in a different area, i.e. emissions due to the respiration of laterally transported
carbon.

One significant change between this work and Petrescu et al. (2021b) is the removal of emissions and sinks
from inversion results due to lateral transport of carbon from crop trade, wood trade, and inland waters. Bottom-up
methods (including all the NGHGIs for European countries) do not consider emissions and removal of atmospheric
$CO_2$ due to lateral transport of carbon, while observations assimilated into top-down inversions record all $CO_2$ fluxes
without separating their components. We followed Eq. (1) of Deng et al. (2021) without prior masking for managed
land. Emissions from lateral transport of carbon ("lateral fluxes") were prepared generally following the approach
described by Ciais et al. (2021), where crop and wood product fluxes are derived from country-level trade statistics
compiled by the FAO. Inland water emissions and riverine export of terrestrial carbon use spatially explicit
climatological data and a statistical model combined with estimates of gas transfer velocities. A more complete
description is given in Appendix A2. This adjustment has been applied to all top-down fluxes reported here unless
indicated otherwise.

The C fluxes from inland waters (rivers and lakes) reported in Petrescu et al. (2021b), were replaced in this
study by maps of sinks/sources of rivers/lakes, wood and crops, accounting for a combined mean of -136 Tg C yr$^{-1}$
(over the 2010-2018 common period of the inversions). For comparing bottom-up methods (including the NGHGI) to



TD estimates in the EU27+UK, it is always necessary to remove the traded wood and crop harvest (see Deng et al.
(2021) for additional explanations). For the NGHGI, this arises due to how harvested wood products are considered.
HWPs can be reported to the UNFCCC by multiple approaches, three of which are outlined in Chapter 12 of Volume
4 of the 2006 IPCC Guidelines. One of these methods (the Atmospheric Flow Approach) would allow for a direct
comparison with the inversions as wood product emissions are accounted for in countries in which they are in use and
in landfills. However, all countries in the EU27 adopt the Production Approach (2022 NIR of the European Union
(EEA/PUBL/2022/023)) in which emissions are considered due to domestic harvest regardless of where the wood is
transformed or used. Inversions, on the other hand, see the HWPs where they transform into $CO_2$, either through
decomposition or incineration. It should be noted that DGVMs also typically implement the Production Approach on
a pixel level (i.e., harvested wood decomposes in the pixel where it is produced). As pixels reported for the high-
resolution simulations here are around 10 km wide, this implicitly assumes that HWP never travel more than 10 km
from the harvest site (this becomes 50 km in coaster resolution simulations like TRENDY). Therefore, removing
emissions from lateral carbon transport makes inversions more comparable not only to NFGHGIs but also to DGVMs.
Flux estimates from inversion methods for $CO_2$ land show much more variability than the NGHGI (Fig. 9).
The mean of the EUROCOM ensemble of European inversions shows good agreement with UNFCCC NGHGI data,
but with a huge spread of annual model results that extends from significant sources into large sinks. This large spread
can be linked to uncertainty in atmospheric transport modeling, inversion methods and assumptions, and to limitations
of the observation system. Furthermore, the EUROCOM inversions were designed for the European geographical
domain (which is larger than the EU27+UK) and are still being developed in particular to better constrain the
latitudinal and longitudinal boundary conditions.
The annual mean (overlapping period 2010-2018) of the EUROCOM v2021 inversions (-80 [-175,-4] Tg C
yr$^{-1}$) is the closest inversion estimate to the timeseries mean of the NGHGI estimates (-88 ± 31 Tg C yr$^{-1}$), where the
error bars for the inversion indicated the [0th,100th] percentiles due to the small size of the ensembles. The mean of
the global GCP2021 inversions (-50 [-320,+122] Tg C yr$^{-1}$) and regional inversions, CSR (-46 [-126,+47] Tg C yr$^{-1}$)
and LUMIA (-65 [-97,-27] Tg C yr$^{-1}$) show a lower absolute value, but report larger interannual variability (min/max).
The new CIF-CIMERE product has a mean of -99 Tg C yr$^{-1}$, showing more negative fluxes since 2010, which is not
seen in other models and is still under investigation.


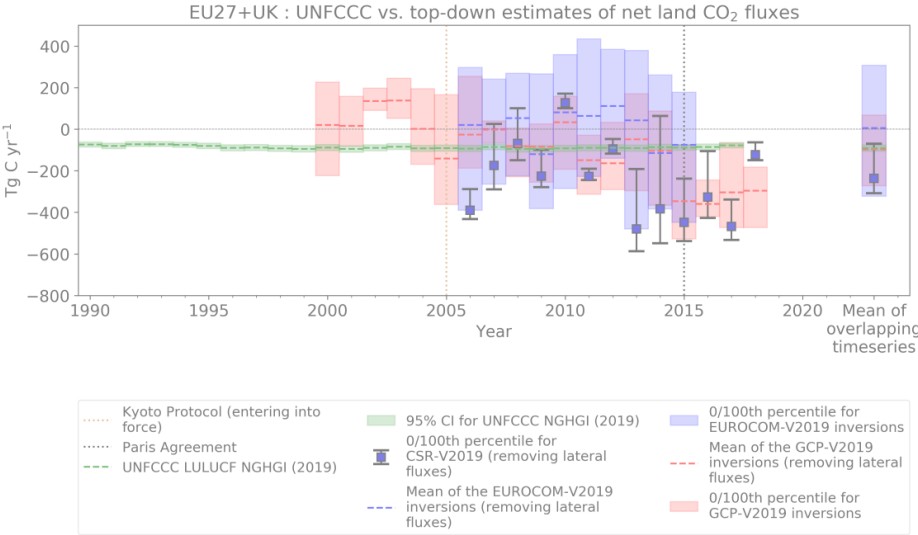

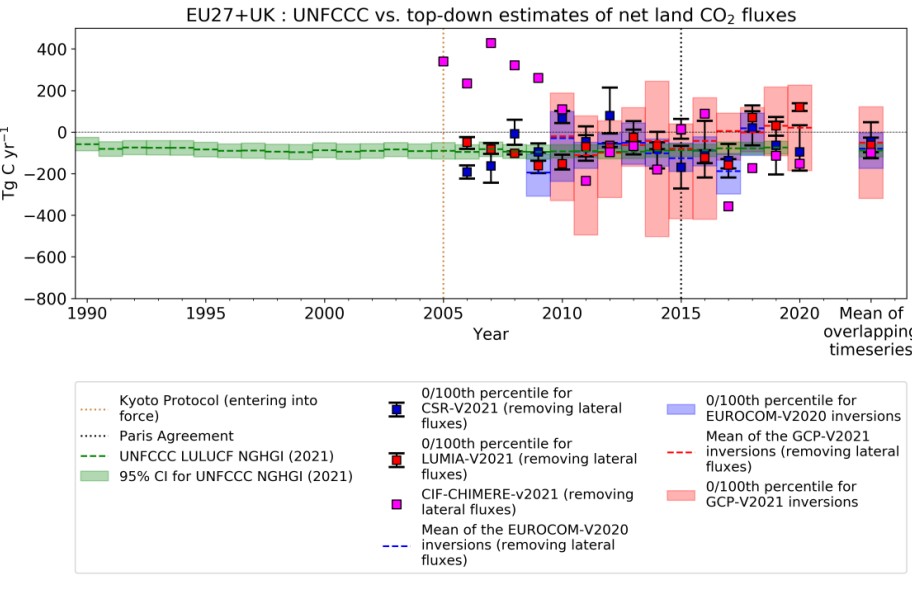


*Figure 9: Comparison of inventories and atmospheric inversions for the total EU27+UK biogenic CO$_2$ fluxes from*
*Petrescu et al. (2021b) (top plot) and updated data from current study (bottom plot). Top-down inversion results are:*
*the global GCB2021 ensemble, the regional EUROCOM ensemble, the regional CarboScopeReg model with multiple*
*variants, the regional LUMIA model with multiple variants, and CIF-CHIMERE. The relative error in the UNFCCC*





*values represents the UNFCCC NGHGI (2021) Member states reported uncertainty computed with the error*
*propagation method (95 % confidence interval) gap-filled and provided for every year of the timeseries. The timeseries*
*mean overlapping period is 2010-2018. The colored area represents the min/max of model ensemble estimates. The*
*same emissions due to lateral fluxes of carbon through rivers, crop trade, and wood trade are removed from the top-*
*down estimates in both the top and bottom graphs for consistency. The fluxes follow the atmospheric convention,*
*where negative values represent a sink while positive values represent a source. Note that Petrescu et al. (2021b)*
*presented the top plot including a suite of bottom-up models, which have been removed here for clarity as they have*
*already been presented in Fig. 8.*

The comparison of past and current versions of the inversions shows changes in specific models. A reduction
in the spread of the estimates is noted over the two past versions of CSR, resulting in a small source in the most recent
estimates. The CSRv2021 (bottom-plot) predicts in 2018 (last common year of both versions) a small source of 19 [-
64, +100] Tg C yr$^{-1}$ compared to the previous CSRv2019 which simulated a very strong sink of -253 [-280, -194] Tg
C yr$^{-1}$. This smaller source appears more in line with more positive fluxes expected in years of extreme drought (e.g.,
2018 in Northern Europe, although this did not impact the whole EU27+UK (Toreti et al., 2019)).

As can be seen in Fig. 9, there is also improved agreement between the EUROCOM ensemble and the
NGHGI, including a greatly reduced IAV compared to the previous version. The small EUROCOM ensemble mean
sink for the 2009-2015 period of -1.9 [-335,+322] Tg C yr$^{-1}$ (top panel) strengthened to -93 [-187,-15] Tg C yr$^{-1}$ in the
v2021 version (bottom panel). The UNFCCC total LULUCF mean is -92 ± 33 Tg C yr$^{-1}$ for the same time period. The
IAV of EUROCOM was dramatically reduced by removing the FLEXINVERT model from the v2021 ensemble as a
clear outlier of annual means due to a slightly shifted seasonal cycle (Appendix A2).

The new GCP2021 inversions show a clear trend towards decreasing the $CO_2$ sink strength of the land surface
after 2017, contrary to the NGHGI estimates which are relatively stable (Fig. 9, bottom). The large variability and
high sink observed in the upper plot of Fig. 9 shifted to a source in 2019 (21 [-185, +226] Tg C yr$^{-1}$) due to the extreme
climatic response of the TD models to the drought year, which can also be observed in the BU simulations (e.g.,
TRENDY v10, ORCHIDEE, and CABLE-POP in Fig. 8). Out of the GCP2021 models, CAMS was the model
responsible for the lower sinks (data not shown), which may be due partly to changes in the stations assimilated.

Table B2 summarizes the processes included in the $CO_2$ land models presented in this work, as these
processes are seen for the moment as the main cause of discrepancies between estimates shown in all the previous
figures. According to Table B2, no bottom-up model or dataset used here contains all of the 13 LULUCF categories
reported in the NGHGIs. A simple analysis of the mean 1990-2020 LULUCF fluxes from the EU27+UK NGHGI
(Table A3 in Appendix A2) shows that six categories account for almost 90 % of the gross flux: Forest land remaining
forest land (56 %), Land converted to cropland (7 %), Land converted to forest land (7 %), Grassland remaining
grassland (6 %), Harvested wood products (6 %), and Land converted to settlements (6 %). DGVMs currently include
more of these categories than other methods. As shown in Fig. 8, the mean 1990-2019 value of the mean of the 15
TRENDY DGVM simulations is -81.9 Tg C yr-1 (with a a range of [-285,118] Tg C yr$^{-1}$), while those of the
ORCHIDEE and CABLE-POP simulations using the high-resolution forcing provided in the VERIFY project are -



171 Tg C yr$^{-1}$ and -84.8 Tg C yr$^{-1}$, respectively. The means agree quite well for TRENDYv10 and CABLE-POP, but
the spread of all the DGVMs is quite large. In addition, the number of categories included may not be a good proxy
for quality of comparison. While an ideal model would include all categories in the NGHGI, it must also represent
these categories well. Figures 4-7 suggest that sector-specific models currently show better agreement with the
NGHGI than DGVMs, although a more detailed analysis including the entire suite of TRENDY models would be
insightful. Note that these categories are used as input to top-down approaches, and therefore cannot be disaggregated
into results after the simulation.

### 3.3.6. Uncertainties in top-down and bottom-up estimates

Uncertainties are essential for complete comparisons between models and approaches. This section
summarizes the main sources of uncertainty estimates interwoven throughout the above text. We also provide a
comparison of available uncertainties between the previous synthesis (V2019) and the current synthesis (V2021) for
both bottom-up and top-down methods. Finally, we give an overview of two important advances in uncertainty
estimation included in this work (one for the NGHGI, and one for top-down approaches), referring the interested
reader to the Appendix for more information.
Several sources of uncertainty arise from the synthesis of bottom-up (BU) inventories and models of carbon
fluxes, which can be summarized as: (a) differences due to input data and structural/parametric uncertainty of models
(Houghton et al., 2012) and (b) differences in definitions (Pongratz et al., 2014; Grassi et al., 2018b, 2021; Petrescu
et al., 2020, 2021b). Posterior uncertainties in top-down (TD) estimates mostly come from: 1) errors in the modeled
atmospheric transport; 2) aggregation errors, i.e., errors arising from the way the flux variables are discretized in space
and time and error correlations in time; 3) errors in the background mole fractions; and 4) incomplete information
from the observations and hence the dependence on the prior fluxes.
Figure 10 summarizes the quantifiable uncertainties in this work, compared to previous results from Petrescu
et al. (2021b). With the exception of the NGHGI, all the other uncertainties are calculated from ensembles of
simulations using either: 1) multiple models of the same general type, either using model-specific inputs or attempting
to harmonize inputs as much as possible (e.g., TRENDY), or 2) multiple simulations with the same model, varying
input parameters and/or forcing data (e.g., CarboScopeRegional, LUMIA). As a complete characterization of model
uncertainty involves exploring the full parameter, input data, and model structure space, none of the uncertainties
reported here can be considered "complete", but they represent best estimates given realistic constraints of resources
and knowledge. The uncertainties represent the mean of overlapping periods for the previous V2019 (overlapping
period: 2006-2015) versus the current V2021 (2010-2018). In general, the differences in mean behaviors between the
two versions falls within uncertainty estimates. Note, however, that this graph can hide certain behaviors. For
example, the similarity in the means for ORCHIDEE-VERIFY for both periods (-128.5 and -131.0 Tg C yr$^{-1}$ for V2019
and V2021, respectively) is likely a coincidence, given the wide fluctuation of annual values and the differences in
the multi-decennial means seen in Fig. 8.
Figure 10 shows notable reductions in the spread of two ensembles: EUROCOM and CSR. Both of these
are regional ensembles. In addition, the CSR results show a weaker sink in the current V2021 version compared to
the previous V2019 version. As noted in Appendix A2, the change for CSR is explained by the inclusion of a corrected



observation dataset for an isolated station in southeastern Europe which heavily influenced the regional results. The
reduction in the spread of the EUROCOM ensemble results from the exclusion of a single member which produces
annual flux results that are clear outliers compared to the remaining three members. More details of this analysis can
be found in Appendix A2. The remaining ensembles retain similar model spread compared to the previous versions.

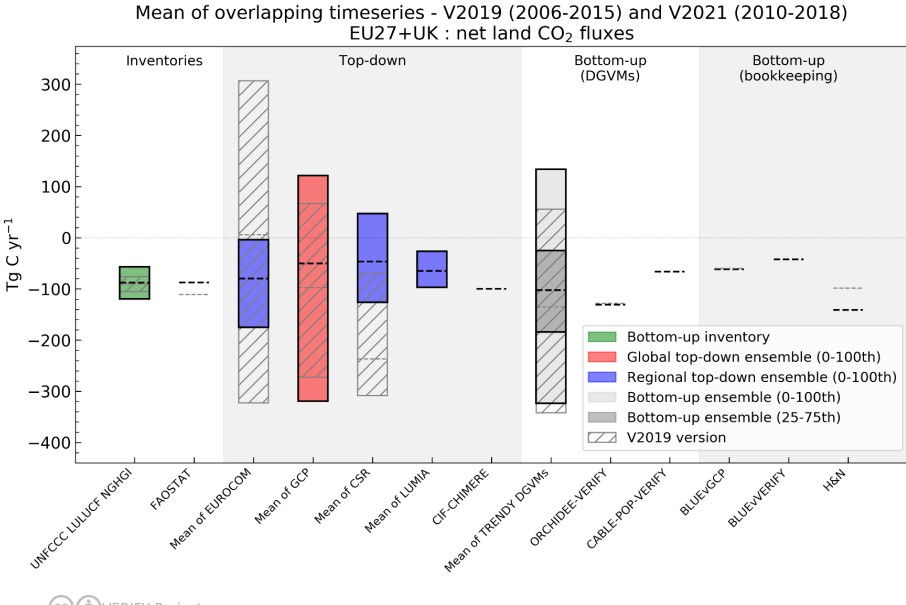


*Figure 10: Mean annual values of overlapping time periods (2006-2015) from Petrescu et al. (2021b) (transparent*
*boxes and light gray lines) and new means for the 2010-2018 period from the current study (Fig. 8 and 9, Sect. 3.3.4*
*and 3.3.5). The hashed boxes and colored boxes depict the "old" and "new" values for ensembles of multiple models,*
*with the top and bottom of the boxes corresponding to minimum and maximum mean values of the overlapping period.*
*For non-ensemble models (e.g., CIF-CHIMERE, FAOSTAT) the mean of the old and new overlapping periods are*
*given by gray dotted and black dashed lines, respectively. The NGHGI UNFCCC uncertainty is calculated for*
*submission year 2021 as the relative error of the NGHGI value, computed with the 95 % confidence interval method*
*gap-filled and provided for every year of the timeseries. Inversions for both V2019 and V2021 have been corrected*
*for emissions of $CO_2$ from lateral transport of carbon using identical datasets to enable a fair comparison. The fluxes*
*follow the atmospheric convention, where negative values represent a sink while positive values represent a source.*

Three advances in uncertainty estimation were made in this study, involving all three classes of models:

NGHGI, bottom-up, and top-down. In Petrescu et al. (2021b), percentage uncertainties for the NGHGI (2019)
LULUCF sector and land use categories were taken from reported uncertainties of the EU Member States and UK that
are used for compiling the National Inventory Reports (NIR) of the EU27+UK bloc, as well as the aggregate
uncertainties for the block reported in the EU NIR. Uncertainty estimates were only given for a single year and were

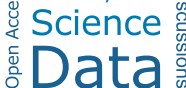

also partially incomplete due to missing uncertainty estimates for some sectors/subsectors of some countries. For the
current work, we use values compiled by the EU inventory team involving a recently developed procedure to
harmonize and gap-fill uncertainties reported by the Member States at the sector level (see EU NIR, 2021). Error
correlations are accounted for, in addition to year-to-year variations in sub-sectoral contributions to the overall
uncertainty. Extensive details are found in Appendix A1, and permit estimates of uncertainty on an annual basis, as
opposed to the single value used in the previous synthesis. Note, however, that this procedure was not applied to sub-
sectoral categories (FL, CL, and GL), for which values were taken directly from EU NIR (2021) and applied across
the whole timeseries. Synthesis plots created for individual countries and reported on the VERIFY website (VERIFY
Synthesis Plots, 2022) take percentages directly from the respective country's NIR.

The second advance relates to the impact of forcing data on bottom up models, in particular DGVMs. Figure
A3 (Appendix A) shows how the ORCHIDEE model responds to both changes in meteorological forcing (for
ORCHIDEE) and nitrogen forcing (for ORCHIDEE-N) over the past several decades. The impact of both is relatively
small compared to interannual variability. This is likely due to at least two reasons. The first reason is that
meteorological forcing used in this work has been re-aligned to the CRU observational dataset at 0.5 degrees and
monthly resolution, thus removing large-scale and long-term differences between the original meteorological datasets.
In addition, extensive spin-up and transient simulations are run for ORCHIDEE before reaching the point at which
the forcing changes (1981 for the meteorological forcing, and 1995 for the nitrogen forcing). Such lengthy simulations
enable woody biomass and soil carbon pools to develop a significant amount of inertia in response to additional
changes. Greater differences may be seen for models where modified forcing data covers the entire length of the pre-
production simulation steps.

The final advance relates to uncertainty characterization in the regional inversion model CSR following the
methodology of Chevallier et al. (2007). Spatially explicit estimates of the uncertainty reduction achieved from the
flux optimization were prepared through a Monte Carlo approach using an ensemble of 40 members. The uncertainty
reduction is then calculated based on the ratio of the prior errors and the posterior spread of the ensemble members,
using a formula such that 0 indicates no reduction and 1 indicates a complete elimination of uncertainty. A preliminary
analysis showed that a considerable reduction may be achieved through the inclusion of more observation stations,
although additional work is needed. For the moment, these maps only reflect random uncertainties, and systematic
uncertainties remain poorly characterized. More information can be found in Appendix A2.

Figure 11 presents an idea of the spatial uncertainties associated with these datasets. Total $CO_2$ land fluxes
from EU27+UK and five main regions in Europe are presented, divided into top-down (top panel) and bottom-up
(bottom panel) approaches for clarity. The regions (North, West, Central, East and South) consist of Annex I Parties
to UNFCCC both inside and outside of the EU27+UK bloc, and are listed in Table A1. Figure 11 shows the total $CO_2$
land fluxes from the NGHGIs for base year 1990, as well as five-year mean values for the 2011-2015 and 2015-2019
periods. The five-year periods are used as an exercise for what could be achieved in the first GST and also because
they provided the most overlap with the datasets reported here. As the BU models in VERIFY include and simulate
$CO_2$ fluxes for at most three out of the six classes reported to the UNFCCC (FL, CL and GL), for comparison and
consistency purposes both UNFCCC total LULUCF (including all six classes and HWP), as well as the UNFCCC



FL+CL+GL estimates are shown. Figure 11 presents $CO_2$ fluxes that include both direct and indirect LULUCF effects
on managed land. The total UNFCCC estimates include the total LULUCF emissions and sinks (by the UNFCCC
definition) belonging to all six IPCC land classes and the HWP class (see Sect. 2.3 and Appendix B for more details).
The NGHGI estimates are plotted and compared against fluxes simulated with statistical global and regional datasets:
bookkeeping models, biosphere and sector-specific models, and inversion model ensembles. The error bar represents
the variability in model estimates as the min and max values in the ensemble.




*Figure 11: Five-year means (2011–2015 and 2015-2019 as hashed and colored bars, respectively) of total CO₂ land*
*flux estimates (in Tg C) for EU27+UK and five European regions (North, West, Central, South and East) for top-*
*down (top) and bottom-up (bottom) methods compared to inventories. Eastern European region does not include*
*European Russia. Northern Europe includes Norway. Central Europe includes Switzerland. The UNFCCC*
*uncertainty for the Republic of Moldova was not available. The data comes from: UNFCCC NGHGI (2021) total*



*LULUCF submissions (dark green) which are plotted with respective base year 1990 (black star) estimates, the UNFCCC NGHGI (2021) FL+CL+GL estimates (light green), sector-specific BU models for FL, CL and GL (CBM, EPIC-IIASA, ECOSSE), ecosystem models (ORCHIDEE, TRENDY v10 DGVMs, CABLE-POP), global dataset FAOSTAT, bookkeeping models (BLUE (vGCP, and vVERIFY) and H&N), total $CO_2$ flux from TD inversion ensembles (GCP2021, EUROCOM) and three regional European inversions (CarboScopeReg (CSR), LUMIA and CIF-CHIMERE). ECOSSE_GL data was not updated beyond 2018. Lateral $CO_2$ fluxes (rivers/lakes, wood and crops sinks/sources) are represented separately (orange) and are removed from the top-down estimates as explained in the text. The fluxes follow the atmospheric convention, where negative values represent a sink while positive values represent a source.*

In general across the regions, BU (observation-based and process-based models) agree well with the UNFCCC-reported total LULUCF sources and sinks, except for the CABLE-POP DGVM which simulates a source for Central and Western Europe. As can be seen from the figure, however, this is not unexpected; the ensemble of TRENDY DGVMs shows a very large spread, and as such some DGVMs will undoubtedly display more extreme behavior. There remain however large disagreements between all estimates for Eastern Europe. This could be related to reduced data coverage for this region, in particular for the top-down approaches which depend on atmospheric measurement stations. In Northern Europe, some inversions agree with the NGHGIs on the magnitude of the sink (mean of 2015-2019 of -65 Tg C yr$^{-1}$), while in Central Europe there is a large variance between the models. The differences are explained by updates and methodological changes detailed in Sect. 3.3.2 (sector specific process-based models and NGHGI), 3.3.3 (DGVMs, bookkeeping models and NGHGI) and 3.3.4 (all BU, TD and NGHGI). Finally, the TD estimates are better in line with the NGHGI and the BU estimates after the removal of emissions due to lateral fluxes of carbon (discussed in Sect. 3.3.4). However, large variations still remain in the range of min/max of model ensembles represented in the figure by the error bars. For some models with high inter-annual variability (e.g., CIF-CHIMERE and CABLE-POP), the five-mean changes drastically between the two time periods but this may not represent a significant trend.

## 4. Data availability

Annual timeseries for the EU27+UK used in creation of the figures in this work for V2019 and V2021 are available for public download at https://doi.org/10.5281/zenodo.7365863 (McGrath et al., 2022). This excludes $CO_2$ fossil data for the IEA, which is subject to license restrictions. The data are reachable with one click (without the need for entering login and password), and downloadable with a second click, consistent with the two click access principle for data published in ESSD (Carlson and Oda, 2018). The data and the DOI number are subject to future updates and only refers to this version of the paper. In addition, figures and annual timeseries for EU27+UK as well as other countries and regions are available from VERIFY Synthesis Plots (2022).

## 5. Summary and concluding remarks

This work represents an update of the Petrescu et al. (2021b) European $CO_2$ synthesis paper presenting and investigating differences between the UNFCCC NGHGI, BU data-based inventories, both coarse and high resolution



process-based BU models, and TD approaches represented by both global and regional inversions. Datasets used in
the previous work have been updated by extending the temporal coverage and updating the models and data behind
the calculations. In addition, several new models to expand the number of independent approaches compared have
been added. Additional efforts have been made to improve uncertainty characterization in two approaches, along with
a first attempt to present as many datasets as possible in a clear single figure to draw overarching conclusions.

$CO_2$ fossil emissions dominate the anthropogenic $CO_2$ flux in the EU27+UK, regardless of the approach

employed and irrespective of uncertainties. Fossil $CO_2$ emissions are more straightforward to estimate than ecosystem
fluxes due to combustion being easier to model and parameterize at large scales. A suite of eight BU methods for
fossil $CO_2$ emissions are within the uncertainty of the NGHGI when methods are harmonized to include similar
categories. The remaining differences can often be attributed to definitions, assumptions about activity data or
emission factors, and the allocation of fuel types to different sectors (see Sect. 3.2 and Fig. B3). The one available TD
method, a regional European inversion system (CIF-CHIMERE) using an NOx proxy to determine $CO_2$ fossil
emissions, shows broad agreement with the BU estimates. However, this initial TD inversion is not yet capable of
distinguishing the minor differences between the various BU estimates and does not yet quantify uncertainties.
However, a substantial decrease in the level of uncertainty of the inverse modeling system is expected in the near-
term with the large-scale deployment of observation networks dedicated to detecting fossil fuel emissions (e.g., with
launch of the $CO_2M$[19] satellite mission in 2025). In the short-term, the CoCO2 project (CoCO2, 2022) aims to advance
methodology around co-assimilation of existing $CO_2$ satellite data (from the OCO-2/3 instruments) and to provide
new analysis of the CO/FFCO2 and NOx/FFCO2 ratios in order to significantly decrease uncertainty in the fossil $CO_2$
estimates.

The $CO_2$ land fluxes belong to the LULUCF sector, which is one of the most uncertain sectors in UNFCCC

reporting. The IPCC guidelines prescribe methodologies that are used to estimate the $CO_2$ fluxes in the NGHGI, but
grant countries significant freedom to adopt methods appropriate to their national circumstances. When analyzing the
different estimates from multiple BU sources (inventories and models) similar sources of uncertainties are observed
such as: (a) differences due to input data and structural/parametric uncertainty of models (Houghton et al., 2012;
Pongratz et al., 2021) and (b) differences in definitions (Pongratz et al., 2014; Grassi et al., 2018b; Petrescu et al.,
2020, 2021b; Grassi et al., 2021). Reducing uncertainties in LULUCF estimates is needed given the increasing
importance of the sector to EU climate policy over the next decades. In contrast to the previous 2020 climate and
energy package, the LULUCF sector will now formally contribute to the binding emission reduction targets of the
Unions 2030 climate and energy framework (EU, 2018a; 2018b). Furthermore, the European Climate Law explicitly
states that LULUCF, together with all sectors of the economy, should contribute to achieving Climate neutrality within
the Union by 2050 (EU, 2021b).

The LULUCF sector in NGHGIs is composed of six land use categories. Of these, Forest land provides the

most important contribution to the net $CO_2$ land flux in the EU27+UK, followed by Cropland and Grassland. HWP
and "Land converted to settlements" also have non-negligible contributions, and changes in HWP strongly influence

---

[19] CO2M: Copernicus Anthropogenic Carbon Dioxide Monitoring,
https://esamultimedia.esa.int/docs/EarthObservation/CO2M_MRD_v3.0_20201001_Issued.pdf



variations in decennial mean net LULUCF fluxes for the region.  Of these, all except "Land converted to settlements"
are represented in general ecosystem models, while Forestland, Cropland, and Grassland are simulated by sector-
specific process-based and data-driven models.  Top-down inversions are capable of simulating net $CO_2$ fluxes to the
atmosphere, but cannot yet attribute them between different categories.
Differences in the detailed sector-specific and inversion model results (Fig. 4-9) often come from choices in
the simulation setup and the type of model used: bookkeeping models, process-based DGVMs, inventory-based
statistical methods, or atmospheric inversions. Results also differ based on whether fluxes are attributed to LULUCF
emissions due to the cause or location of occurrence. For example, indirect fluxes on managed land are included in
NGHGI and FAOSTAT, while additional sink capacity (e.g., Petrescu et al., 2021b) is included in estimates from
process-based models (e.g., ORCHIDEE or TRENDY DGVMs). The use of gross land use changes fluxes (e.g., in
the NGHGI, bookkeeping models, and CABLE-POP) as opposed to net fluxes also likely plays an important role.  We
found that adjusting top-down models by emissions/removals resulting from later transport of carbon through trade
and the inland water network improves the agreement with the NGHGI of the EU27+UK (Fig. 9, compared to Petrescu
et al., 2021b).
Observation-based BU estimates of LULUCF provide large year-to-year flux variability (Fig. 4-7, in
particular for DGVMs like ORCHIDEE, CABLE-POP and the TRENDY ensemble), contrary to the NGHGI,
primarily due to the effect of varying meteorology. In particular, the duration and intensity of the summer growing
season can vary significantly between years (e.g., Bastos et al., 2020a; Thompson et al., 2020). In the framework of
periodic NGHGI assessments, the choice of a reference period (such as 2015-2019, as used here) or the use of a
moving window to calculate the means may be critical to smooth out high inter-annual variability and facilitate
comparisons. One can also imagine incorporating IAV into the NGHGIs through the use of annual anomalies of
emission factors calculated from Tier 3 observation-based approaches (either BU or TD).  TD estimates also show
very large inter-annual variability (Fig. 9). Uncertainties in the inversion results are primarily due to uncertainties in
atmospheric transport modeling, boundary conditions, technical simplifications and uncertainty inherent to the
limitation of the observation network. Currently, regional inversions (LUMIA, CSR and EUROCOM) are still under
development and face different challenges from the coarser resolution global systems used here to represent regional
results (GCP).  Based on this work, it is difficult to claim that one or the other provides a more accurate result for the
net $CO_2$ land fluxes across the EU27+UK, although two regional inversion ensembles (EUROCOM and CSR)
dramatically reduced their uncertainties between the previous and current versions of this synthesis, with CSR showing
much more overlap now with the NGHGI (Fig. 10).
Uncertainties can be reflected in space as well as in time. Fig. 11 separates mean BU and TD values for all
methods into five different regions in Europe.  From this figure, it's clear that some regions suffer from higher
uncertainties than others.  Part of this is likely linked to the spareness of atmospheric observation data for the TD
estimates (e.g., Eastern Europe).  Reconciling differences across aggregated EU regions may be challenging due to
diverse methodologies and drivers in each country. On the other hand, the analysis of smaller regions or individual
countries may represent a productive first step towards monitoring the current state of emissions as national data and
experts can be used to help clarify differences across models. Country-level case studies may help inform the design



of future monitoring and verification systems (MVS) for $CO_2$ which aim to supply additional evidence for the
emissions levels and trends, coupling anthropogenic activities and associated emissions with the atmospheric patterns
of greenhouse gas concentrations, and perform data assimilation and modeling over a wide variety of environmental
conditions (Pinty et al., 2017).
As seen in figures throughout this work, reducing uncertainties of both individual models and classes of
models remains a priority. Some categories (Forestland, Cropland) produce results for multiple category-specific
models which lie within the uncertainty of the NGHGI. This likely reflects relatively the use of data-driven models
and the relatively high quality of data that is available due to the economic importance of these categories. On the
other hand, generalized ecosystem models (the DGVMs, like ORCHIDEE and CABLE-POP) may create mean
estimates which fall within uncertainties, but fall outside of NGHGI uncertainties for any given year due to the
sensitivity of processes in these models to rapidly changing meteorology and the necessity for these models to operate
globally, including in data-poor regions for which parameterization may be impossible. Two advances in
characterizing uncertainty were presented here: one for the case of the NGHGI, and one for the case of the TD model
CSR. Additional characterization of uncertainty both within and across models will enable more fair comparisons
between methods.
A more detailed analysis of LULUCF fluxes at the regional/country level is foreseen as part of projects linked
to VERIFY including the RECCAP2 initiative (RECCAP2, 2022) and current and future Horizon Europe funded
projects (e.g., CoCO2, EYE-CLIMA, AVENGERS, PARIS) which will highlight examples of good practice in
LULUCF flux monitoring amongst European countries. Sect. 3.3.6 presents a summary of uncertainties to provide
insight into ground observation systems assimilated by inversions. This lays the basis of future improvements for
establishing best practices on how to configure atmospheric inversions and systematically quantify uncertainties. For
the overall estimation of emissions from LULUCF activities on all land types (Fig. 8), the comparison is made more
challenging as results from both land use and land use changes are presented. Comparing only the "effect of land use
change" (conversion) is non-trivial. A methodology for reconciling LULUCF country estimates from the FAOSTAT
datasets with the NGHGIs is presented in Grassi et al. (2022a) and Grassi et al. (in prep) for the global scale.
The next steps needed to improve and facilitate the reconciliation between BU and TD estimates are the same
as those discussed in Petrescu et al. (2021b): 1) BU process-based models incorporating unified protocols and
guidelines for uniform definitions should be able to disaggregate their estimates to facilitate comparison to NGHGI
and 2006 IPCC practices (e.g., managed vs. unmanaged land, 20-year legacy for classes remaining in the same class,
distinction of fluxes arising solely from land use change, Grassi et al. (2022a)); 2) for sector-specific models, in
particular for cropland and grassland, improving treatment of the contribution of soil organic carbon dynamics to
the budget; 3) for TD estimates, using the recently developed Community Inversion Framework (Berchet et al., 2021)
to better assess the different sources of uncertainties from the inversion set-ups (model transport, prior fluxes,
observation networks), 4) standardize methods to compare datasets with and without interannual variability, and 5)
develop a clear way to report key system boundary, data, or definitional issues, as it often necessary to have deep
understanding of each estimate to know how to do a like-for-like comparison.



Similar to Petrescu et al. (2021b), this updated study concludes that a complete, ready-for-purpose monitoring
system providing annual carbon fluxes across Europe is still under development, but data sources are beginning to
show improved agreement compared to previous estimates. Therefore, significant effort must still be undertaken to
reduce the uncertainty across all potential methods (i.e., structural uncertainty in the models as well as the input data
supplied to the models or inventory approaches) used in such a system (e.g. Maenhout et al., 2020). Future activities
in the CoCO$_2$ project (CoCO2, 2022) will investigate the one and five-year carbon budgets across the data-rich area
of the EU27+UK and deepen the analysis for both global and regional/local (city level) estimates.

Achieving the well-below 2$^\circ$C temperature goal of the Paris Agreement requires, among other things, low-
carbon energy technologies, forest-based mitigation approaches, and engineered carbon dioxide removal (Grassi et
al., 2018a; Nabuurs et al. 2017). Currently, the EU27+UK reports a sink for LULUCF and forest management will
continue to be the main driver affecting the productivity of European forests for the next decades (Koehl et al., 2010),
shown as well by the domination of Forestland CO$_2$ fluxes to the LULUCF sector in the NGHGI for the bloc. Forest
management changes forest composition and structure, which affects the exchange of energy with the atmosphere
(Naudts et al., 2016), and therefore the potential of mitigating climate change (Luyssaert et al., 2018; Grassi et al.,
2019). Meteorological extremes can also affect the efficiency of the sink (Thompson et al., 2020). The EU forest sink
is projected to decrease in the near future (Vizzarri et al., 2021). Consequently, for the EU to meet its ambitious climate
targets, it is necessary to maintain and even strengthen the LULUCF sink (EU, 2020). Understanding the evolution of
the CO$_2$ land fluxes is critical to enable the EU27+UK to meet its ambitious climate goals.


**6. Appendices**

*Appendix A: Data sources, methodology and uncertainty descriptions*
Plots for all countries in Europe as well as dozens of country groups and some countries outside of Europe are available
following a simple registration (VERIFY Synthesis Plots, 2022).

**VERIFY project**
VERIFY's primary aim is to develop scientifically robust methods to assess the accuracy and potential biases
in national inventories reported by the parties through an independent pre-operational framework. The main concept
is to provide observation-based estimates of anthropogenic and natural GHG emissions and sinks as well as associated
uncertainties. The proposed approach is based on the integration of atmospheric measurements, improved emission
inventories, ecosystem data, and satellite observations, and on an understanding of processes controlling GHG fluxes
(ecosystem models, GHG emission models).
Two complementary approaches relying on observational data-streams were combined in VERIFY to
quantify GHG fluxes:
1) atmospheric GHG concentrations from satellites and ground-based networks (top-down atmospheric inversion
models) and



2) bottom-up activity data (e.g., fuel use and emission factors) and ecosystem measurements (bottom-up models).
For $CO_2$, a specific effort was made to separate fossil fuel emissions from ecosystem fluxes.

The objectives of VERIFY were:
**Objective 1**. Integrate the efforts between the research community, national inventory compilers, operational centers
in Europe, and international organizations towards the definition of future international standards for the verification
of GHG emissions and sinks based on independent observation.
**Objective 2**. Enhance the current observation and modeling ability to accurately and transparently quantify the sinks
and sources of GHGs in the land-use sector for the tracking of land-based mitigation activities.
**Objective 3.** Develop new research approaches to monitor anthropogenic GHG emissions in support of the EU
commitment to reduce its GHG emissions by 40 % by 2030 compared to the year 1990.
**Objective 4.** Produce periodic scientific syntheses of observation-based GHG balance of EU countries and practical
policy-oriented assessments of GHG emission trends, and apply these methodologies to other countries.

For more information on the project team and products/results please visit the VERIFY website (VERIFY, 2022).

Table A1: *Country grouping used for comparison purposes between BU and TD emissions as reported for the*
*country- and regional-level synthesis plots available through the VERIFY web portal.*

| Country name – geographical Europe | BU-ISO3 | Aggregation from TD-ISO3 |
|---|---|---|
| Luxembourg | LUX | |
| Belgium | BEL | BENELUX |
| Netherlands | NLD | BNL |
| Bulgaria | BGR | BGR |
| Switzerland | CHE | |
| *Lichtenstein* | *LIE* | *CHL* |
| Czech Republic | CZE | Former Czechoslovakia |
| Slovakia | SVK | CSK |
| Austria | AUT | AUT |
| Slovenia | SVN | North Adriatic countries |
| Croatia | HRV | NAC |
| Romania | ROU | ROU |
| Hungary | HUN | HUN |
| Estonia | EST | |

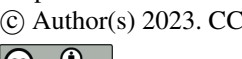



| Lithuania | LTU | Baltic countries |
|---|---|---|
| Latvia | LVA | BLT |
| Norway | NOR | NOR |
| Denmark | DNK | |
| Sweden | SWE | |
| Finland | FIN | DSF |
| Iceland | ISL | ISL |
| Malta | MLT | MLT |
| Cyprus | CYP | CYP |
| France (Corsica incl.) | FRA | FRA |
| *Monaco* | *MCO* | |
| *Andorra* | *AND* | |
| Italy (Sardinia, Vatican incl.) | ITA | ITA |
| *San Marino* | *SMR* | |
| United Kingdom (Great Britain + N Ireland) | GBR | UK |
| *Isle of Man* | *IMN* | |
| Iceland | | |
| Ireland | IRL | IRL |
| Germany | DEU | DEU |
| Spain | ESP | IBERIA |
| Portugal | PRT | IBE |
| Greece | GRC | GRC |
| *Russia (European part)* | *RUS European* | |
| *Georgia* | *GEO* | *RUS European+GEO* |
| *Russian Federation* | *RUS* | *RUS* |
| Poland | POL | POL |
| *Turkey* | *TUR* | *TUR* |
| EU27+UK (Austria, Belgium, Bulgaria, Cyprus, Czech Republic, Germany, Denmark, Spain, Estonia, Finland, France, Greece, Croatia, Hungary, Ireland, Italy, Lithuania, Latvia, Luxembourg, | AUT, BEL, BGR, CYP, CZE, DEU, DNK, ESP, EST, FIN, FRA, GRC, HRV, HUN, IRL. | E28 |





| | | |
|---|---|---|
| Malta, Netherlands, Poland, Portugal, Romania, Slovakia, Slovenia, Sweden, United Kingdom) | ITA, LTU, LVA, LUX, MLT, NDL, POL, PRT, ROU, SVN, SVK, SWE, GBR | |
| Western Europe (Belgium, France, United Kingdom, Ireland, Luxembourg, Netherlands) | BEL, FRA, UK, IRL, LUX, NDL | WEE |
| Central Europe (Austria, Switzerland, Czech Republic, Germany, Hungary, Poland, Slovakia) | AUT, CHE, CZE, DEU, HUN, POL, SVK | CEE |
| Northern Europe (Denmark, Estonia, Finland, Lithuania, Latvia, Norway, Sweden) | DNK, EST, FIN, LTU, LVA, NOR, SWE | NOE |
| *South-Western Europe (Spain, Italy, Malta, Portugal)* | *ESP, ITA, MLT, PRT* | *SWN* |
| *South-Eastern Europe (all) (Albania, Bulgaria, Bosnia and Herzegovina, Cyprus, Georgia, Greece, Croatia, Macedonia, the former Yugoslav, Montenegro, Romania, Serbia, Slovenia, Turkey)* | *ALB, BGR, BIH, CYP, GEO, GRC, HRV, MKD, MNE, ROU, SRB, SVN, TUR* | *SEE* |
| *South-Eastern Europe (Albania, Bosnia and Herzegovina, Macedonia, the former Yugoslav, Georgia, Turkey, Montenegro, Serbia)* | *ALB, BIH, MKD, MNE, SRB, GEO, TUR* | *SEA* |
| *South-Eastern Europe (EU) (Bulgaria, Cyprus, Greece, Croatia, Romania, Slovenia)* | *BGR, CYP, GRC, HRV, ROU, SVN* | *SEZ* |
| *Southern Europe (all) (SOE) (Albania, Bulgaria, Bosnia and Herzegovina, Cyprus, Georgia, Greece, Croatia, Macedonia, the former Yugoslav, Montenegro, Romania, Serbia, Slovenia, Turkey, Italy, Malta, Portugal, Spain)* | *ALB, BGR, BIH, CYP, GEO, GRC, HRV, MKD, MNE, ROU, SRB, SVN, TUR, ITA, MLT, PRT, ESP* | *SOE* |
| *Southern Europe (SOY) Albania, Bosnia and Herzegovina, Georgia, Macedonia, the former Yugoslav, Montenegro, Serbia, Turkey)* | *ALB, BIH, GEO, MKD, MNE, SRB, TUR,* | *SOY* |
| Southern Europe (EU) (SOZ) (Bulgaria, Cyprus, Greece, Croatia, Romania, Slovenia, Italy, Malta, Portugal, Spain) | BGR, CYP, GRC, HRV, ROU, SVN, ITA, MLT, PRT, ESP | SOZ |
| Eastern Europe (Belarus, Moldova, Republic of, *Russian Federation*, Ukraine) | BLR, MDA, *RUS,* UKR | EAE |
| *EU-15 (Austria, Belgium, Germany, Denmark, Spain, Finland, France, United Kingdom, Greece, Ireland, Italy, Luxembourg, Netherlands, Portugal, Sweden)* | *AUT, BEL, DEU, DNK, ESP, FIN, FRA, GBR, GRC, IRL, ITA, LUX, NDL, PRT, SWE* | *E15* |
| *EU-27 (Austria, Belgium, Bulgaria, Cyprus, Czech Republic, Germany, Denmark, Spain, Estonia, Finland, France, Greece, Croatia, Hungary, Ireland, Italy, Lithuania, Latvia, Luxembourg,* | *AUT, BEL, BGR, CYP, CZE, DEU, DNK, ESP, EST, FIN, FRA, GRC, HRV,* | *E27* |



| *Malta, Netherlands, Poland, Portugal, Romania, Slovakia, Slovenia, Sweden)* | *HUN, IRL. ITA, LTU, LVA, LUX, MLT, NDL, POL, PRT, ROU, SVN, SVK, SWE* | |
|---|---|---|
| *All Europe (Aaland Islands, Albania, Andorra, Austria, Belgium, Bulgaria, Bosnia and Herzegovina, Belarus, Switzerland, Cyprus, Czech Republic, Germany, Denmark, Spain, Estonia, Finland, France, Faroe Islands, United Kingdom, Guernsey, Greece, Croatia, Hungary, Isle of Man, Ireland, Iceland, Italy, Jersey, Liechtenstein, Lithuania, Luxembourg, Latvia, Moldova, Republic of, Macedonia, the former Yugoslav, Malta, Montenegro, Netherlands, Norway, Poland, Portugal, Romania, Russian Federation, Svalbard and Jan Mayen, San Marino, Serbia, Slovakia, Slovenia, Sweden, Turkey, Ukraine)* | *ALA, ALB, AND, AUT, BEL, BGR, BIH, BLR, CHE, CYP, CZE, DEU, DNK, ESP, EST, FIN, FRA, FRO, GBR, GGY, GRC, HRV, HUN, IMN, IRL, ISL, ITA, JEY, LIE, LTU, LUX, LVA, MDA, MKD, MLT, MNE, NDL, NOR, POL, PRT, ROU, RUS, SJM, SMR, SRB, SVK, SVN, SWE, TUR, UKR* | *EUR* |

*countries highlighted in *italic* are not discussed in the current 2021 synthesis mostly because unavailability of UNFCCC NGHGI reports (non-
Annex I countries[20]) but are present on the web-portal (VERIFY Synthesis Plots, 2022). Results for Annex I countries (NOR, CHE, ISL) and
Eastern European countries (EAE) are represented in Fig. 11.

---

[20]Non-Annex I countries are mostly developing countries. The reporting to UNFCCC is implemented through national communications (NCs) and biennial update reports (BURs): https://unfccc.int/national-reports-from-non-annex-i-parties





*Table A2: Methodological changes (**in bold**) of the current study with respect to Petrescu et al. (2020), Petrescu et al.*
*(2021b) and an internal VERIFY update (v2020); n/a cells mean that there is no data available.*

| Publication year | Bottom-up anthropogenic CO₂ estimates (fossil CO₂) | | | Top-down fossil CO₂ estimates | Bottom-up natural CO₂ (NBP) emissions/removals (land CO₂) | | | Top-down land CO₂ emissions | | Uncertainty and other changes |
|---|---|---|---|---|---|---|---|---|---|---|
| | Inventories | Global databases | Emission models | | Inventories | Emission models | Global Databases | Regional models | Global models | |
| Petrescu et al. (2020) AFOLU bottom-up synthesis | n/a | n/a | n/a | n/a | National emissions from UNFCCC (2018) 1990-2016 *LULUCF Forest land, -* EU28 data for five years (1995, 2000, 2005, 2010 and 2015) *Cropland and Grassland* (1990, 2005, 2010 and 2016) *All land uses* EU28 timeseries 1990-2016 | CBM Forest land (2000, 2005, 2010 and 2015) EFISCEN Forest land (1995, 2000, 2005, 2010 and 2015) BLUE All land uses 1990-2017 H&N All land uses 1990-2015 DGVMs (TRENDY v6) All land uses 1990-2017 | FAOSTAT Timeseries Remaining and conversions 1990-2016 | n/a | n/a | UNFCCC (2018) uncertainty estimates for 2016 (error propagation 95 % interval method) |
| Petrescu et al., 2021b | National emissions from UNFCCC | EDGAR v5.0 BP | n/a | IAP RAS fast-track inversion | National emissions from UNFCCC (2019) | CBM Forest land Timeseries | FAOSTAT Timeseries | CSR 2006-2018 | GCP 2019 inversions 2000-2018 | UNFCCC (2019) uncertainty estimates for 2016 (error propagation |



| | | | | | | | | | |
|---|---|---|---|---|---|---|---|---|---|
| (2019) CRFs 2014 All anthropogenic (excl. LULUCF) sectors, timeseries 1990-2015 | EIA CDIAC IEA GCP CEDS 2014 estimates split by fuel type EDGAR v5.0 All anthropogenic sectors, timeseries 1990-2015 | | 2014 (EU11+CHE) | 1990-2017 EU27 + UK timeseries of Forest Land, Cropland and Grassland Regional EU27 + UK totals (incl. NOR, CHE, UKR, MLD and BLR) | 1990-2015 EFISCEN Forest land timeseries 2005-2018 $CO_2$ emissions from inland waters ORCHIDEE Forest, cropland and grassland and all land uses 1990-2018 ECOSSE Cropland and grassland 1990-2018 EPIC-IIASA Cropland 1990-2018 BLUE All land uses 1990-2018 H&N All land uses 1990-2015 DGVMs (TRENDY v7) All land uses | Remaining and conversions 1990-2017 | EUROCOM 2006-2015 | | 95 % interval method) For model ensembles reported as variability in extremes (min/max) |





| | | | | | 1990-2018 | | | | |
|---|---|---|---|---|---|---|---|---|---|





| This study | National emissions from UNFCCC (**2021**) CRFs 2017 <br><br> All anthropogenic (excl. LULUCF) sectors, timeseries 1990-**2019** | EDGAR **v6.0** <br> BP <br> EIA <br> CDIAC <br> IEA <br> GCP <br> CEDS <br> **PRIMAP-hist 2.3** UNFCCC NGHGI **2021** <br><br> **2017 estimates split by fuel type** <br><br> EDGAR v6.0 <br> All anthropogenic sectors, timeseries 1990-2018 | n/a | **CIF-CHIMERE fast-track inversion** 2005-**2020** (EU27+UK) | National emissions from UNFCCC (**2021**) 1990-**2019** EU27 + UK of Forest land | CBM <br> Historical flux timeseries from Forest land remaining forest land 2000-2015 and new **2017-2020 estimate** <br><br> **EFISCEN-SPACE (updated model)** Forest land timeseries 2005-**2020** **For 15 EU countries** <br><br> ORCHIDEE <br> Forest, cropland and grassland and all land uses, **model updated** 1990-**2020** <br><br> **CABLE-POP 1990-2020** <br><br> **BLUE-VERIFY and BLUE-GCP)** All land uses 1990-**2019, 1990-2020** <br><br> **H&N GCP2021** All land uses 1990-**2020** <br><br> DGVMs **(TRENDY v10)** | FAOSTAT timeseries Remaining and conversions 1990-**2019** | CarboScope Reg 2006-**2020** <br><br> EUROCOM 2009-**2018** <br><br> LUMIA **2006-2020** <br><br> **CIF-CHIMERE** **2005-2020** | **GCP 2021** inversions 2010-2020 | UNFCCC (**2021**) uncertainty estimates for **2019** (error propagation 95 % interval method) <br><br> For model ensembles reported as the annual extremes (min/max) |
| --- | --- | --- | --- | --- | --- | --- | --- | --- | --- | --- |



| | | | | | | Forest, cropland and grassland and all land uses<br><br>1990-**2020**<br><br><br>**ECOSSE**<br><br>**Cropland**<br><br>1990-**2020**<br><br>Grassland<br><br>1990-2018<br><br><br>**EPIC-IIASA**<br><br>Cropland<br><br>1990-**2020**<br><br>**Grassland**<br><br>1990-**2020** | | | | |
|---|---|---|---|---|---|---|---|---|---|---|



**A1: Fossil CO₂ emissions**
***Bottom-up emission estimates***
For further details of all datasets, see Andrew (2020).

***UNFCCC NGHGI (2021)***

Annex I NGHGIs should follow principles of transparency, accuracy, consistency, completeness and
comparability (TACCC) under the guidance of the UNFCCC (UNFCCC, 2014) and as mentioned above, shall be
completed following the 2006 IPCC guidelines (IPCC, 2006). In addition, the IPCC 2019 Refinement (IPCC, 2019),
which may be used to complement the 2006 IPCC guidelines, has updated sectors with additional emission sources
and provides guidance on the use of atmospheric data for independent verification of GHG inventories.
Both approaches (BU and TD) provide useful insights on emissions from two different points of view. First,
as outlined in Volume 1, Chapter 6 of the 2019 IPCC Refinement (IPCC, 2019), TD approaches act as an additional
quality check for BU and NGHGI approaches, and facilitate a deeper understanding of the processes driving changes
in different elements of GHG budgets. Second, while independent BU methods do not follow prescribed standards
like the IPCC Guidelines, they do provide complementary information based on alternative input data at varying
temporal, spatial, and sectoral resolution. This complementary information helps build trust in country GHG estimates,



which form the basis of national climate mitigation policies. Additionally, BU estimates are needed as input for TD
estimates. As there is no formal guideline to estimate uncertainties in TD or BU approaches, uncertainties are usually
assessed from the spread of different estimates within the same approach, though some groups or institutions report
uncertainties for their individual estimates using a variety of methods, for instance, by performing Monte Carlo
sensitivity simulation by varying input data parameters. However, this can be logistically and computationally difficult
when dealing with complex process-based models.
Despite the important insights gained from complementary BU and TD emission estimates, it should be noted
that comparisons with the NGHGI are not always straightforward. BU estimates often share common methodology
and input data, and through harmonization, structural differences between BU estimates and NGHGIs can be
interpreted. However, the use of common input data restricts the independence between the datasets and, from a
verification perspective, may limit the conclusions drawn from the comparisons. On the other hand, TD estimates are
constrained by independent atmospheric observations and can serve as an additional, nearly independent quality check
for NGHGIs. Nonetheless, structural differences between NGHGIs (what sources and sinks are included, and where
and when emissions/removals occur) and the actual fluxes of GHGs to the atmosphere must be taken into account
during comparison of estimates. While NGHGIs go through a central QA/QC review process, the UNFCCC reporting
requirements do not mandate large-scale observation-derived verification. Nevertheless, the individual countries may
use atmospheric data and inverse modeling within their data quality control, quality assurance and verification
processes, with expanded and updated guidance provided in chapter 6 of the 2019 Refinement of IPCC 2006
Guidelines (IPCC, 2019). So far, only a few countries (e.g. Switzerland, UK, New Zealand and Australia) have used
atmospheric observations to constrain national emissions and documented these verification activities in their national
inventory reports (Bergamaschi et al., 2018), and none do so for $CO_2$.
Under the UNFCCC convention and its Kyoto Protocol, national greenhouse gas (GHG) inventories are the
most important source of information to track progress and assess climate protection measures by countries. In order
to build mutual trust in the reliability of GHG emission information provided, national GHG inventories are subject
to standardized reporting requirements, which have been continuously developed by the Conference of the Parties
(COP)[21]. The calculation methods for the estimation of greenhouse gasses in the respective sectors is determined by
the methods provided by the 2006 IPCC Guidelines for National Greenhouse Gas Inventories (IPCC, 2006). These
Guidelines provide detailed methodological descriptions to estimate emissions and removals, as well as
recommendations to collect the activity data needed. As a general overall requirement, the UNFCCC reporting
guidelines stipulate that reporting under the Convention and the Kyoto Protocol must follow the five key principles
of transparency, accuracy, completeness, consistency and comparability (TACCC).
The reporting under UNFCCC shall meet the TACCC principles. The three main GHGs are reported in
timeseries from 1990 up to two years before the due date of the reporting. The reporting is strictly source category
based and is done under the Common Reporting Format tables (CRF), downloadable from the UNFCCC official
submission portal: https://unfccc.int/ghg-inventories-annex-i-parties/2021.

---

[21] The last revision has been made by COP 19 in 2013 (UNFCCC, 2013)





The UNFCCC NGHGI $CO_2$ emissions/removals include estimates from five key sectors for the EU27+UK:
1 Energy, 2 Industrial processes and product use (IPPU), 3 Agriculture, 4 LULUCF and 5 Waste. The tiers method a
country applies depends on the national circumstances and the individual conditions of the land, which explains the
variability of uncertainties among the sector itself as well as among EU countries. This annual published dataset
includes all $CO_2$ emissions sources for those countries, and for most countries for the period 1990 to t-2. Some eastern
European countries' submissions began in the 1980s.

***NGHGI uncertainties***
The presented uncertainties in the reported emissions of the individual countries and the EU27+UK bloc
were calculated by using the methods and data used to compile the official GHG emission uncertainties that are
reported by the EU under the UNFCCC (NIRs, 2022). The EU uncertainty analysis reported in the bloc's National
Inventory Report (NIR) is based on country-level, Approach 1 uncertainty estimates (IPCC, 2006, Vol. 1, Chap. 3)
that are reported by EU Member States, Iceland and United Kingdom under Article 7(1)(p) of EU (2013). These
country-level uncertainty estimates are typically reported at beginning of a submission cycle and are not always
revised with updated CRF submissions later in the submission cycle. Furthermore, the compiled uncertainties of some
countries are incomplete (e.g., uncertainties not estimated for LULUCF and/or indirect $CO_2$ emissions, certain
subsector emissions are confidential) and the sector and gas resolution at which uncertainties are provided varies
between the countries. The EU inventory team therefore implements a procedure to harmonize and gap-fill these
uncertainty estimates. A processing routine reads the individual country uncertainty files that are pre-formatted
manually to assign consistent sector and gas labels to the respective estimates of emissions/removals and uncertainties.
The uncertainty values are then aggregated to a common sector resolution, at which the emissions and removals
reported in the uncertainty tables of the countries are then replaced with the respective values from the final CRF
tables of the countries. Due to the issue of incompleteness mentioned above, the country-level data are then screened
to identify residual GHG emissions and removals for which no uncertainty estimates have been provided. Where
sectors are partially complete, the residual net emission is quantified in $CO_2$ equivalents and incorporated. An
uncertainty is then estimated, by calculating the overall sector uncertainty of the sources and sinks that were included
in that country's reported uncertainties estimates and assigning this percentage average to the residual net emission.
In cases where for certain sectors no uncertainties have been provided at all (e.g., indirect $CO_2$ emissions, LULUCF),
an average (median) sector uncertainty in percent is calculated from all the countries for which complete sectoral
emissions and uncertainties were reported, and this average uncertainty is assigned to the country's sector GHG total
reported in its final CRF tables.
The country-level uncertainties presented in this paper, have been compiled using this same processing
routine and using the uncertainties and CRF data reported by the countries in the 2021 submission. However, here the
method has been expanded to gap-fill at the individual greenhouse gas level ($CO_2$ emissions and removals only) rather
than at the aggregate GHG level. Furthermore, the expanded method here assigns the sub-sectoral uncertainties to the
emissions and removals of the entire timeseries (1990-2019), rather than just the base year and latest year of the
respective timeseries. This allows uncertainties to be sensitive to the sub-sectoral contributions to sectoral and national



total emissions, which of course change over time. For each year of the timeseries, uncertainties in the total and
sectoral $CO_2$ emissions are calculated using Gaussian error propagation, by summing the respective sub-sectoral
uncertainties (expressed in kt $CO_2$) in quadrature and assuming no error correlation. In contrast, for the EU27+UK
bloc, uncertainties in the total and sectoral $CO_2$ emissions were calculated to take into account error correlations
between the respective country estimates at the subsector level. This was done by applying the same methods and
assumptions described in the 2022 EU NIR (UNFCCC NIR, 2022). The subsector resolution applied for gap-filling
allows the routine to access respective data on emission factors from CRF Table *Summary 3* and apply correlation
coefficients (r) when aggregating the uncertainties. For a given subsector, it is assumed that the errors of countries
using default factors are completely correlated (r = 1), while errors of countries using country-specific factors are
assumed uncorrelated (r = 0). For countries using a mix of default and country-specific factors at the given subsector
level, it is assumed that these errors are partially correlated (r = 0.5) with one another and with the errors of countries
using the default factors only.
Based on these correlation assumptions, the routine then aggregates $CO_2$ emissions/removals and
uncertainties for the specified subsector resolution at the EU27+UK level. Uncertainties at sector total level are then
aggregated from the subsector estimates assuming no correlation between subsectors. However, for countries reporting
very coarse resolution estimates (e.g., total sector $CO_2$ emissions/removals) or where the sector has been partially or
completely gap-filled, it is assumed that these uncertainties are partially correlated (r = 0.5) with one another and with
the other reported subsector level estimates. Level uncertainties on the total EU27+UK $CO_2$ emissions and removals
(with and without LULUCF) are then aggregated from the sector estimates assuming no error correlation between
sectors.
Note that the above procedure does not apply to LULUCF categories (FL, CL, and GL). Estimates for these
values were taken directly from the EU NIR (2021 without gap-filling or consideration of correlations. As the values
are given for only one single year, this value is applied uniformly across the whole timeseries.
***EDGAR v6.0***
The first edition of the Emissions Database for Global Atmospheric Research was published in 1995. The
dataset now includes almost all sources of fossil $CO_2$ emissions, is updated annually, and reports data for 1970 to year
n-1. Estimates for v6.0 are provided by sector. Emissions are estimated fully based on statistical data from 1970 till
2018 https://data.jrc.ec.europa.eu/dataset/97a67d67-c62e-4826-b873-9d972c4f670b.
**Uncertainties:** EDGAR uses emission factors (EFs) and activity data (AD) to estimate emissions. Both EFs and AD
are uncertain to some degree, and when combined, their uncertainties need to be combined too. To estimate EDGAR's
uncertainties (stemming from lack of knowledge of the true value of the EF and AD), the methodology devised by
IPCC (2006, Chapter 3) is adopted, that is the overall uncertainty is the square root of the sum of squares of the
uncertainty of the EF and AD (uncertainty of the product of two variables). A log-normal probability distribution
function is assumed in order to avoid negative values, and uncertainties are reported as the 95 % confidence interval
according to IPCC (2006, chapter 3, equation 3.7). For emission uncertainty in the range 50 % to 230 % a correction



factor is adopted as suggested by Frey et al. (2003) and IPCC (2006, chapter 3, equation 3.4). Uncertainties are
published in Solazzo et al. (2021).
***BP***
BP releases its Statistical Review of World Energy annually in June, the first report being published in 1952.
Primarily an energy dataset, BP also includes estimates of fossil-fuel $CO_2$ emissions derived from its energy data (BP
2011, 2017). The emissions estimates are totals for each country starting in 1965 to year n-1.
***CDIAC***
The original Carbon Dioxide Information Analysis Center included a fossil $CO_2$ emissions dataset that was
long known as CDIAC. This dataset is now produced at Appalachian State University, and has been renamed CDIAC-
FF (CDIAC, 2022). It includes emissions from fossil fuels and cement production from 1751 to year n-3. Fossil-fuel
emissions are derived from UN energy statistics, and cement emissions from USGS production data.
***EIA***
The US Energy Information Administration publishes international energy statistics and from these derives
estimates of energy combustion $CO_2$ emissions. Data are currently available for the period 1980-2016.
***IEA***
The International Energy Agency publishes international energy statistics and from these derives estimates
of energy combustion $CO_2$ emissions including from the use of coal in the iron and steel industry. Emissions estimates
start in 1960 for OECD members and 1971 for non-members, and run through n-1 for OECD members' totals, and
year n-2 for members' details and non-members. Estimates are available by sector for a fee.
***GCP***
The Global Carbon Project includes estimates of fossil $CO_2$ emissions in its annual Global Carbon Budget
publication. These include emissions from fossil fuels and cement production for the period 1750 to year n-1.
***CEDS***
The Community Emissions Data System has included estimates of fossil $CO_2$ emissions since 2018, with an
irregular update cycle (CEDS, 2022). Energy data are directly from IEA, but emissions are scaled to higher-priority
sources, including national inventories. Almost all emissions sources are included and estimates are published for the
period 1750 to year n-1. Estimates are provided by sector.
***PRIMAPv2.2***
The PRIMAP-hist dataset combines several published datasets to create a comprehensive set of greenhouse
gas emission pathways for every country and Kyoto gas, covering the years 1850 to 2018, and all UNFCCC (United
Nations Framework Convention on Climate Change) member states as well as most non-UNFCCC territories. The
data resolves the main IPCC (Intergovernmental Panel on Climate Change) 2006 categories. For $CO_2$, $CH_4$, and $N_2O$
subsector data for Energy, Industrial Processes and Product Use (IPPU), and Agriculture is available. Due to data
availability and methodological issues, version 2.2 of the PRIMAP-hist dataset does not include emissions from Land
Use, Land-Use Change, and Forestry (LULUCF). More info at https://zenodo.org/record/4479172#.YUsc6p0zbIU.
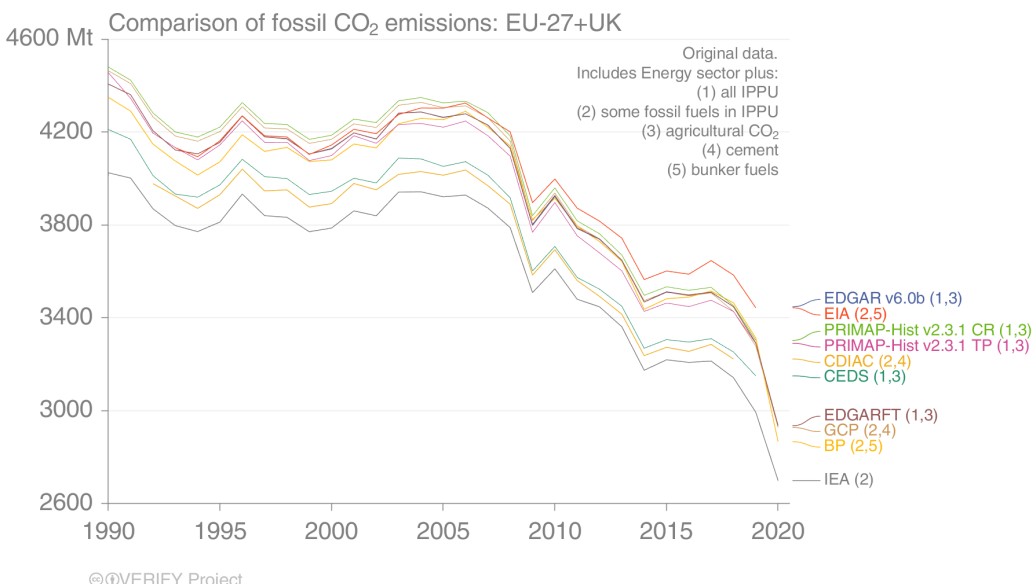


*Figure A1: Comparison of EU27+UK fossil CO₂ emissions from multiple inventory datasets; Identical to Fig. 2,*
*except that no system boundaries harmonization has been done. CDIAC does not report emissions prior to 1992 for*
*former-Soviet Union countries. CRF: UNFCCC NGHGI from the Common Reporting Format tables.*


***Top-down CO₂ emission estimates***
***CIF-CHIMERE - fossil CO₂ emission inversion***
CIF-CHIMERE is used for both CO₂ land and CO₂ fossil emission estimates, and this section only describes
the CO₂ fossil estimates. The product is explained in more detail by Fortems-Cheiney and Broquet, 2021.
Results from previous atmospheric inversions of the European fossil CO₂ emissions indicated that there were
much larger uncertainties associated with the assimilation of CO data than with that of NO₂ data for such a purpose
(Konovalov et al, 2016; Konovalov and Llova, 2018). In this context, we have developed an atmospheric inversion
configuration quantifying monthly to annual budgets of the national emissions of fossil CO₂ in Europe based on the
assimilation of the long-term series of NO₂ spaceborne observations; the Community Inversion Framework (CIF); the
CHIMERE regional chemistry transport model (CTM); corrections to the TNO-GHGco-v3 inventory of NOx
anthropogenic emissions at 0.5° horizontal resolution; and the conversion of NOx anthropogenic emission estimates
into CO₂ fossil emission estimates. For the first time, to our knowledge, variational regional inversions have been
performed to estimate the European CO₂ fossil emissions using NOx emissions from OMI satellite observations.
Particular attention is paid in the analysis assessing the consistency between the fossil CO₂ emissions estimates from
our processing chain with the fossil CO₂ emission budgets provided by the TNO-GHGco-v3 inventory based on the



emissions reported by countries to UNFCCC, which are assumed to be accurate in Europe. The algorithm first
optimizes NOx emissions and then assumes a fixed ratio of NOx to fossil $CO_2$ emissions. However, long-term plans
include the simultaneous inversion of all three gasses ($CO_2$, $NO_2$, and CO).
The analysis is conducted over the period 2005 to 2020. CHIMERE is run over a 0.5°×0.5° regular grid and
17 vertical layers, from the surface to 200hPa, with 8 layers within the first two kilometers. The domain includes 101
(longitude) x 85 (latitude) grid-cells (15.25°W-35.75°E; 31.75°N-74.25°N) and covers Europe. CHIMERE is driven
by the European Centre for Medium-Range Weather Forecasts (ECMWF) meteorological forecast (Owens and
Hewson, 2018). The chemical scheme used in CHIMERE is MELCHIOR-2, with more than 100 reactions (Lattuati,
1997; CHIMERE 2017), including 24 for inorganic chemistry. Climatological values from the LMDZ-INCA global
model (Szopa et al., 2008) are used to prescribe concentrations at the lateral and top boundaries and the initial
atmospheric composition in the domain. Considering the short $NO_2$ lifetime, we do not consider its import from outside
the domain: its boundary conditions are set to zero. Nevertheless, we take into account peroxyacetyl nitrate (PAN)
and the associated NOx reservoir for the large-scale transport of NOx.
Several critical aspects of this workflow need to be highlighted: (i) Fortems-Cheiney and Broquet (2021)
have not yet reported estimates of the uncertainty in the fossil $CO_2$ emissions (this requires the derivation of the
uncertainties in the NOx emission inversions and in the NOx-to-FFCO2 emission conversion), and (ii) the fossil $CO_2$
emission budgets provided by the TNO-GHGco-v3 inventory are based on the emissions reported by countries to
UNFCCC, which are assumed to be accurate in Europe, and therefore the NOx inversion prior estimate is consistent
with the inventory estimates (with respect to the NOx-to-FFCO2 emission conversion used to infer fossil $CO_2$
emissions from the NOx inversions).
**Uncertainty**: There is no uncertainty estimate currently available for this product.

**A2: Land $CO_2$ emissions/removals**
***Bottom-up $CO_2$ estimates***
***UNFCCC NGHGI 2021 - LULUCF***
Under the convention and its Kyoto Protocol, national greenhouse gas (GHG) inventories are the most
important source of information to track progress and assess climate protection measures by countries. In order to
build mutual trust in the reliability of GHG emission information provided, national GHG inventories are subject to
standardized reporting requirements, which have been continuously developed by the Conference of the Parties
(COP)[22]. The calculation methods for the estimation of greenhouse gasses in the respective sectors is determined by
the methods provided by the 2006 IPCC Guidelines for National Greenhouse Gas Inventories (IPCC, 2006). They
provide detailed methodological descriptions to estimate emissions and removals, as well as recommendations to

---

[22]The last revision has been made by COP 19 in 2013 (UNFCCC, 2013)

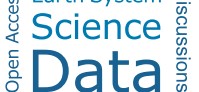

collect the activity data needed. As a general overall requirement, the UNFCCC reporting guidelines stipulate that
reporting under the Convention and the Kyoto Protocol must follow the five key principles of transparency, accuracy,
completeness, consistency and comparability (TACCC).

The reporting under UNFCCC shall meet the TACCC principles. The three main GHGs are reported in

timeseries from 1990 up to two years before the due date of the reporting. The reporting is strictly source category
based and is done under the Common Reporting Format tables (CRF), downloadable from the UNFCCC official
submission portal: https://unfccc.int/ghg-inventories-annex-i-parties/2021.

For the biogenic $CO_2$ emissions from sector 4 LULUCF, methods for the estimation of $CO_2$ removals differ

enormously among countries and land use categories. Each country uses its own country specific method which takes
into account specific national circumstances (as long as they are in accordance with the 2006 IPCC guidelines), as
well as IPCC default values, which are usually more conservative and result in higher uncertainties. The EU GHG
inventory underlies the assumption that the individual use of national country specific methods leads to more accurate
GHG estimates than the implementation of a single EU wide approach (UNFCCC, 2018b). Key categories for the
EU27 are 4.A.1 Forest Land: Land Use $CO_2$, 4.A.2. Forest Land: Land Use $CO_2$, 4.B.1 Cropland Land Use $CO_2$, 4.B.2
Cropland Land Use $CO_2$, 4.C.1 Grassland Land Use $CO_2$, 4.C.2 Grassland Land Use $CO_2$, 4.D.1 Wetlands Land Use
$CO_2$, 4.E.2 Settlements Land Use $CO_2$, and 4.G Harvested Wood Production Wood product $CO_2$. The tiered method
a country applies depends on the national circumstances and the individual conditions of the land, which explains the
variability of uncertainties among the sector itself as well as among EU countries.

Table A3 shows the mean values of all LULUCF categories for the EU27+UK NGHGI (2021).   The

contribution is calculated as the percentage of the sum of the absolute values of all the categories, in order to account
for differing signs.
*Table A3: LULUCF categories for the EU27+UK NGHGI (2021)*

| Category | Mean value for 1990-2020 [Tg C] | Contribution to gross LULUCF flux [%] |
|---|---|---|
| Forest land remaining forest land | -107 | 56.0 |
| Land converted to forest land | -13.0 | 6.80 |
| Cropland remaining cropland | 8.45 | 4.41 |
| Land converted to cropland | 14.0 | 7.33 |
| Grassland remaining grassland | 11.8 | 6.16 |
| Land converted to grassland | -8.22 | 4.23 |
| Wetlands remaining wetlands | 2.89 | 1.51 |
| Land converted to wetlands | 1.09 | 0.567 |
| Settlements remaining settlements | 1.42 | 0.744 |
| Land converted to settlements | 11.8 | 6.15 |





| Other land remaining other land | N/A | N/A |
|---|---|---|
| Land converted to other land | 0.135 | 0.0706 |
| Harvested wood products | -11.5 | 5.99 |



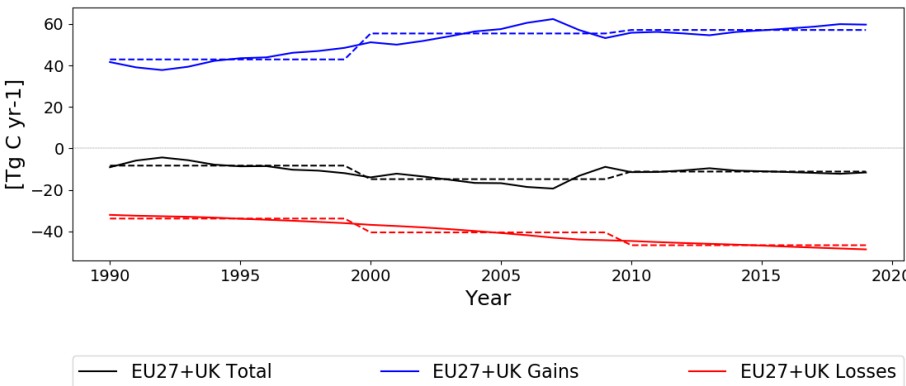


*Figure A2: The gains, losses, and total HWP pools from the Common Reporting Format tables for the European*
*Union (Convention), which covers the EU27+UK. Dashed lines show the averages for 1990-1999, 2000-2009, and*
*2010-2019 for easy comparison with Fig. 3.*

**Uncertainty:** Methodology for the NGHGI UNFCCC submissions are based on Chapter 3 of 2006 IPCC Guidelines
for National Greenhouse Gas Inventories and is the same as described in Appendix A1.

***ORCHIDEE***
ORCHIDEE is a general ecosystem model designed to be coupled to an atmospheric model in the context of
modeling the entire Earth system. As such, ORCHIDEE calculates its prognostic variables (i.e., a multitude of C,
$H_2O$ and energy fluxes) from the following environmental drivers: air temperature, wind speed, solar radiation, air
humidity, precipitation and atmospheric $CO_2$ concentration. As the run progresses, vegetation grows on each pixel,
divided into fifteen generic types (e.g., broadleaf temperate forests, C3 crops), which cycle carbon between the soil,
land surface, and atmosphere, through such processes such as photosynthesis, litter fall, and decay. Limited human
activities are included through the form of generic wood and crop harvests, which remove aboveground biomass on
an annual basis. The version reported here, ORCHIDEE-N v3, includes a dynamic nitrogen cycle coupled to the



vegetation carbon cycle which results in, among other things, limitations on photosynthesis in nitrogen-poor
environments (Vuichard et al., 2019)

Among other environmental indicators, ORCHIDEE simulates positive and negative $CO_2$ emissions from

plant uptake, soil decomposition, and harvests across forests, grasslands, and croplands. Activity data is based on land
use and land cover maps.  For VERIFY, pixel land cover/land use fractions were based on a combination of the land
use map LUH2v2h and the land cover project of the Climate Change Initiative (CCI) program of the European Space
Agency (ESA).  The latter is based on purely remotely sensed methods, while the former makes use of national harvest
data from the U.N. Food and Agricultural Organization.
**LUH2v2-ESACCI**: "We describe here the input data and algorithms used to create the land cover maps specific for
our CMIP6 simulations using the historical/future reconstruction of land use states provided as reference datasets for
CMIP6 within the land use harmonization database LUH2v2h (Hurtt et al., 2020). More details are provided on the
devoted web page https://orchidas.lsce.ipsl.fr/dev/lccci which shows further tabular, graphical and statistical data. The
overall approach relies on the combination of the LUH2v2 data with present-day land cover distribution derived from
satellite observations for the past decades. The main task consists in allocating the land-use types from LUH2v2 in
the different PFTs for the historical period and the future scenarios. The natural vegetation in each grid cell is defined
as the PFT distribution derived from the ESA-CCI land cover product for the year 2010 to which pasture fraction and
crop fraction from LUH2v2 (for the year 2010) have been subtracted from grass and crop PFTs. This characterization
of the natural vegetation in terms of PFT distribution is assumed invariant in time and is used for both the historical
period and the different future scenarios" (Lurton et al., 2020).

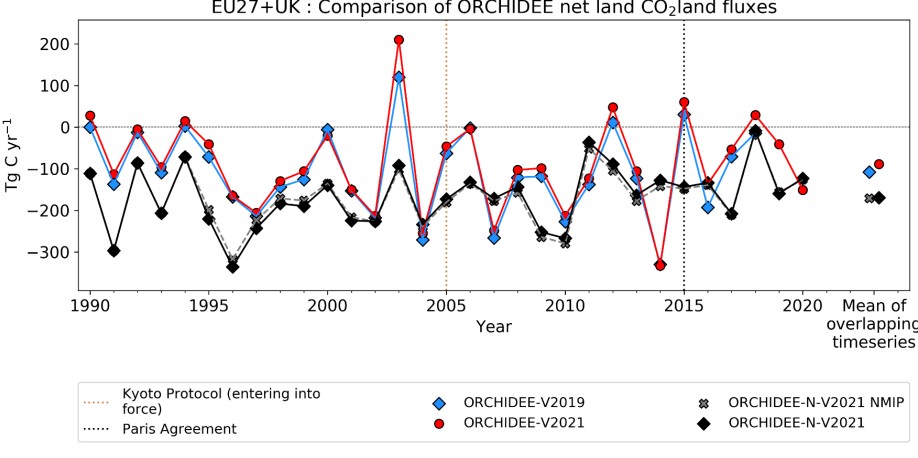


*Figure A3: A comparison of the version of ORCHIDEE used in previous synthesis of Petrescu et al. (2021b) compared*
*to the same version using the forcing prepared for this work (ORCHIDEE-V2021) and the version with the coupled*
*C-N cycle from this work (ORCHIDEE-N-V2021).  For the current work, both the version shown with the Europe-*



*specific nitrogen forcing prepared under VERIFY for the years 1995-2018 (ORCHIDEE-N-V2021) and that using the*
*standard nitrogen forcing from the N₂O Model Intercomparison Project (NMIP; Tian et al., 2018) as supplied to the*
*TRENDY model intercomparison is shown (ORCHIDEE-N-V2021 NMIP).*

**Uncertainty:** In the ORCHIDEE model, uncertainty arises from three primary sources: parameters, forcing data
(including spatial and temporal resolution), and model structure.  Some researchers argue that the initial state of the
model (i.e., the values of the various carbon and water pools at the beginning of the production run, following model
spinup) represents a fourth area. However, the initial state of this version of ORCHIDEE is defined by its equilibrium
state, and therefore a strong function of the parameters, forcing data, and model structure, with the only independent
choice being the target year of the initial state. Out of the three primary areas of uncertainty, the climate forcing data
is dictated by the VERIFY project itself, thus removing that source from explaining observed differences among the
models, although it can still contribute to uncertainty between the ORCHIDEE results and the national inventories.
The land use/land cover maps, another major source of uncertainty for ORCHIDEE carbon fluxes, have also been
harmonized to a large extent between the bottom-up carbon budget models in the project. Parameter uncertainty and
model structure thus represent the two largest sources of potential disagreement between ORCHIDEE and the other
bottom-up carbon budget models. Computational cost prevents a full characterization of uncertainty due to parameter
selection in ORCHIDEE (and dynamic global vegetation models in general), and uncertainties in model structure
require the use of multiple models of the same type but including different physical processes. Such a comparison has
not been done in the context of VERIFY, although the results from the TRENDY suite of models shown in Fig. 8 give
a good indication of this.  Figure A3 shows a small influence from the nitrogen forcing, likely because the European
nitrogen forcing is only available from 1995-2018 and ORCHIDEE carries out almost 500 years of simulation prior
to this point.  Many major carbon pools (i.e., woody biomass, soil carbon) have built up a large amount of inertia over
that time and are unlikely to undergo dramatic changes for any realistic forcing over the past.  A similar conclusion
can be reached from simulations ORCHIDEE-V2019 and ORCHIDEE-V2021 in Fig. A3, which only differ in
meteorological forcing from 1981-2020.

***CABLE-POP***
CABLE-POP (Haverd *et al.*, 2018) is a global terrestrial biosphere model developed around a
core biogeophysics module (Wang & Leuning, 1998) and a biogeochemistry module including cycles of nitrogen and
phosphorus (Wang *et al.*, 2010). Only nitrogen cycling was turned on for the present simulations. The model also
includes modules simulating woody demography (Haverd *et al.*, 2013) as well as land use change and land
management (Haverd *et al.*, 2018). The model distinguishes seven plant functional types which can co-occur in a
given grid cell. CABLE-POP does not simulate (natural) dynamic vegetation and the distribution and cover fraction
of PFTs is only affected by land use change. Forest demography (establishment, age class distribution, mortality) is
accounted for in the simulations, as are natural disturbances and forest management (wood harvest).
For the simulations described here, a baseline land cover map was created from the HILDA+ dataset for the year 1901
and vegetation classes in the dataset were reclassified to correspond to PFTs represented in CABLE-POP. Land use



transitions as well as land management (harvest) were prescribed from the LUH2v2h dataset over the entire simulation
period. Crops and pastures are treated as C3 grasses but are subject to agricultural harvest fluxes as given by LUH2v2h.
The use of HILDA+ data for the land cover distribution and the LUH2v2h for the representation of land cover/land
use change likely introduced additional uncertainties resulting from a potential mismatch between the two data sets.


***$CO_2$ Emissions from inland waters***
In this study we did not update these estimates and they are therefore identical to those in Petrescu et al.
(2021b). These estimates represent a climatology of average annual $CO_2$ emissions from rivers, lakes and reservoirs
at the spatial resolution of 0.1°. The approach combines $CO_2$ evasion fluxes from the global river network, as estimated
by the empirical model of Lauerwald et al. (2015), with the lakes and reservoirs estimates by Hastie et al. (2019) for
the boreal biome and by Raymond et al. (2013) for the lower latitudes. The Lauerwald et al. and Hastie et al. studies
follow the same approach and rely on the development of a statistical prediction model for inland water $pCO_2$ at 0.5°
using global, high-resolution geodata. The $pCO_2$ climatology was then combined with different estimates of the gas
transfer velocity k to produce the resulting map of $CO_2$ evasion. The Raymond et al. study only provides mean flux
densities at the much coarser spatial resolution of the so-called COSCAT regions. All estimates were then downscaled
to 0.1° using the spatial distribution of European inland water bodies. Note that in contrast to Hastie et al. (2019), the
areal distribution of lakes was extracted from the HYDROLAKES database (Messager et al., 2016), to be consistent
with the estimates of inland water $N_2O$ and $CH_4$ presented by Petrescu et al. (2021b).
**Uncertainty**: Monte Carlo simulations were performed to constrain uncertainties resulting from both the $pCO_2$
prediction equation and the choice of the k formulation.

***CBM***
The Carbon Budget Model developed by the Canadian Forest Service (CBM-CFS3), can simulate the
historical and future stand- and landscape-level C dynamics under different scenarios of harvest and natural
disturbances (fires, storms), according to the standards described by the IPCC (Kurz et al., 2009). Since 2009, the
CBM has been tested and validated by the Joint Research Centre of the European Commission (JRC), and adapted to
the European forests. It is currently applied to 26 EU Member States, both at country and NUTS2 level (Pilli et al.,

2016).

Based on the model framework, each stand is described by area, age and land use classes and up to 10
classifiers based on administrative and ecological information and on silvicultural parameters (such as forest
composition and management strategy). A set of yield tables define the merchantable volume production for each
species while species-specific allometric equations convert merchantable volume production into aboveground
biomass at stand-level. At the end of each year the model provides data on the net primary production (NPP), carbon
stocks and fluxes, as the annual C transfers between pools and to the forest product sector.



The model can support policy anticipation, formulation and evaluation under the LULUCF sector, and it is

used to estimate the current and future forest C dynamics, both as a verification tool (i.e., to compare the results with
the estimates provided by other models) and to support the EU legislation on the LULUCF sector (Grassi et al., 2018a).
In the biomass sector, the CBM can be used in combination with other models, to estimate the maximum wood
potential and the forest C dynamic under different assumptions of harvest and land use change (Jonsson et al., 2018).
**Uncertainty**: Quantifying the overall uncertainty of CBM estimates is challenging because of the complexity of each
parameter. The uncertainty in CBM arises from three primary sources: parameters, forcing data (including spatial and
temporal resolution) and model structure. It is linked to both activity data and emission factors (area, biomass volume
implied by species specific equation to convert the merchantable volume to total aboveground biomass (used as a
biomass expansion factor)) as well to the capacity of each model to represent the original values, in this case estimated
through the mean percentage difference between the predicted and observed values. A detailed description of the
uncertainty methodology is found in Pilli et al. (2017).

**Explanatory note on the extrapolation of Net Biome Productivity for the period 2017-2020 (**Matteo Vizzarri,
Roberto Pilli, Giacomo Grassi, EC-JRC)
*Background*
We performed a linear extrapolation of forest Net Biome Productivity (NBP) by country (EU 25 Member States and
UK) in the period 2017-2020 based on the correlation between NBP and harvest from the period 2000-2015. Cyprus
and Malta are excluded from the analysis because of missing historical data.
*Input data*
Table A4 reports a summary of input data sources.
*Table A4: main input data used in the extrapolation of NBP for the period 2017-2020.*

|  | Unit | Temporal resolution | Source |
|---|---|---|---|
|  |  |  |  |

| **Wood removals (HWP pool)** | t C | Annual (2000-2015) | CBM calibration run |
|---|---|---|---|
| **Forest area** | ha | Annual (2000-2020) | FAOSTAT[23] |
| **Roundwood amount** | m³ | Annual (2000-2020) | FAOSTAT[24] |
| **NBP** | t C | Annual (2000-2015) | CBM calibration run |


*Assessment procedure*


The extrapolation of the NBP for the period 2017-2020 was obtained throughout the following steps:
1. For each country (EU 25 Member States + UK), we first calculated the **average conversion factor** –
representing a correspondence between one ton of biomass carbon removed and one cubic meter of wood per
hectare – for the period 2000-2015 through equation [1]:
$$CF_{2000-2015} = \sum_{t=2000}^{2015} \frac{HWP_t}{\frac{RW_t}{A_{2015}}} \quad \text{eq. (1)}$$

where: $CF_{2000-2015}$ is the average conversion factor per hectare in the period 2000-2015 (t C     m⁻³   ha⁻
¹); $HWP_t$ is the carbon content per ha in harvested wood products in year $t$ (t C year⁻¹),    as derived from the CBM
model run; $RW$ is the total roundwood removals in year $t$ (m³ year⁻¹)    (source: FAOSTAT[25]); $A_{2015}$ is the managed
forest area in year 2015 (ha; source: Forest Europe    2015).
2. Using the average conversion factor estimated in eq. 1, we converted, for each country, the total roundwood
removals per ha derived from FAOSTAT for the period 2017 2020, to the corresponding amount of carbon
removals per ha, through equation [2]:

$$HWP_{conv} = CF_{2000-2015} \cdot \left(\frac{RW_t}{A_{2015}}\right) [\forall t = 2017 \div 2020] \qquad \text{eq. (2)}$$

where: $HWP_{conv}$ is the amount of carbon removals per hectare in year $t$ (t C ha⁻¹ year⁻¹); $CF_{2000-2015}$ is the
average conversion factor per hectare in the period 2000-2015 (t C m⁻³ ha⁻¹); $RW_t$ is the total roundwood in year $t$
(m³ year⁻¹) (source: FAOSTAT[26]); $A_{2015}$ is the managed forest area in the year 2015 (ha).
3. Then, for each country and the period 2000-2015, we performed a **linear regression** to search for significant
correlation between the harvest amount (i.e. HWP in t C ha⁻¹ yr⁻¹) and NBP, according to the generalized
equation:
$$NBP = a + b \cdot (HWP) \quad \text{eq. (3)}$$
In this case, we assumed NBP as the dependent variable (t C ha⁻¹ year⁻¹), the amount of harvest (t C ha⁻¹ year⁻
¹) as the main driver affecting the short term evolution of NBP, in absence of other exogenous natural
disturbances; $a$ is the intercept of the linear trendline; $b$ is the coefficient of the independent variable harvest

---

[23] https://www.fao.org/faostat/en/#data/RL
[24] https://www.fao.org/faostat/en/#data/FO
[25] https://www.fao.org/faostat/en/#data/FO
[26] https://www.fao.org/faostat/en/#data/FO



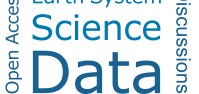

amount (i.e. HWP) (m$^3$ ha$^{-1}$ year$^{-1}$). This approach is consistent with the methodological assumptions reported
in Jonsson et al. (2021).
4. We finally calculated the **NBP in the period 2017-2020** for each country through equation [4]:
$$NBP_{t,m} = (a + b \cdot HWP_{conv})_{t,m} \quad \text{eq. (4)}$$

where: $NBP_{t,m}$ is the Net Biome Productivity for year t, country m (t C ha$^{-1}$ year$^{-1}$); $a_{t,m}$ is the intercept
of the linear trendline for year $t$, country $m$; $b_{t,m}$ is the coefficient of the independent variable in the trendline;
$HWP_{conv\,(t,m)}$ is the amount of carbon removal per ha for year $t$, country $m$ (t C ha$^{-1}$ year$^{-1}$).
Forest area and parameters used in equation [4] by country are reported in Table A5.
*Table A5: country-based forest area in 2015 and parameters used in equation [4]. *: significant (p<0.05); NS: not*
*significant (p>0.05).*

| EU 25 + UK | CF (2000-2015) | Intercept (*a*) | Coefficient (*b*) | p<0.05 |
|---|---|---|---|---|
| **Austria** | 0.28 | 2.60 | -1.57 | * |
| **Belgium** | 0.18 | 2.97 | -1.54 | * |
| **Bulgaria** | 0.22 | 1.17 | -2.13 | * |
| **Croatia** | 0.28 | 1.42 | -1.27 | * |
| **Czechia** | 0.22 | 2.55 | -1.21 | * |
| **Denmark** | 0.16 | 1.92 | -1.21 | * |
| **Estonia** | 0.20 | 1.16 | -1.08 | * |
| **Finland** | 0.23 | 1.15 | -1.20 | * |



| EU 25 + UK | CF (2000-2015) | Intercept (*a*) | Coefficient (*b*) | *p*<0.05 |
|---|---|---|---|---|
| **France** | 0.19 | 1.63 | -1.17 | * |
| **Germany** | 0.21 | 2.55 | -1.23 | * |
| **Greece** | 0.20 | 1.17 | -1.75 | ns |
| **Hungary** | 0.27 | 1.50 | -1.54 | * |
| **Ireland** | 0.18 | 6.12 | -5.45 | * |
| **Italy** | 0.23 | 0.69 | 0.39 | ns |
| **Latvia** | 0.19 | 2.00 | -1.77 | * |
| **Lithuania** | 0.22 | 1.11 | -0.89 | * |
| **Luxembourg** | 0.20 | 1.79 | -1.40 | * |
| **Netherlands** | 0.22 | 2.44 | -2.01 | * |
| **Poland** | 0.21 | 2.49 | -2.16 | * |
| **Portugal** | 0.29 | 1.39 | -1.01 | * |
| **Romania** | 0.32 | 1.54 | -1.65 | * |
| **Slovakia** | 0.28 | 2.57 | -1.42 | * |
| **Slovenia** | 0.24 | 2.07 | -1.55 | * |
| **Spain** | 0.28 | 0.26 | 0.18 | ns |
| **Sweden** | 0.23 | 1.02 | -1.20 | * |



| EU 25 + UK | CF (2000-2015) | Intercept (*a*) | Coefficient (*b*) | *p* < 0.05 |
|---|---|---|---|---|
| **United Kingdom** | 0.19 | 2.27 | -1.34 | * |


*Additional notes*
Because of biased estimates, values for the year 2016 were excluded from this analysis.
Extrapolated NBP for Czech Republic, Ireland and Netherlands were negative (thus showing emissions) because of
an increase of harvest in the corresponding years (2017-2020) compared to the previous period 2000-2015. Estonia
shows negative extrapolated NBP only for the year 2018.
**EFISCEN-Space**
The European Forest Information SCENario Model (EFISCEN) is a large-scale forest model that projects
forest resource development on a regional to European scale. The model uses aggregated national forest inventory
data as a main source of input to describe the current structure and composition of European forest resources. The
model projects the development of forest resources, based on scenarios for policy, management strategies and climate
change impacts. With the help of biomass expansion factors, stem wood volume is converted into whole-tree biomass
and subsequently to whole tree carbon stocks. Information on litter fall rates, felling residues and natural mortality is
used as input into the soil module YASSO (Liski et al., 2005), which is dynamically linked to EFISCEN and delivers
information on forest soil carbon stocks. The core of the EFISCEN model was developed by Prof. Ola Sallnäs at the
Swedish Agricultural University (Sallnäs, 1990). It has been applied to European countries in many studies since then,
dealing with a diversity of forest resource and policy aspects. A detailed model description is given by Verkerk et al.
(2016), with online information on availability and documentation of EFISCEN at http://efiscen.efi.int. The model
and its source code are freely available, distributed under the GNU General Public License conditions
(www.gnu.org/licenses/gpl-3.0.html).
In this report the follow-up of the EFISCEN model was used, called EFISCEN-Space (Schelhaas et al., in
prep). EFISCEN-Space simulates the development of the forest at the level of the plots as measured in the national
forest inventories, thereby providing a much higher spatial detail. The simulation is based on the distribution of trees
over diameter classes rather than age as in the old EFISCEN model. This allows the simulation of a wider variety of
stand structures, species mixtures and management options. Similar to the EFISCEN model, biomass expansion
factors and the YASSO soil carbon model are used to provide carbon balances for the forest. For use within VERIFY,
individual plot results are aggregated to a 0.125 degree grid. For the moment only 15 European member states are
included, partly due to the lack of an appropriate national forest inventory in the other member states, or because the
data could not be shared. No formal sensitivity and uncertainty analysis has been conducted yet.



Figure 5 shows results which vary from year-to-year.  In practice, the model was initialized with starting
years depending on the country, assuming that all data applied to this year. The model then produced stock and flux
changes for the subsequent five-year period, reporting a single mean value per pixel. To compute timeseries for the
EU27+UK, it was further assumed that these values were valid across 2005-2020. As the fluxes were given per square
meter of forest, they were scaled by the total area of the forest in each pixel found on the land use/land cover maps
used by the ORCHIDEE DGVM. This explains why the numbers vary from year to year; the flux per square meter of
forest does not change, but the total amount of forest area changes slightly. It should be noted that country-level values
available on the VERIFY website are only available for the five-year period for which the model produces a mean
result.

**Uncertainties: A** sensitivity analysis of EFISCEN v3 is described in detail in Chapter 6 of the user manual (Schelhaas
et al., 2007). Total sensitivity is caused by especially young forest growth, width of volume classes, age of felling and
few other variables. Scenario uncertainty comes on top of this when projecting in future. Within VERIFY, a full
uncertainty analysis has been completed, enabling the estimation of uncertainty ranges of the various output variables
(Schelhaas et al., 2020).

***EPIC-IIASA***
The Environmental Policy Integrated Climate (EPIC) model is a field-scale process-based model (Izaurralde
et al., 2006; Williams, 1990) which calculates, with a daily time step, crop growth and yield, hydrological, nutrient
and carbon cycling, soil temperature and moisture, soil erosion, tillage, and plant environment control. Potential crop
biomass is calculated from photosynthetically active radiation using the radiation-use-efficiency concept modified for
vapor pressure deficit and atmospheric $CO_2$ concentration effect. Potential biomass is adjusted to actual biomass
through daily stress caused by extreme temperatures, water and nutrient deficiency, or inadequate aeration. The
coupled organic C and N module in EPIC (Izaurralde et al., 2006) distributes organic C and N between three pools of
soil organic matter (active, slow and passive) and two litter compartments (metabolic and structural). EPIC calculates
potential transformations of the five compartments as regulated by soil moisture, temperature, oxygen, tillage and
lignin content. Daily potential transformations are adjusted to actual transformations when the combined N demand
in all receiving compartments exceeds the N supply from the soil. The transformed components are partitioned into
$CO_2$ (heterotrophic respiration), dissolved C in leaching (DOC) and the receiving SOC pools. EPIC also calculates
SOC loss with erosion.
The EPIC-IIASA (version EU) modeling platform was built by coupling the field-scale EPIC version 0810
with large-scale data on land cover (cropland and grasslands), soils, topography, field size, crop management practices
and grassland cutting intensity aggregated at a 1x1 km grid covering European countries (Balkovič et al., 2018, 2013).
In VERIFY, a total of 10 major European crops including winter wheat, winter rye, spring barley, grain maize, winter
rapeseed, sunflower, sugar beet, potatoes, soybean and rice were used to represent agricultural production systems in
European cropland. Crop fertilization and irrigation were estimated for NUTS2 statistical regions between 1995 and
2010 (Balkovič et al., 2013). For VERIFY, the simulations were carried out assuming conventional tillage, consisting

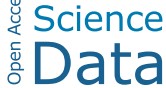

of two cultivation operations and moldboard plowing prior to sowing and an offset disking after harvesting of cereals.
Two row cultivations during the growing season were simulated for maize and one ridging operation for potatoes. It
was assumed that 20 % of crop residues are removed in the case of cereals (excluding maize), while no residues are
harvested for other crops.
A total of five managed grassland types with distinct temperature requirements, biomass productivity, and
phenology were used to represent the C-cycle in European grasslands. High-productive generic winter pasture and tall
fescue-based grasslands were used for Atlantic Europe, low fescue grasslands for the cool climates of Nordic regions
and high mountains, high-productive tall fescue-based grasslands and low-productive bluegrass types for continental
Europe, and low-productive bromegrass and high-productive winter pastures in the Mediterranean regions. Annual
nitrogen and carbon inputs, including inorganic and manure fertilization, and atmospheric N deposition, were obtained
from ISIMIP3 (Jägermeyr et al., 2021). In this dataset, the annual manure production and the fraction of manure from
livestock applied to cropland and rangeland were used from Zhang et al. (2017). The original manure data were re-
gridded to half-degree spatial resolution in ISMIP3. In the model, manure is applied as an organic fertilizer with a
C:N ration of 14.5:1. The organic carbon and nitrogen are added to the fresh organic litter pool where they decompose
in a manner identical to the fresh litter from vegetation, while mineral N from manure is added to the soil nitrate and
ammonium pools. The distribution of herbage biomass export intensity was constructed based on (Chang et al., 2016).
**Uncertainty:** In EPIC, uncertainties arise from three primary sources which were described in detail by ORCHIDEE.
A detailed sensitivity and uncertainty analysis of EPIC-IIASA regional carbon modeling is presented in (Balkovič et
al., 2020).

### *ECOSSE (grasslands)*
ECOSSE is a biogeochemical model that is based on the carbon model ROTH-C (Jenkinson and Rayner,
1977; Jenkinson et al. 1987; Coleman and Jenkinson, 1996) and the nitrogen-model SUNDIAL (Bradbury et al., 1993;
Smith et al., 1996). All major processes of the carbon and nitrogen dynamics are considered (Smith et al., 2010a,b).
Additionally, in ECOSSE processes of minor relevance for mineral arable soils are implemented as well (e.g., methane
emissions) to have a better representation of processes that are relevant for other soils (e.g., organic soils). ECOSSE
can run in different modes and for different time steps. The two main modes are site specific and limited data. In the
later version, basic assumptions/estimates for parameters can be provided by the model. This increases the uncertainty
but makes ECOSSE a universal tool that can be applied for large scale simulations even if the data availability is
limited. To increase the accuracy in the site-specific version of the model, detailed information about soil properties,
plant input, nutrient application and management can be added as available.
During the decomposition process, material is exchanged between the SOM pools according to first order
rate equations, characterized by a specific rate constant for each pool, and modified according to rate modifiers
dependent on the temperature, moisture, crop cover and pH of the soil. The model includes five pools with one of
them being inert. The N content of the soil follows the decomposition of the SOM, with a stable C:N ratio defined for
each pool at a given pH, and N being either mineralized or immobilized to maintain that ratio. Nitrogen released from
decomposing SOM as ammonium ($NH_4^+$) or added to the soil may be nitrified to nitrate ($NO_3^-$).





For spatial simulations the model is implemented in a spatial model platform. This allows to aggregate the

input parameter for the desired resolution. ECOSSE is a one-dimensional model and the model platform provides the

input data in a spatial distribution and aggregates the model outputs for further analysis. While climate data are

interpolated, soil data are represented by the dominant soil type or by the proportional representation of the different

soil types in the spatial simulation unit (this is in VERIFY a grid cell).

**Uncertainty:** In ECOSSE, uncertainty arises from three primary sources: parameters, forcing data (including spatial

and temporal resolution), and model structure. These uncertainties are not yet quantified.

### *Bookkeeping models*

We make use of data from two bookkeeping models: **BLUE** (Hansis et al., 2015) and **H&N** (Houghton &

Nassikas, 2017).

The **BLUE** model provides a data-driven estimate of the net land use change fluxes. BLUE stands for

"bookkeeping of land use emissions". Bookkeeping models (Hansis, 2015; Houghton, 1983) calculate land-use change

$CO_2$ emissions (sources and sinks) for transitions between various natural vegetation types and agricultural lands. The

bookkeeping approaches keep track of the carbon stored in vegetation, soils, and products before and after the land-

use change. In BLUE, land-use forcing is taken from the Land Use Harmonization, LUH2, for estimates within the

annual global carbon budget. The model provides data at annual time steps and 0.25 degree resolution. Temporal

evolution of carbon gain or loss, i.e., how fast carbon pools decay or regrow following a land-use change, is based on

response curves derived from literature. The response curves describe decay of vegetation and soil carbon, including

transfer to product pools of different lifetimes, as well as carbon uptake due to regrowth of vegetation and subsequent

refilling of soil carbon pools. In this report we present two versions of BLUE: BLUEvVERIFY and BLUEvGCP. The

BLUEvVERIFY version is a set of runs made for VERIFY, using the Hilda+[27] product (Ganzenmüller et al., 2022).

The **H&N** model (Houghton et al., 1983) calculates land-use change $CO_2$ emissions and uptake fluxes for

transitions between various natural vegetation types and agricultural lands (croplands and pastures). The original

bookkeeping approach of Houghton (2003) keeps track of the carbon stored in vegetation and soils before and after

the land-use change. Carbon gain or loss is based on response curves derived from literature. The response curves

describe decay of vegetation and soil carbon, including transfer to product pools of different life-times, as well as

carbon uptake due to regrowth of vegetation and consequent re-filling of soil carbon pools. Natural vegetation can

generally be distinguished into primary and secondary land. For forests, a primary forest that is cleared can never

return back to its original carbon density. Instead, long- term degradation of primary forest is assumed and represented

by lowered standing vegetation and soil carbon stocks in the secondary forests. Apart from land use transitions

between different types of vegetation cover, forest management practices in the form of wood harvest volumes are

included. Different from dynamic global vegetation models, bookkeeping models ignore changes in environmental

conditions (climate, atmospheric $CO_2$, nitrogen deposition and other environmental factors). Carbon densities at a

given point in time are only influenced by the land use history, but not by the preceding changes in the environmental

---

[27]https://landchangestories.org/hildaplus/



state. Carbon densities are taken from observations in the literature and thus reflect environmental conditions of the
last decades. In this study an updated H&N version submitted to the GCP2021 is used.
**Uncertainty:** Uncertainties can be captured through simulations varying uncertain parameters, input data, or process
representation. A large contribution of uncertainty can be expected from various input datasets. Apparent uncertainties
arise from the land-use forcing data (Gasser et al., 2020; Hartung et al., 2021; Ganzenmüller et al., 2022), the
equilibrium carbon densities of soil and vegetation and allocation of material upon a land-use transition (Bastos et al.,
2021), and the response curves built to reflect carbon pool decay and regrowth after land-use transitions. Furthermore,
studies have shown that different accounting schemes (Hansis et al., 2015) and initialization settings at the start of the
simulations (Hartung et al., 2021) lead to different emission estimates even decades later.

*FAOSTAT*
FAOSTAT: Statistics Division of the Food and Agricultural Organization of the United Nations provides
updates for the LULUCF $CO_2$ emissions for the period 1990-2019, available at:
https://www.fao.org/faostat/en/#data/GT and its sub-domains. The FAOSTAT emissions land use database is
computed following a Tier 1 approach of IPCC (2006). Geospatial data are the source of AD for the estimates of
emissions from cultivation of organic soils, biomass and peat fires. GHG emissions are provided by countries, regions
and special groups, with global coverage, relative to the period 1990-present (with annual updates). Land Use Total
contains all GHG emissions and removals produced in the different Land Use sub-domains, representing four IPCC
Land Use categories, of which three land use categories: forest land, cropland, grassland and biomass burning.
LULUCF emissions consist of $CO_2$ associated with land use and change, including management activities. $CO_2$
emissions/removals are computed at Tier 3 using carbon stock change. To this end, FAOSTAT uses Forest area and
carbon stock data from FRA (2015), gap-filled and interpolated to generate annual time-series. As a result $CO_2$
emissions/removals are computed for forest land and net forest conversion, representing respectively IPCC categories
"Forest land" and "Forest land converted to other land uses". $CO_2$ emissions are provided as by country, regions and
special groups, with global coverage, relative to the period 1990-most recent available year (with annual updates),
expressed as net emissions/removals as Gg $CO_2$, by underlying land use emission sub-domain and by aggregate (land
use total).
**Uncertainty:** FAOSTAT uncertainties are not available.

*TRENDY DGVMs*
The TRENDY (Trends in net land-atmosphere carbon exchange over the period 1980-2010) project
represents a consortium of dynamic global vegetation models (DGVMs) following identical simulation protocols to
investigate spatial trends in carbon fluxes across the globe over the past century. As DGVMs, the models require
climate, carbon dioxide, and land use change input data to produce results. In TRENDY, all three of these are
harmonized to make the results across the whole suite of models more comparable. In the case of VERIFY, 15 of the
16 models for TRENDY v10 (except for ISAM) were used. While describing the details of all the models used here



is clearly not possible, DGVMs calculate prognostic variables (i.e., a multitude of C, $H_2O$ and energy fluxes) from the following environmental drivers: air temperature, wind speed, solar radiation, air humidity, precipitation and atmospheric $CO_2$ concentration. As the run progresses, vegetation grows on each pixel, divided into generic types which depend on the model (e.g., broadleaf temperate forests, C3 crops), which cycle carbon between the soil, land surface, and atmosphere, through such processes such as photosynthesis, litter fall, and decay. Limited human activities are included depending on the model, typically removing aboveground biomass on an annual basis.

Among other environmental indicators, DGVMs simulate positive and negative $CO_2$ emissions from plant uptake, soil decomposition, and harvests across forests, grasslands, and croplands. Activity data is based on land use and land cover maps and generally follows Approach 1 as described by the IPCC 2006 guidelines (enabling calculation of only net changes from year to year). For TRENDY, pixel land cover/land use fractions were based on the land use map LUH2 (Hurtt et al., 2020) and the HYDE land-use change data set (Klein Goldewijk et al., 2017a, b). Both of these maps rely on FAO statistics on agricultural land area and national harvest data.

**Uncertainty:** In TRENDY v10 uncertainties are model specific and described by Friedlingstein et al. (2022). The spread of the 15 TRENDY models used by this study (Fig. 8) gives an idea of the uncertainty due to model structure in dynamic global vegetation models, as the forcing data was harmonized for all models.

### *Emissions from lateral transport of carbon (crops, wood, and inland waters)*

Production and consumption of carbon do not always occur on the same grid points. This is particularly relevant for the land surface in the case of crops, wood products, and carbon transfers through the inland water network. The purpose of the work here is primarily to convert the flux changes of the top-down inversions into NGHGI-like stock changes. To convert the flux changes of the inversions (where a positive number represents a flux to the atmosphere, i.e., a source) into NGHGI-like stock changes, one needs to add the crop sink and remove the crop source. The crop sink comes from production numbers in the FAO food balance sheets, while the source is estimated by production plus import minus export (all from the FAO food balance sheets), and both terms make use of conversion factors for each commodity. We take the forestry balance sheets of FAO (production, import and export per commodity), and convert to C mass. For a given year, the fraction of this mass that is released later in the atmosphere in each country is modeled with an e-folding decrease driven by experimental data per country (Mason Earles et al., 2012). Lateral transfers of carbon through inland waters also need to be removed from the inversion results as the terrestrial biospheric CO2 uptake leached into the inland water network represents a carbon sink, while the fraction that is subsequently re-emitted as $CO_2$ before reaching the ocean is a carbon source. The inland water $CO_2$ outgassing originates from carbon imported with runoff as dissolved $CO_2$ or produced in-situ from the decomposition of terrestrial carbon inputs. Note further that a fraction of the net-uptake of atmospheric $CO_2$ over the continents does not accumulate on land, but is instead exported through the inland water network to the oceans; this fraction is included in the calculation. For regional carbon budgets, any river carbon export outside the boundaries of the region of interest (in this case, EU27+UK) needs to be known to separate net uptake of atmospheric C from the actual land C sink.





Carbon fluxes to the atmosphere from rivers and lakes were obtained from maps described in Zscheischler
et al. (2017). These methods are similar to those described previously in Petrescu et al. (2021b). The primary difference
is that the updated estimates include smaller lakes and reservoirs not represented in the Global Lakes and Wetland
Database through the use of a scaling law, in addition to the older results being created specifically for Europe, while
the newer results are part of a global product. The emissions from the previous work totaled 25.5 Tg C yr-1 for the
EU27+UK, while those used here are 19.8 Tg C yr-1 (with no variability from year-to-year). This difference is
therefore small compared to the river C export, which is included this year for the first time and averages -73.8 Tg for
the period 1990-2020.
One important difference between the fluvial carbon exports reported here and those from a previous work
(Ciais et al., 2021) are that those reported here are rescaled to reasonable global flux reflecting bias in inter-
hemispheric exchange. Similar to Bastos et al. (2020), the dissolved organic carbon (DOC) and particulate organic
carbon (POC) exports were rescaled per basin to match the estimates of Resplandy et al. (2018).  The global total
organic C was finally rescaled to 500 Tg C/yr, which is considered a reasonable global number based on different
reviews and synthesis efforts (Regnier et al., 2013).


***Top-down $CO_2$ emissions estimates***
***CarboScope-Regional***
**CarboScopeRegional (CSR)** (Munassar et al., 2022): CSR is a Bayesian Framework inversion system that employs
a-priori knowledge of the surface-atmosphere carbon fluxes to regularize the solution of the ill-posed inverse problem
arising from the sparseness of observations sampled over limited geographical locations throughout the domain of
interest. Due to the heterogeneity of biogenic fluxes, the convention in CSR is to optimize Net Ecosystem Exchange
(NEE) against measurements of $CO_2$ dry model fraction at 3-hourly temporal and 0.5° horizontal resolutions, while
ocean fluxes and anthropogenic emissions are prescribed given their better knowledge available compared with NEE.
The prior flux uncertainty is assumed to have a uniform shape in space and time and its spatial correlation is fitted to
a hyperbolic decay function following the assumption of Kountouris et al. (2018a, b). Model-data mismatch
uncertainty is defined weekly in the measurement covariance matrix varying over sites from 0.5 to 4 (ppm) according
to the ability for atmospheric transport models to sample the true concentration at such locations (Rödenbeck, 2005).
This uncertainty implicitly encompasses the combinations of atmospheric transport, representation, and measurement
errors and is assumed to be independent at different locations. To separate the lateral influences originating from
outside of the regional domain, the two-step scheme inversion (Rödenbeck et al., 2009) is applied to run a global
inversion with the Eulerian model TM3 at coarse resolutions to provide the lateral boundary conditions to the regional
inversion. In the regional inversion runs, the Lagrangian model STILT (Lin et al., 2003), forced by IFS data from
ECMWF, is used to calculate the surface sensitivities "footprints" over the regional site network (receptors) at hourly
temporal and 0.25° spatial resolutions. Typically, the prior fluxes of $CO_2$ are obtained from bottom-up model
estimations. Thus, the diagnostic biosphere model VPRM calculates the biogenic fluxes at hourly temporal resolution



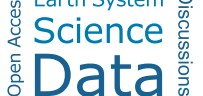

preserving the diurnal cycle. Ocean fluxes are obtained from the CarboScope ocean-based fluxes developed in-house by Rödenbeck et al. (2014). Emissions of fossil fuel are taken from EDGAR_v4.3 inventories updated every year based on the British Petroleum statistics (BP), and are distributed in space and time using the COFFEE approach (Steinbach et al., 2011) according to fuel-type and sector.

The v2021 CSR inversions underwent updates in comparison with the previous v2019:

- v2019 from Petrescu et al. (2021b) excluded observations from two sites: La Muela (LMU) in Spain because of inconsistent datasets between releases, and Finokalia (FKL) in Greece due to errors in the dataset. These exclusions resulted in a larger C sink from 2013 onwards (Fig. 9, upper plot). FKL observations start at this time and are the dominant impact over south-east Europe, as it is the only site located there. In v2021 inversions, we included corrected datasets from the FKL site.
- Two new flask sites were included in the v2021 inversions: Shetland Islands in the UK and Centro Investigacion Baja in Spain. These sites are also used in the CarboScope global inversion that provides the far-field contributions to the EU domain.

**Uncertainty**: Uncertainties from top-down (TD) estimates can be reported as posterior Bayesian uncertainties. Following the methodology of Chevallier et al. (2007) the CSR inversion system computed maps of uncertainty reductions for 2006 and 2018 (Fig. A4). The reduction is carried out through an ensemble of 40 members of inversions using error realizations following a Monte Carlo (MC) approach. Circles on maps refer to locations of stations. In the inversion system, a MC method is used to generate N ensembles of realizations of prior errors and model-data mismatch errors. The inversion is repeated for each ensemble member starting from each set of prior and model-data mismatch errors to generate posterior fluxes. The posterior uncertainty is calculated as the spread over the optimized fluxes across the whole ensemble. The uncertainty reduction is then calculated as 1- ($\sigma$post / $\sigma$prior). It is clear that larger ensembles will lead to better convergence of the error reduction. However, due to computational limitations, 40 ensemble members were selected as a good compromise.

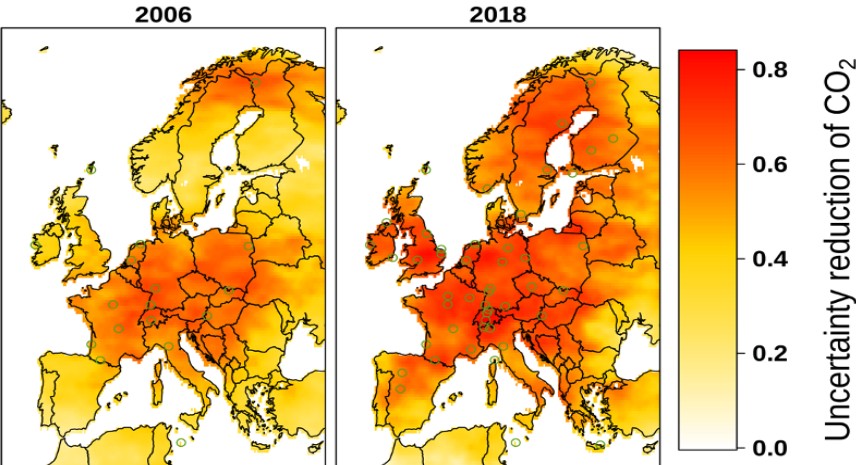


*Figure A4: CSR uncertainty reduction maps computed as* $1 - (\sigma_{post} / \sigma_{prior})$ *for 2006 and 2018 using a Monte Carlo*

*approach focused on prior errors. The circles represent the observation stations network.*



Figure A4 represents a preliminary attempt at how the inclusion of additional observation stations (additional
circles in the right-side figure for Germany, Switzerland, Finland compared to the left-side figure) might reduce the
uncertainty. However, the two different simulation years (2006 and 2018) might also differ in terms of other factors
which may lead to lower uncertainties in a given year (e.g., climatological conditions, such as the 2018 drought year).
Several caveats remain. When comparing the uncertainty over pixels or subregions in the domain of interest,
the maps of uncertainty reduction should be interpreted together with the maps of posterior uncertainty to give a better
illustration of the magnitude of uncertainty. The maps of uncertainty reduction reflect only the random uncertainties.
The systematic uncertainties are still poorly characterized, including uncertainties due to atmospheric transport
modeling, dependence on the prior fluxes, and the weighting between the prior and observation uncertainties. To
improve knowledge of the systematic uncertainties, dedicated studies with controlled comparisons between inversions
using different atmospheric transport models (such as planned with the Community Inversion Framework, Berchet et
al., 2021) are still needed. Furthermore, the posterior uncertainty and uncertainty reductions between inversions
depend on internal parameterizations, e.g., the weighting of prior and observation uncertainties. Future efforts should
focus on establishing best practices on how to set-up inversions and quantification of systematic uncertainties,
including as well tests of the fidelity of models against data (Simmonds et al., 2021).


***LUMIA***
The LUMIA inversion system (Monteil and Scholze, 2021) is a regional atmospheric inversion system, which
was designed to produce estimates of the land-atmosphere carbon exchanges based on in-situ $CO_2$ observations from
the ICOS network. It relies on the FLEXPART 10.4 Lagrangian transport model (Pisso et al., 2019) to compute the
transport of $CO_2$ fluxes within a regional domain (15°W; 33°N to 35°E, 73°N) at a 0.5°, 3-hourly resolution. Boundary





conditions are provided in the form of timeseries of far-field contributions at the observation sites, obtained from a
global TM5-4DVAR inversion (using the 2-step inversion approach of Rödenbeck et al., 2009). Both transport models
were driven by ECMWF ERA-Interim data, up to 2018, and by ECMWF ERA5 data afterwards.
The inversions solve for weekly offsets to the prior NEE/NBP estimate, at a variable spatial resolution,
highest where the observational coverage is better (up to 0.5° upwind of the observation sites). The optimal solution
is searched for using a variational inversion approach (preconditioned `conjugate gradient`). The
inversions were constrained by in−situ and flask observations from 66 European
observation sites, although only a subset of these sites is usually available at a
given time. The observation uncertainties were set to 1 ppm/week at all sites (the
uncertainty of a single observation is therefore higher, on average 5.2 ppm, and given
by √n, with n the number of assimilated observations at the same site in a ±3.5 day
window around the observation time). The prior NEE was produced using the LPJ−
GUESS model (Smith et al., 2014), driven by ECMWF ERA5 meteorological data.
The inversion also accounts for (prescribed) anthropogenic $CO_2$ fluxes from the EDGAR/TNO product
(https://doi.org/10.18160/Y9QV-S113) and for atmosphere-ocean $CO_2$ exchanges from the Jena-CarboScope
oc_v2021 product (https://www.bgc-jena.mpg.de/CarboScope/oc/oc_v2021.html). The uncertainties on the prior NEE
were set proportional to the sum of the absolute value of the 3-hourly fluxes in each 7-day optimization interval (so
the uncertainty is not zero even if the net flux is zero), and scaled to a total value of 0.45 PgC/year, accounting for
covariances based on Gaussian (spatial) and exponential (temporal) correlation decay functions, with correlation
lengths of respectively 500 km and 1 month (see Monteil and Scholze, 2021, for details).
The main differences from the LUMIA setup used in Thompson et al. (2014) are the specification of prior
and observation uncertainties (here made, on purpose, more comparable to those used in the CSR inversions), and the
implementation of flux optimization at a variable spatial resolution (which has negligible impact on the results but
improves the model performance).
### *CIF-CHIMERE - land $CO_2$*
CIF-CHIMERE is used for both $CO_2$ land and $CO_2$ fossil emission estimates, and this section only describes
the $CO_2$ land estimates.
The CIF-CHIMERE inversions have been generated with the variational mode of the Community Inversion
Framework (CIF, Berchet et al., 2021) coupled to the regional Eulerian atmospheric chemistry-transport model
CHIMERE (Menut et al., 2013; Mailler et al., 2017) and to its adjoint code. They are set-up in a manner that is close
to that of the PYVAR-CHIMERE inversions of Broquet et al. (2013), of Thompson et al. (2020) and of Monteil et al.
2068 (2020).

A European configuration of CHIMERE is used; this configuration covers latitudes 31.75-73.25°N and
longitudes 15.25°W -34.75°E with a 0.5°×0.5° horizontal resolution and 17 vertical layers up to 200 hPa.



Meteorological forcing for CHIMERE is generated using the European Center for Medium Range Weather
Forecasting (ECMWF) operational forecasts. Initial, lateral and top boundary conditions for $CO_2$ concentrations are
generated from the new CAMS global $CO_2$ inversions v20r2 (Chevallier et al., 2010).
The inversion assimilates in situ $CO_2$ data from continuous measurements stations compiled in the VERIFY
Deliverable D3.12 and in the Table A1 from the VERIFY CIF Inversion Protocol (Thompson et al., 2021). More
specifically, the inversion assimilates 1-hour averages of the measured $CO_2$ mole fractions during the time window
12:00-18:00 UTC for low altitude stations (below 1000 masl) and 0:00-6:00 UTC for high altitude stations (above
1000 masl). The inversion optimizes 6-hourly mean NEE and ocean fluxes at the 0.5°×0.5° resolution of CHIMERE.
The anthropogenic $CO_2$ emissions, considered as perfect and consequently not optimized in the inversions, are based
on the spatial distribution of the EDGAR-v4.2 inventory, on national and annual budgets from the BP (British
Petroleum) Statistics and on temporal profiles at hourly resolution derived with the COFFEE approach (Steinbach et
al., 2011).
The prior estimate of NEE and its uncertainty covariance matrix are specified using ORCHIDEE model
simulations of NEE and respiration, respectively, following the general approach of Broquet et al. (2011). The
temporal and spatial correlation scales for the prior uncertainty in NEE are set to ~1 month and 200 km (following the
diagnostics of Kountouris et al., 2015), with no correlation between the four 6-hour windows of the same day. The
ocean prior fluxes come from a hybrid product of the University of Bergen coastal ocean flux estimate and the
Rödenbeck global ocean estimate (Rodenbeck et al., 2014). Fluxes from biomass burning are ignored. The observation
error covariance matrix is set-up to be diagonal, ignoring the correlations between errors for different hourly averages
of the $CO_2$ measurements (which has been justified by the analysis of Broquet et al., 2011). The variances for hourly
data are based on the values from Broquet et al. (2013), which vary depending on the sites and season, and which are
derived from Radon model-data comparisons.
About 12 iterations are needed to reduce the norm of the gradient of J by 95 %, using the M1QN3 limited
memory quasi-Newton minimisation algorithm (Gilbert et Lemaréchal, 1989). To cover the whole analysis period
(2005-2020), a series of 7-month (including an overlapping of 15 days between consecutive periods) inversions is
performed. Posterior estimates of NEE at 1-hourly and 0.5°×0.5° spatial resolution are generated for the full period of
analysis.

**Uncertainty**: Estimates of the uncertainty of regional inversions over Europe can be found by comparing against the
results of the other regional inversions in this work (the ensembles of EUROCOM, CarboScopeRegional, and
LUMIA).

***GCP 2021***
Top-down estimates of land biosphere fluxes are provided by a number of different inverse modeling systems
that use atmospheric concentration data as input, as well as prior information on fossil emissions, ocean fluxes, and
land biosphere fluxes. The land biosphere fluxes, and in some systems the ocean fluxes, are estimated using a statistical
optimization involving atmospheric transport models. The inversion systems differ in the transport models used,



optimization methods, spatiotemporal resolution, boundary conditions, and prior error structure (spatial and temporal
correlation scales), thus using ensembles of such systems is expected to result in more robust top-down estimates.
For this study, the global inversion results are taken from all six of the models reported in the GCP 2021:
CTE (CarbonTracker Europe), CAMS (Copernicus Atmosphere Monitoring Service), CMS-Flux, JENA, NIES-
NIWA, and UoE, with spatial resolutions ranging from 1°x1° for certain regions to 4°x5°. For details see Friedlingstein
et al. (2022). Note that one of the ensemble members (CMS-Flux) only covers the period 2010-2020, and therefore
the ensemble results are only shown from 2010 until the last year common between all models (2018).

***EUROCOM***
Top-down estimates at regional scales (up to 0.25°x0.25° resolution) for the period 2009 – 2018 are taken
from three models used within EUROCOM (Monteil et al., 2020): LUMIA, PYVAR, and CSR. The NAME model
was excluded as visual inspection of monthly values identified it as a clear outlier. FLEXINVERT was excluded after
visual inspection of annual values identified it as a clear outlier (Fig. A5). These inversions make use of more than
30 atmospheric observing stations within Europe, including flask data and continuous observations. The CarboScope-
Regional (CSR) inversion system results were re-run for VERIFY using the extended period 2009-2020 using four
different settings: three network configurations using 15, 40, or 46 sites, and one using all 46 sites but a factor two
larger prior error correlation length scale (200 instead of 100 km). The CSR results reported to EUROCOM were not
used, being instead replaced by the mean of the four updated CSR runs.





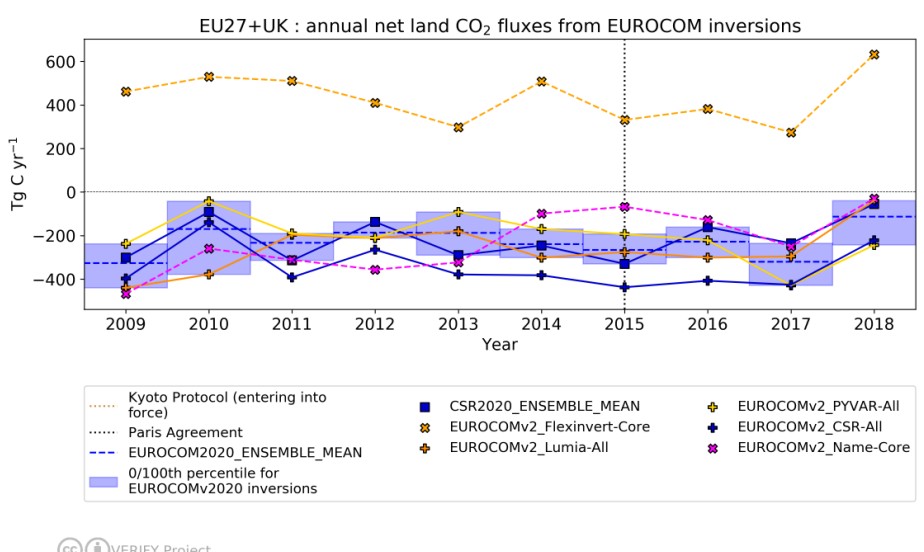

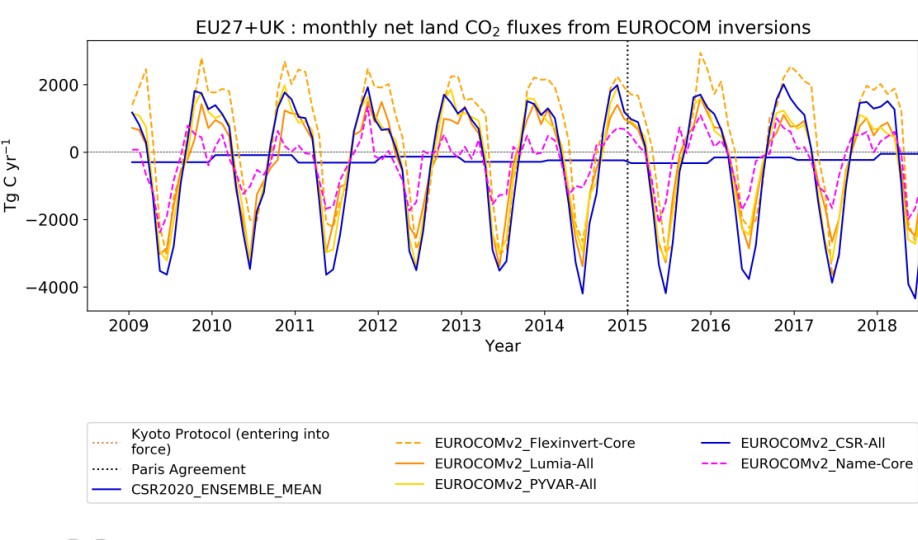

Figure A5: Annual (top) and monthly (bottom) timeseries for inversions in EUROCOM (Monteil et al., 2020).
Inversions with solid lines were retained for the ensemble used in this work (shown in blue in the top figure for clarity).
Note that the CSR values from EUROCOM have been replaced by the mean of four CSR simulations submitted under
the VERIFY project (Appendix A). Negative fluxes represent a sink into the land surface.




### Input data


### CRUERA


The ERA5-Land (Muñoz-Sabater, 2019; 2021) dataset at 0.1-degree resolution over the global land surface
at hourly resolution was aggregated to three-hourly resolution and extracted for a 0.125 degree grid over Europe
(35N:73N, 25W:45E) to match the grid used in previous efforts within the VERIFY project. The variables extracted
are:air temperatures, wind components, surface pressure, downwelling longwave radiation, downwelling shortwave
radiation, snowfall, and total precipitation. From these, additional variables were calculated: total windspeed, specific
humidity, relative humidity, and rainfall. Of these, the air temperature, downwelling shortwave radiation, specific
humidity, and total precipitation were re-aligned with the CRU observation dataset (Harris et al., 2020) from 1901–
2020 so that monthly means at 0.5 degree pixels correspond exactly. Variation from observations is therefore present
only on sub-monthly temporal scales and sub-0.5 degree spatial scales. At the time of the model intercomparison,
ERA5-Land was only available from 1981-2020. Consequently, the years 1901-1980 were taken from the UERRA
HARMONIE-V1 dataset from ECMWF re-aligned with CRU observations under the VERIFY project and used in
Petrescu et al. (2021b). For both datasets, results were aggregated to daily and monthly temporal resolution for use
as needed in some models.

### HILDA+


The full Hilda+ dataset is described in detail elsewhere (Winkler et al., 2020; Winkler et al., 2021). Hilda+
is available at 1x1km spatial and annual temporal resolution across the whole globe from 1960-2019 for six land use
classes (urban, cropland, pasture/rangeland, forest, unmanaged grass/shrubland, and sparse/no vegetation). The
algorithm uses earth observation data and land use statistics to generate annual land use/cover maps and transitions.
Probability maps for land use change categories are generated by using multiple earth-observation-based data
estimates of the extent of a given land cover category on a given pixel. The VERIFY project requires additional work
to satisfy the needs of the various modeling groups. For example, the maps were extended back to 1900 to meet the
needs of the DGVM groups. As observational data is lacking for the years pre-1960, the temporal trend of the
probability maps and the FAO land use database were used for extrapolation. In addition, forest areas were further
subdivided into six forest types (Evergreen, needle leaf; Evergreen, broad leaf; Deciduous, needle leaf; Deciduous,
broad leaf; Mixed; Unknown/Other) based on the ESA CCI land cover dataset (ESA 2017). Spatiotemporal forest type
dynamics within the forest category were included for 1992-2015. Before 1992 and after 2015, the static forest type
distribution as found in the years 1992 and 2015 in the ESA CCI land cover was assumed, respectively.

### NITROGEN DEPOSITION


Wet and dry deposition maps of ammonium and nitrate covering Europe from 1995-2018 were calculated at
0.5 degree spatial and monthly temporal resolution by the EMEP MSC-W model ("EMEP model" hereafter). The
EMEP model is a 3-D Eulerian chemistry transport model (CTM) developed at the EMEP Centre MSC-W under the
Framework of the UN Convention on Long-Range Transboundary Air Pollution (CLRTAP). The EMEP model has



traditionally been used to assess acidification, eutrophication and air quality over Europe, to underpin air quality policy
decisions (e.g., the Gothenburg Protocol), and has been under continuous development reflecting new scientific
knowledge and increasing computer power. The model was described in detail by Simpson et al. (2012) and later
updated as described in the annual EMEP status reports (Simpson et al., 2022, and references therein). For the
VERIFY project, output from the EMEP model version rv4.33 was used (Simpson et al., 2019), and averaged to annual
temporal resolution. In these simulations, the model was driven by meteorological data from the ECWMF IFS
(European Centre for Medium-Range Weather Forecasts – Integrated Forecast System) version cy40r1. Land-use data
were taken from the CORINE land-cover maps (de Smet and Hettelingh, 2001), the Stockholm Environment Institute
at York (SEIY), the Global Land Cover (GLC2000) database, and the Community Land Model (Oleson et al., 2010;
Lawrence et al., 2011). For more details see Simpson et al. (2017).

***COASTAL OCEAN FLUXES***
Ocean $CO_2$ fluxes were prepared for use as prior estimates in the regional inversions by combining the Rödenbeck
global ocean estimate (Rödenbeck et al., 2014) with coastal ocean fluxes for Europe prepared under the VERIFY
project. The combined dataset was prepared by choosing the coastal flux map when available and otherwise the open
ocean map. The coastal ocean fluxes were generated for an area extending from the western Mediterranean to the
Barents Sea and cover shelf areas down to 500 m water depth or 100 km distance from shore. First, surface ocean
$fCO_2$ observations are taken from the annually updated SOCAT database (Bakker et al., 2016) and gridded to a
monthly $0.125°x0.125°$ grid. $pCO_2$ maps are created based on fitting a set of driver data (including sea surface
temperature, mixed layer depth, chlorophyll concentration, and ice concentration) against the gridded $fCO_2$
observations. Both random forest and multi-linear regressions were used. The general procedure is described
elsewhere (Becker et al., 2021), but for the version reported here, random forest regressions were used instead of
multi-linear regression and the region was extended to the south. The dataset was divided into seven subregions
(Barents Sea, Norwegian Coast, North Sea, Baltic Sea, Northern Atlantic Coast/Celtic Sea, Southern Atlantic
coast/Bay of Biscay, western Mediterranean) and each region was fitted separately (leaf size: 20, bag size: 500). The
root mean square error (RMSE) of the random forest regressions was determined to be between 34 micro-atm (Baltic
Sea) and 10 micro-atm (Barents Sea). Random forest regressions consist of many regression trees, each based on a
random subset of data. Due to this internal structure, the overall RMSE can be seen as an out-of-box error estimate.
The final fluxes are calculated from the $pCO_2$ maps with the atmospheric $xCO_2$ in the marine boundary layer and six-
hourly wind speed data using the gas transfer coefficient and the Schmidt number after Wanninkhoff (2014), the
coefficient $a_q$ of 0.2814 calculated after Naegler (2009) and 6-hourly winds from the NCEP-DOE Reanalysis 2 product
(Kanamitsu et al., 2002).






*Appendix B*
*Overview figures*

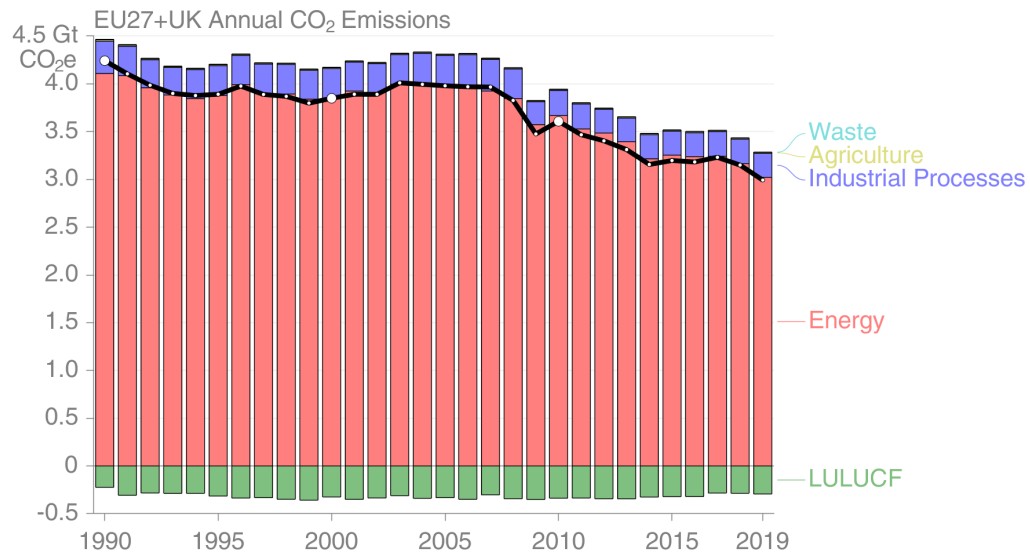


*Figure B1: EU27+UK total annual GHG emissions from UNFCCC NGHGI (2021) submissions split per*
*sector.*

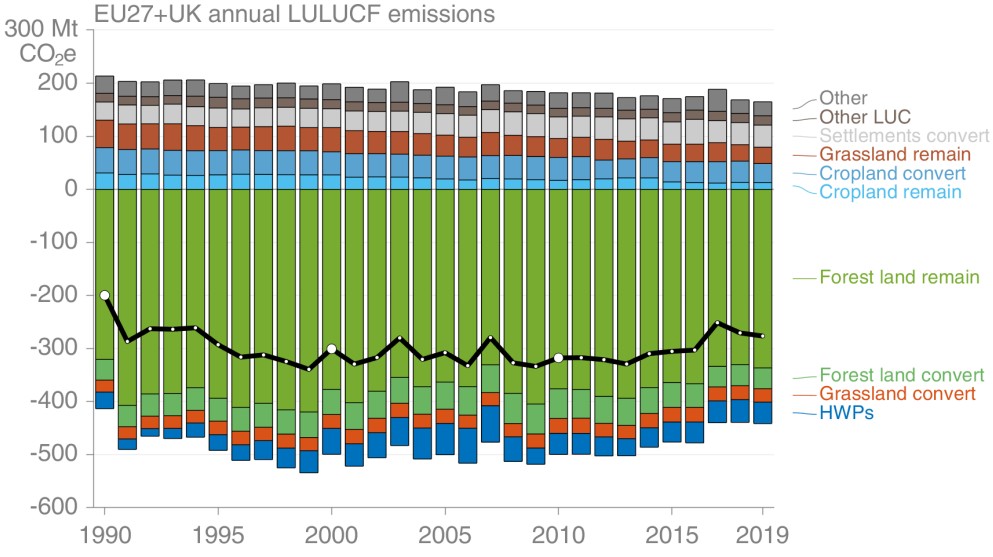




*Figure B2: EU27+UK total annual GHG emissions from the LULUCF sector split in categories and sub-categories,*
*according to UNFCCC NGHGI (2021).*

CO2 fossil breakdown by fuel type
Figure B3 shows the $CO_2$ fossil emission estimates from EU27+UK split by major source categories for each
dataset for a single year. Sectors 1, 2, 3, and 5 are included for the UNFCCC NGHGI (2021) total, without indirect
emissions. A breakdown of the nine other fossil BU data sources corresponding to UNFCCC NGHGI sectors or
categories is not currently available.
As in Andrew (2020), we observe good agreement for the EU27+UK between all BU data sources and the
UNFCCC NGHGI (2021) data. The figure presents updated estimates for the year 2017, the most recent year when
all datasets reported estimates. Sectors 1, 2, 3, and 5 are included for the UNFCCC NGHGI (2021) total, without
indirect emissions.
While most datasets agree well on total emissions, there are some differences. Both BP and the EIA include
bunker fuels and exclude most industrial process emissions. CEDS appears to be underestimating emissions from solid
fuels, for example lignite in Germany and oil shale in Estonia. IEA's emissions are lower because they exclude most
industrial processes. GCP's total matches the NGHGIs exactly by design but remaps some of the fossil fuels used in
non-energy processes from "Others" to the fuel types used. CDIAC, PRIMAP, and EDGAR v6.0 all report total
emissions very similar to the UNFCCC NGHGI (2021). Larger differences are seen in the disaggregation of fuel types,
generally because of differing definitions.

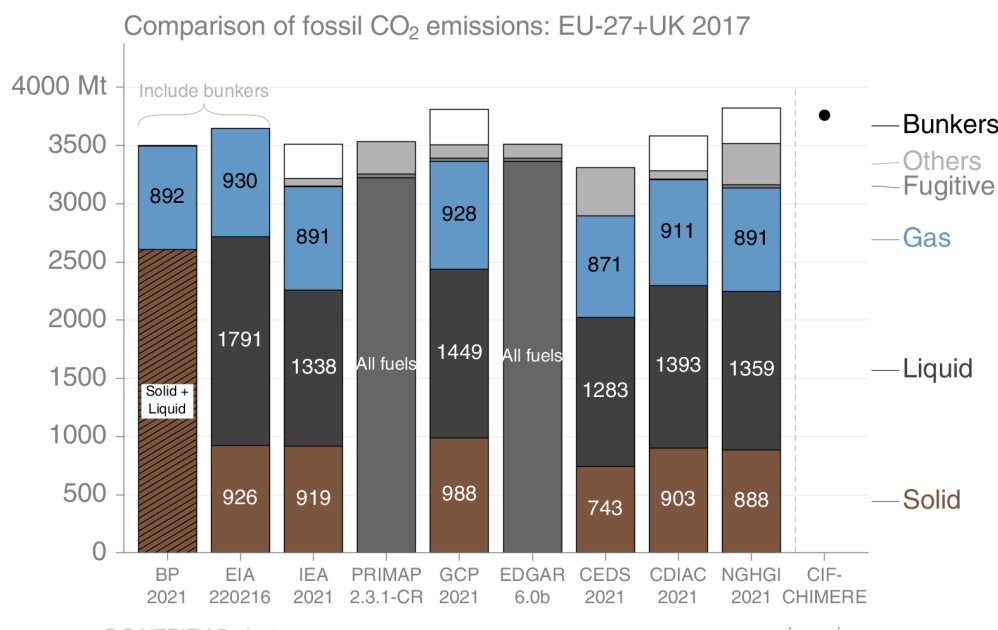






*Figure B3: EU27+UK total CO₂ fossil emissions, as reported by nine bottom-up data sources: BP, EIA, CEDS, EDGAR v6.0, GCP, IEA,*
*CDIAC, PRIMAPv2.3.1-CR and the UNFCCC NGHGI (2021) along with a top-down CIF-CHIMERE atmospheric inversion (black dot)*
*(Fortems-Cheiney and Broquet, 2021). This figure presents the split per fuel type for year 2017. "Others" is other*
*emissions in the UNFCCC's IPPU, and international bunker fuels (the white boxes) are not usually included in total*
*emissions at sub-global level. Neither* EDGAR[28] *(v6.0) nor PRIMAP publish a break-down by fuel type, so only the*
*total is shown. For BP, the method description allows for emissions from natural gas to be calculated from BP's*
*energy data, but the data for solid and liquid fuels are insufficiently disaggregated to allow replication of BP's*
*emissions calculation method for those fuels.*

***Source specific methodologies: AD, EFs and uncertainties***

*Table B1: Source specific activity data (AD), emission factors (EF) and uncertainty methodology for all current*
*VERIFY and non-VERIFY 2021 data products.*

| *Data sources CO₂ emission calculation* | *AD/Tier* | *EFs/Tier* | *Uncertainty assessment method* | *Emission data availability* |
|---|---|---|---|---|
| **UNFCCC NGHGI (2021)** | Country-specific information consistent with the IPCC Guidelines | IPCC guidelines / Country specific information for higher Tiers | IPCC guidelines (https://www.ipcc-nggip.iges.or.jp/public/2006gl/) for calculating the uncertainty of emissions based on the uncertainty of AD and EF, two different approaches: 1. Error propagation, 2. Monte Carlo Simulation<br><br>UBA Vienna provided yearly harmonized and gap-filled uncertainties | NGHGI official data (CRFs) are found at https://unfccc.int/ghg-inventories-annex-i-parties/2021 (last access: June 2022). |


| *Fossil CO₂* |
|---|

---

[28]EDGAR v6.0 provides significant sectoral disaggregation of emissions, but not by fuel type due to license restrictions with the underlying energy data from the IEA.



| | | | | |
|---|---|---|---|---|
| **BP**<br>**CDIAC**<br>**EIA**<br>**IEA**<br>**GCP**<br>**CEDS**<br>**PRIMAP-Hist** | *For further details, see Andrew (2020)* | | | |
| **EDGAR v6.0** | International Energy Agency (IEA) for fuel combustion<br>Food and Agricultural Organisation (FAO) for agriculture<br>US Geological Survey (USGS) for industrial processes (e.g., cement, lime, ammonia and ferroalloys production)<br>GGFR/NOAA for gas flaring<br>World Steel Association for iron and steel production<br>International Fertilisers Association (IFA) for urea consumption and production<br>Complete description of the data sources can be found in Janssens-Maenhout et al. (2019) and in Crippa et al. (2019) | IPCC (2006): Tier 1 or Tier 2 depending on the sector | Tier 1 with error propagation by fuel type for $CO_2$ and accounting for covariances. | https://edgar.jrc.ec.europa.eu/dataset_ghg60 |
| **CIF-CHIMERE** | Tier 3 top-down 0.1° x 0.1° resolution maps of annual averages of fossil $CO_2$ anthropogenic emissions from EDGAR v4.3.2 Assimilation of satellite atmospheric concentration data: total column CO from IASI, and tropospheric column $NO_2$ from OMI | Tier 3 top-down regional inversions of CO and NOx emissions using EMEP/CEIP as prior knowledge of the emissions and $CO_2$/CO and $CO_2$/NOx emission ratios associated with the combustion of fossil fuel from EDGARv4.3.2. | Bayesian analysis in the CO and NOx inversions along with propagation of uncertainties in f$CO_2$/CO and f$CO_2$/NOx emission ratios | Detailed gridded data can be obtained by contacting the data providers:<br>Gregoire Broquet<br>gregoire.broquet@lsce.ipsl.fr |


| |
|---|
| **$CO_2$ land: bottom-up** |



| | | | | |
|---|---|---|---|---|
| **BLUEvGCP** | From LUH2: data on wood harvest, land cover types (primary, secondary, pasture, crop), and gross land use transitions (e.g. from secondary to pasture and back); Based on Pongratz et al. (2008) and Ramankutty and Foley (1999): Plant functional types (PFTs) of natural vegetation types | Tier 3 (IPCC, 2006); PFT and land-cover type specific response curves describing the decay and regrowth of vegetation and soil carbon | N/A | Detailed gridded data can be obtained by contacting the data provider: Julia Pongratz: julia.pongratz@lmu.de |
| **BLUEvVERIFY** | Same as above with land cover from HILDA+ (Ganzenmüller et al., 2022) | | | |
| **H&N** | Simple assumptions about C-stock densities (per biome or per biome/country) based on literature | Transient change in C-stocks following a given transition (time dependent EF after an land use transition) | N/A | Detailed gridded data can be obtained by contacting the data provider: Richard A. Houghton rhoughton@woodwellclimate.org |
| **ECOSSE** | Tier 3 approach. The model is a point model, which provides spatial results by using spatial distributed input data (lateral fluxes are not considered). The model is a Tier 3 approach that is applied on grid map data, polygon organized input data or study sites. | IPCC (2006): Tier 3  The simulation results will be allocated due to the available information (size of spatial unit, representation of considered land use, etc.). | N/A | Detailed gridded data can be obtained by contacting the data providers: Kuhnert, Matthias matthias.kuhnert@abdn.ac.uk Pete Smith: pete.smith@abdn.ac.uk |
| **EPIC-IIASA Croplands** | Tier 3 approach. Cropland: static 1×1 km cropland mask from CORINE-PELCOM. Initial SOC stock from the Map of organic carbon content in the topsoil (Lugato et al., 2014). "Static" crop management and input intensity by NUTS2 calibrated for 1995-2010 (Balkovič et al., 2013). Crop harvested areas by NUTS2 from EUROSTAT. Parameterization of soil carbon routine was | IPCC (2006): Tier 3  Land management and input factors for the cropland remaining cropland category as simulated by the EPIC-IIASA modeling platform, assuming the business-as-usual crop management calibrated for the 1995-2010 | Sensitivity and uncertainty analysis of EPIC-IIASA regional soil carbon modeling (Balkovič et al, 2020). | Detailed gridded data can be obtained by contacting the data provider: Balkovič Juraj balkovic@iiasa.ac.at |



| | | | | |
|---|---|---|---|---|
| | updated based on Balkovič et al. (2020). | period. A 50-ha field is considered in each grid cell. | | |
| **EPIC-IIASA grasslands** | Tier 3 approach. Grassland: static 1x1 km mask from CORINE & PELCOM 2000, including pastures, herbaceous vegetation, heterogeneous agricultural areas, and permanent cropland. Initial SOC stock from the map of organic carbon content in the topsoil (Lugato et al., 2014) with a spin-up. Static grassland management and input intensity as adopted from (Chang et al., 2016) and ISIMIP (Jägermeyr et al., 2021). | IPCC (2006): Tier 3 Land management and input factors for the grassland remaining grassland category as simulated by the EPIC-IIASA modeling platform, calibrated for the 1995–2020 period. | N/A | Detailed gridded data can be obtained by contacting the data provider: Juraj Balkovič: balkovic@iiasa.ac.at |
| **ORCHIDEE** | For the land cover/land use input maps: data on wood harvest from the FAO | Tier 3 model, process based. Any emission factors enter in the form of generic parameters for a given ecosystem type fit against observational data (both site-level and remotely sensed). | None, though some information on uncertainty due to model structure is given by looking at the spread from the TRENDY suite of models, of which ORCHIDEE is a member. | Detailed gridded data can be obtained by contacting the data providers: Matthew McGrath matthew.mcgrath@lsce.ipsl.fr Philippe Peylin: peylin@lsce.ipsl.fr |


| | | | | |
|---|---|---|---|---|
| **CABLE-POP** | For the land cover/land use input maps: data on wood harvest and agricultural land from the FAO | Tier 3 model, process based. Any emission factors enter in the form of generic parameters for a given ecosystem type fit against | None, though some information on uncertainty due to model structure is given by looking at the spread from the TRENDY suite of models, of which CABLE-POP is a member. | Model output (gridded data) can be obtained by contacting the data provider: Jürgen Knauer: J.Knauer@westernsydney.edu.au |



| | | observational data (both site-level and remotely sensed). | | |
|---|---|---|---|---|


| | | | | |
|---|---|---|---|---|
| **TRENDY v10** | For the land cover/land use input maps: data on wood harvest and agricultural land from the FAO | Tier 3 models, process based. Any emission factors enter in the form of generic parameters for a given ecosystem type fit against observational data (both site-level and remotely sensed). | The spread of the 15 TRENDY models used gives an idea of the uncertainty due to model structure in dynamic global vegetation models, as the forcing data was harmonized for all models. | Detailed gridded data can be obtained by contacting the data provider: Sitch, Stephen S.A.Sitch@exeter.ac.uk |
| **Statistical prediction model for CO$_2$ in inland waters** | Hydrosheds 15s (Lehner et al., 2008) and Hydro1K (USGS, 2000) for river network, HYDROLAKES for lakes and reservoirs network and surface area (Messager et al., 2016); river pCO2 data from GloRiCh (Hartmann et al., 2014), lake pCO$_2$ database from Sobek et al. (2005); river channel slope and width calculated from GLOBE-DEM (GLOBE-Task-Team et al., 1999) and runoff data from Fekete et al. (2002). Geodata for predictors of pCO$_2$ and gas transfer coefficient include air temperature, precipitation and wind speed (Hijmans et al., 2005), population density (CIESIN and CIAT), catchment slope gradient (Hydrosheds 15s), and terrestrial NPP (Zhao et al., 2005) | N/A | Monte Carlo runs (uncertainty on pCO$_2$ and gas transfer velocity) | Detailed gridded data can be obtained by contacting the data providers: Ronny Lauerwald Ronny.Lauerwald@ulb.ac.be Pierre Regnier Pierre.Regnier@ulb.ac.be |
| **CBM** | National forest inventory data, Tier 2 | EFs directly calculated by model, based on specific parameters (i.e., turnover and decay rates) | N/A used from IPCC | Detailed gridded data can be obtained by contacting the data providers: Giacomo Grassi Giacomo.GRASSI@ec.europa.eu Matteo Vizzarri Matteo.VIZZARRI@ec.europa.eu |

Earth System
Open Access · Science
Data · Discussions

|  |  | defined by the user |  | Roberto Pilli roberto.pilli713@gmail.com |
| --- | --- | --- | --- | --- |
| **EFISCEN-Space** | National forest inventory data, Tier 3 | emission factor is calculated from net balance of growth minus harvest | Sensitivity analysis on EFISCEN V3 in the user manual (Schelhaas et al., 2007). Total sensitivity is caused by esp. young forest growth, width of volume classes, age of felling and few more. Scenario uncertainty comes on top of this when projecting in future. | Detailed gridded data can be obtained by contacting the data providers: Gert-Jan Nabuurs gert-jan.nabuurs@wur.nl Mart-Jan Schelhaas martjan.schelhaas@wur.nl |
| **FAOSTAT** | FAOSTAT Land Use Domain; Harmonized world soil; ESA CCI; MODIS 6 Burned area products | IPCC guidelines | IPCC (2006, Vol.4, p.10.33) - confidential Uncertainties in estimates of GHG emissions are due to uncertainties in emission factors and activity data. They may be related to, inter alia, natural variability, partitioning fractions, lack of spatial or temporal coverage, or spatial aggregation. | Agriculture total and subdomain specific GHG emissions are found for download at http://www.fao.org/faostat/en/#data/GT (last access: April 2022). |
| **CO$_2$ land: Top-down** | | | | |
| **CSR** **GCP ensemble** (CTE, CAMS, CarboScope) **EUROCOM** (PYVAR-CHIMERE, LUMIA, FLEXINVERT, CSR, CTE-Europe) **LUMIA** **CIF-CHIMERE** | Tier 3 top-down approach, prior information from fossil emissions, ocean fluxes, and biosphere-atmosphere exchange Spatial resolutions ranging from 1°x1° for certain regions to 4°x5°. EUROCOM uses more than 30 atmospheric stations. CSR uses four different settings (as described in Appendix A2) | Tier 3 top-down Inversion systems based on atmospheric transport models | **CSR** - Gaussian probability distribution function, where the error covariance matrix includes errors in prior fluxes, observations and transport model representations. **GCP**: the different methodologies, the land-use and land-cover data set, and the different processes represented trigger the uncertainties between models. a semi-quantitative measure of uncertainty for annual and decadal emissions as best value judgment = at least a 68 % chance (±1σ) **EUROCOM**: account for source of uncertainties via prior and model and observation error covariance matrices; assessment of the resulting uncertainties in fluxes based on spread **LUMIA**: The prior uncertainties are constructed using standard | Detailed gridded data can be obtained by contacting the data providers: **CSR**: Christoph Gerbig cgerbig@bgc-jena.mpg.de Saqr Munassar smunas@bgc-jena.mpg.de **GCP**: Pierre Friedlingstein P.Friedlingstein@exeter.ac.uk **EUROCOM**: Marko Scholze marko.scholze@nateko.lu.se Gregoire Broquet gregoire.broquet@lsce.ipsl.fr **LUMIA**: Guillaume Monteil guillaume.monteil@nateko.lu.se **CIF-CHIMERE**: Gregoire Broquet gbroquet@lsce.ipsl.fr |

| | | | deviations proportional to the sum of the absolute value of the hourly NEE aggregated in each weekly optimization interval (so, in essence, uncertainties are large when the daily cycle of NEE is large), spatial correlation lengths of 500 km (Gaussian) and temporal correlation lengths of 1 month (Exponential). | |
|---|---|---|---|---|



*Table B2: Comparison of the processes included in the inventories, bottom-up models and inversions.*

| Descri-ption | NGHGI | Global database | Process-based models | | | | DGVMs | | | Bookkeeping Models | | | Inversions[#] |
|---|---|---|---|---|---|---|---|---|---|---|---|---|---|
| | UNFCCC[a] | FAOSTAT[a] | ECOSSE | EPIC-IIASA | CBM | EFISCEN-Space | CABLE-POP | TRENDY V10 | ORCHIDEE | BLUE vGCP | BLUE vVERIFY | H&N | |
| Forest total | E | E | N | N | E | E | E | Acc. table A1 in GCB 2021 (Friedlingstein et al., 2022) | E | E[h] | E[h] | E[h] | |
| Split FL-FL / FL-X / X-FL | E | E | N | N | E | E/N/N | E | | E | E[h]/E/E | E[h]/E/E | E[h]/E/E | |
| Cropland total | E | N | E | E | N | N | I | | E | E[h] | E[h] | E[h] | |
| Split CL-CL / CL-X / X-CL | E | N | E | E/N/N | N | N | I | | E | N/E/E | N/E/E | N/E/E | |
| Grassland total | E | N | E | N | N | N | E | | E | E | E | E | |
| Split GL-GL / GL-X / X-GL | E | N | E | N | N | N | E | | E | N/E/E | N/E/E | N/E/E | |
| Peatland accounting | E | E | N | N | N | N | N | | N | N | N | N | |

| | | | | | | | | | | | | |
|---|---|---|---|---|---|---|---|---|---|---|---|---|
| CO₂ fertilization | I | I | N | E | N | N | E | Acc. table A1 in GCB 2021 (Friedlingstein et al., 2022) | E | N[i] | N[i] | N[i] |
| Climate induced impacts | I | I | N | E[f] | I[b] | I[c] | E | | E | N[i] | N[i] | N[i] |
| Natural disturbances (fires, insect, wind) | I | I | N | N | E | N | E | | N | N[i] | N[i] | N[i] |
| Soil Organic C dynamic | I | | E | E | E | E | E | | E | N | N | N |
| Lateral C transport (river) | N | N | N | N | N | N | N | | N | N | N | N |
| Flux from Harvested Wood Products | E | N | N | N | I | N[d] | E | Acc. table A1 in GCB 2021 (Friedlingstein et al., 2022) | E | E | E | E |
| Flux from Crop/Grass harvest | ? | N | E | E[e] | N | N | E | | E | I[i] | I[i] | I[i] |
| Biomass burning | E | E | E | N[g] | E | N | N | | N | E[j] | E[j] | E[j] |
| N fertilization (with N dep) | I | N | E | N | N | N | E | | N | N | N | N |
| Flux from drained organic soils | I | E | E | N | I | N | N | | I | E[j] | E[j] | E[j] |

Not included : **N,** Explicitly modeled : **E,** Implicitly modeled: **I,** Partly modeled : **P**

[a]UNFCCC and FAOSTAT are ensemble of country estimates calculated with specific methodology for each country, following some guidelines
[b]The climate effects can be estimated indirectly by CBM, using external additional input provided by other models
[c]EFISCEN Space: Increment is sensitive to weather, but average weather
[d]EFISCEN has only production in m³ but doesn't have a direct HWP module
[e]Crop yield and residue harvest from cropland (20 % of residues harvested in case of cereals, no residue harvest for other crops)
[f]EPIC-IIASA partly accounts for soil drought, i.e., plant growth limitation due to a lack of water in the soils. Heat stress and floods are not accounted for, though
[g]In principle, burning of crop residues on cropland can be explicitly simulated by EPIC-IIASA. However, not done for VERIFY as it is not a relevant scenario for the business as usual cropland management in Europe
[h]forest/cropland/grassland exist and have carbon stocks, but have carbon fluxes only through change to management. FL-FL includes all land-use induced effects (harvest slash and product decay, regrowth after agric abandonment and harvesting)
[i]implicit by using observation-based carbon densities that reflect harvest/climate/natural disturbances
[j]peat burning and peat drainage are not bookkeeping model output, but are added from various data sources during post processing
[*]According Table 2 in Monteil et al. (2020) and Table A3 in Friedlingstein et al. (2019)



[#]These categories are inputs to the inversions, not a result; the inversions adjust the total land-atmosphere C flux, regardless of what went into the prior, and the posterior flux cannot really be disaggregated into contributions from separate processes. In a sense, as long as a process is sufficiently significant to influence the $CO_2$ observations, it will have an impact on the inversion results

## Author contributions

MJM processed original data, made Fig. 1,3-10, A2, A3, A5, and edited the final manuscript; AMRP designed the initial research, led the discussions, wrote the initial draft of the paper and helped edit all the following versions; RMA made Fig. 2, A1, B3; BM provided the new UNFCCC gap-filled uncertainties and provided extensive support on questions related to NGHGIs; PP, VB, and MJM processed the original data submitted to the VERIFY portal; PP, PB, and MJM designed and are managing the web portal; GP provided Fig. B1 and B2; GP, RMA, FD, BM, and GG made detailed reviews; CQ made Fig. 11; SM made Fig. A4; PC, GB, PIP, MJ, RL, MK, JK, FC, OT, JP, RG, FNT, JB and GG gave detailed comments and advice on previous versions of the manuscript; all remaining co-authors provided data and commented on specific parts of the text related to their data sets.

## Competing interests

The authors declare that they have no conflict of interest.

## Acknowledgements

We thank Aurélie Paquirissamy, Géraud Moulas and all ARTTIC team, for the great managerial support offered during the VERIFY project. FAOSTAT statistics are produced and disseminated with the support of its member countries to the FAO regular budget. The views expressed in this publication are those of the author(s) and do not necessarily reflect the views or policies of FAO. We acknowledge the work of other members of the EDGAR group (Edwin Schaaf, Jos Olivier). We acknowledge Stephen Sitch and the authors of the DGVMs TRENDY v10 ensemble models for providing us with the data. We thank all the national forest inventories that have made their data available: Ireland (John Redmond), Norway (Rasmus Astrup), Sweden (Jonas Fridman), Poland (Andrzej Talarczyk), Germany (BMEL), The Netherlands (WUR & Stichting Probos), Belgium (Flanders: Leen Govaere), Luxembourg (Thierry Palgen), France (IGN), Spain (MAPA), Switzerland (Esther Thürig), Italy (CREA), Czech Republic (Emil Cienciala), Slovak Republic (Vladimír Šebeň). We thank all the NFI field crews for their hard work. Timo Vesala thanks ICOS-Finland, University of Helsinki. Ingrid T. Luijkx and Wouter Peters thank the HPC cluster Aether at the University of Bremen, financed by DFG within the scope of the Excellence Initiative. MJM and VB were granted access to the HPC resources of GENCI-TGCC under the allocation A0130106328.

## Financial support

This research has been supported by the European Commission, Horizon 2020 Framework Programme (VERIFY, grant no. 776810, for AB, AFC, AMRP, AP, CG, GB, GJM, GJN, GM, GP, HACDG, JB, LP, MJ, MJM, MK, MV,



PP, PR, PS, RG, RMA, SD).  MJM and GM also acknowledge funding from the European Union's Horizon 2020
research and innovation programme under Grant Agreement No. 958927 (CoCO2).  Philippe Ciais acknowledges
the support of European Research Council Synergy project SyG-2013-610028 IMBALANCE-P and from the ANR
CLand Convergence Institute. Ronny Lauerwald thanks the CLand Convergence Institute. Pierre Regnier
acknowledges the ESM 2025. Gert-Jan Nabuurs thanks the Dutch National Forest Inventory funded by the Ministry
Agriculture Nature Management and Food Quality. Guillaume Monteil's model computations were enabled by
resources provided by the Swedish National Infrastructure for Computing (SNIC) at NSC partially funded by the
Swedish Research Council through grant agreement no. 2018-05973.



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
