# Peer review of "The consolidated European synthesis of CO2 emissions and removals for 1 EU27 and UK: 1990-2020"

_Earth System Science Data, 2022_

## Referee Comment (RC1)

**General comments**

The authors attempted to clarify the emissions differences between datasets developed with different approaches. Overall, this paper is well written and would be helpful, particularly for those communities working closely with the datasets used in the paper. The dataset itself may not be something unique, instead, this paper could support the future GHG emissions inventory synthesis work in Europe and potentially other Annex I parties in the world.

However, possibly by the nature of this type of dataset, the text seems to be a bit lengthy and tedious, especially the result section. Although the authors described the details of the data sources and models used in this work, the dataset itself may not be easy to use/understand for users new to this field. And, the focus appeared to be more on LULCF rather than fossil fuel emissions (no economic sector breakdown), which looks a bit unfair in comparing uncertainty between anthropogenic emissions and land fluxes.

A list of jargon and abbreviations as an appendix, a brief description of targeted users, and potential data usage/applications might be helpful. Lastly, providing a list of the key take-home messages in the introduction or conclusion remarks may help the reader to grasp the paper's contents (and narrow down sections relevant to the readers).

**Specific comments**

l:148-163 It looks a bit controversial to see "A comprehensive investigation of detailed differences between all datasets is beyond the scope of this paper" in the intro and relatively detailed comparisons between datasets (especially LULUCF) in the result. Which is the actual focus?

l:390-395 Adding a specific figure number that support each statement (e.g., Fig1a) would be helpful.

l:413-415 (Figure 1 caption) Should this statement be replaced with Top-down or the graphic representation is wrong? The LULUC breakdown is not presented for Top-down in the plot.

l: 484-1024 Section 3.3 $CO_2$ land fluxes (3.3.1 through 3.3.5), about the Figures (two panel comparisons), how to interpret the differences between this work and Petrescu et al., (2021)? The difference sometimes very big.

l:552 Figure 3 is a bit hard to understand. Could this be a table instead?

l: 1025 This sub-section could be one independent section (i.e., section 3.4) as the content is more focused on uncertainty as a whole. Bringing this section before diving

into details for each components (fossil fuel emissions and land fluxes) could be an option.

l:1169 It might be true that ecosystem flux estimations are more complex than fossil fuel emission estimations but should note that this is only the case when fuel statistics and accurate emission factors are available (as discussed in section 2.2 first paragraph).

**Technical corrections**
- l:307 Correct the font of "The nations"
- l:591 Bold the sub-section title "forest land" to be consistent with the following section.
- Figure 3,4,7,8,9 have two images (one is faded out, the other is clear). Why tp panels are faded?
- Figure 11 caption is center-aligned.
- [l:1025] "3.3.6. Uncertainties in top-down and bottom-up estimates" Check the section format/font.

---

## Referee Comment (RC2)

Review of McGrath et al., 2023, for publication in ESSD "The consolidated European synthesis of [1] CO₂ emissions and removals for EU27 and UK: 1990-2020" by John Miller

First, apologies to the authors for the time it's taken me to review this. Also, I want to acknowledge that I was only able to review the paper through page 66.

General comments are below and specific line by line comments are embedded as notes in the accompanying .pdf.

**General comments:**

This is a hugely impressive (and huge) piece of work. Generally, the writing, graphing, and interpretation are at a very high level. I hope that the acceptance of the general and specific comments below and embedded in attached .pdf will serve to further improve the paper.

A comment to the editor(s), mainly: As someone (like most scientists) who thinks of data as measurements/observations and thus distinct from model output (forward or inverse, bottom-up or top-down) and inventories, I'm not sure I agree that the best venue for these types of papers is ESSD.

Be that as it may, though, since this paper is being submitted to ESSD, I think the bar for transparency and accessibility for all the "data" (obs, models, inventories) should be set very high and represent best (and not necessarily common) practice. In particular, before this paper is accepted for publication, I think there needs to be clear and "one-step" links to freely available, time and space-resolved model output, and inputs, for both the terrestrial biosphere and atmospheric models as well as inventories (realizing that inventories are not often gridded). Along these lines, unless a special arrangement can be made allowing free accessibility to IEA data, the EIA-based emissions need to be removed from the paper. Detailed codes and/or methods also need to be made available to the reader. I don't simply mean that a reference is provided to, for example, an ORCHIDEE methods paper, but rather that the ORCHIDEE (and variants thereof) code(s) need to be made accessible via GitHub or similarly accessible platforms. While I realize this will take substantial time and effort, I think this is the required corollary of thinking of these sources of information as "data" and thus publishing in ESSD. More generally, I'm a strong believer in maximum accessibility and transparency and feel that it's important that our community continues to move more and more in this direction. Ultimately, this is an important aspect of democratizing science: a reader should not have to know someone or "be in the club" to have unfettered access to the details behind published results (e.g., codes, inputs and gridded fluxes).

Some technical, yet still general comments:

1. In the presentation of an ensemble of fossil $CO_2$ estimates, I'm interested to know more about their independence. To what extent are the different estimates using similar activity data and/or emissions factors? This question bears directly on the use of an ensemble as a proxy for uncertainty estimation which is critical when trying to understand the extent to which bottom-up and top-down estimates are different. The same comment applies to inventories (and to a lesser extent models) of terrestrial biosphere fluxes. There is brief mention of this issue around line 1360 in the Appendix, but this needs to be elevated to the main text and expanded.

2. The definition of managed land is very important for comparing top-down and bottom-up TBM fluxes with national inventories, as is the correction of top-down fluxes for lateral carbon transport. I think it would help to explain early on how both of these issues need to be accounted for when comparing TD and BU (e.g., TBM) estimates to national inventories. I realize that in the case of managed land definitions, these are not necessarily easy to obtain, but I would expect that these maps are available for most if not all of the countries in the present study. Moreover, in the case of the US, at least, > 99% of the area of the 48 contiguous states are considered managed, making this a minor issue. Even a rough initial description of how much a complication this is in TD/BU comparison would be helpful. Regarding lateral fluxes, in addition to early presentation of the necessity for correcting for these fluxes to arrive at stock/reservoir changes (at least in the case of using NEE estimates), discussion of the uncertainty of lateral flux uncertainties would be helpful.

3. Relatedly, there is no mention of masks of managed lands being applied to TD or BU fluxes when comparing to the inventories. Can you confirm this and also discuss the possible uncertainties associated with not doing this?

4. Use of the word "natural" (line 108 and elsewhere). There are several issues here. First, in the carbon cycle community more generally, "natural" often refers to emissions and removal by oceans and the terrestrial biosphere. In the context of national inventories, though, terrestrial ecosystem sources and sinks over managed land are counted as anthropogenic, so this usage needs be clarified. Second, and more generally, even in unmanaged lands (and the oceans), using "natural" in the post-industrial era is not appropriate. Given that we exist in a world strongly affected by anthropogenic impacts on CO2, temperature and precipitation (and also O3, N deposition, etc.), even so-called "natural" terrestrial biosphere and oceanic fluxes have a strong human component, and at this point we need to be using precise language in the scientific literature that reflects this. Finally, even if there were no anthropogenic component to NEE or net ocean fluxes, the typical dichotomy is that between "natural" and "anthropogenic" fluxes. But what does it say about our worldview? Is *homo sapiens*, and even fossil fuels, not part of the natural world? They're certainly not supernatural. One way to distinguish fluxes might be as terrestrial biospheric, oceanic, and fossil, with perhaps some distinction for "TB" fluxes over managed and unmanaged land.

5. Finally, I think a table or glossary of terms would useful: e.g., class, sector, category; acronyms/abbreviations like NBE, LUC, LUCC, LULUCF, etc.

For **specific line by line comments** in the attached .pdf, yellow highlights generally represent scientific/technical comments and blue highlights represent language-related comments.

[revised manuscript text omitted]

---

## Author Comment (AC1)

| | | Original manuscript | | | | Revised manuscript | | | |
|---|---|---|---|---|---|---|---|---|---|
| | | Start | Stop | Total | # Fig | Start | Stop | Total | # Fig |
| Abstract | | 63 | 104 | 41 | 0 | 67 | 110 | 43 | 0 |
| Introduction | | 107 | 188 | 81 | 0 | 112 | 218 | 106 | 0 |
| Methods | | | | | | | | | |
| | Overview | 191 | 239 | 48 | 0 | 221 | 271 | 50 | 0 |
| | NGHGI | 242 | 252 | 11 | 0 | 274 | 287 | 13 | 0 |
| | Fossil | 255 | 284 | 29 | 0 | 290 | 326 | 36 | 0 |
| | Land | 290 | 370 | 80 | 0 | 332 | 425 | 93 | 0 |
| | Independence | - | - | - | - | 428 | 458 | 30 | 0 |
| Results | | | | | | | | | |
| | Synthesis | 373 | 431 | 58 | 1 | 462 | 537 | 75 | 1 |
| | Fossil | 434 | 480 | 46 | 1 | 540 | 589 | 49 | 1 |
| | Land | 485 | 1149 | 664 | 9 | 593 | 1061 | **468** | **4** |
| | --Intro | 485 | 522 | 37 | 0 | 593 | 637 | 44 | 0 |
| | --Decade | 526 | 577 | 51 | 1 | - | - | **0** | **0** |
| | --Overview of BU | 580 | 589 | 9 | 0 | 641 | 650 | 9 | 0 |
| | --FL | 592 | 674 | 82 | 2 | 654 | 721 | 67 | **1** |
| | --CL | 678 | 753 | 75 | 1 | 725 | 817 | **92** | **1** |
| | --GL | 756 | 807 | 51 | 1 | | | | |
| | --BU | 811 | 921 | 110 | 1 | 821 | 978 | **157** | **1** |
| | --TD | 925 | 1024 | 99 | 1 | | | | |
| | --Uncert | 1026 | 1149 | 123 | 2 | 982 | 1061 | 59 | **1** |
| Conclusion | | 1161 | 1288 | 127 | 0 | 1079 | 1212 | 133 | 0 |
| Total | | 63 | 1288 | 1225 | 11 | 67 | 1212 | **1145** | **6** |

---

## Author Comment (AC4)

**John Miller**

We are happy that the reviewer finds our work to be "hugely impressive", and that he has taken the time to go through the main text and make significant comments. It's clear the reviewer has made an effort to understand the entire text, which helps improve the overall readability, accuracy, and consistency of our manuscript. We detail our responses below. We first respond to those comments requiring substantial amounts of work/reflection, which are open to interpretation, or to which we hold an alternative view. For straightforward comments (33 out of around 170 comments), we have made the proposed changes and noted them at the end of our response. Some of the comments in the text are covered by the general reviewer comments (1-5), and in those cases we group them together.

We apologize for any comments we may have missed.

**General comment about transparency and accessibility**: we have partially addressed this with our response to Alex Vermeulen. In summary, we generally agree with John Miller's aspirational principles, and have already taken steps above and beyond a traditional paper in this regard, including extensive citations (180 references) and lengthy descriptions in the appendices, often summarizing how results here differ from those given by the same model in citations. In addition, we include representatives of these models as co-authors to help increase the accuracy of what we report, along with contact information as possible in Table C1 (L2252).

We emphasize that the request is, truly, aspirational: open-source is no guarantee that results are reproducible by others, despite that the developers themselves often have extensive testing in place to ensure numerical reproducibility in their own development environments. Models (including land-surface models like ORCHIDEE) often require supercomputing clusters (with unique computational environments, e.g., specific libraries) and highly specialized knowledge to install, compile, and run, even if all input and forcing data is provided. Similarly, to perhaps a lesser extent, it's not obvious that someone lacking prior experience with the product or close ties to the community will be able to reproduce the complex set of calibration and correction procedures required to generate atmospheric concentration measurements from raw data, even with the scripts. We say this not to discourage our community from moving in this direction, but more-so to keep expectations realistic for those both inside and outside of science.

We would even argue that extensive intermodel comparison projects, such as what we have done here, are more useful for establishing general reproducibility in complex models as they harmonize inputs and outputs, enabling examination of differences between models to the greatest extent possible. They also foster communication between groups, which leads to deeper process understanding and progress. Until the models either show excellent agreement with each other under these circumstances or the differences are precisely explained, the ability of non-developers to reproduce any individual model run to high precision appears to be of secondary importance. We, of course, defer to any journal-specific requirements in this regard which are consistently applied to all manuscripts under consideration.

This being said, we recognize that there is always room for improvement, and we have taken additional steps in response to John Miller's comment. Table C1 in the Appendix summarizes information on the datasets used in this work, separated into national inventories, fossil, land (top-down), and land (bottom-up) approaches, including references and names of contact people as possible. We have made this table more prominent by referring to it in the Data Available section, as well as by adding "An overview of the datasets, including contact information, is provided in Table C1." at two places: L252, and the caption of Table 2. We note that the Global Carbon Budget, a very high-profile work, faces similar challenges to what we face here, and to our knowledge has so far been unable to propose a better solution in regards to transparency and access. We do believe our community is making progress, however.

Finally, many of the gridded datasets at monthly temporal and various spatial resolutions are freely available from the VERIFY project website. We have added these links as applicable in Table C1. While they do require

a username and password to access, such credentials are offered free of charge after submitting an email address for confirmation. Other gridded products, such as TRENDY, the GCP inversions, and the EUROCOM regional inversions, are available from their respective sources. We do not duplicate their efforts here, in part because such projects are extremely valuable to the community and may need certain statistics to ensure continued funding. We understand that the use of the VERIFY website does not fully comply with an open-data one-click policy, but we also hope it is recognized that these efforts exceed current standard practices, and represent a significant effort on our part. We have added the following text summarizing this to the Data Availability section:

L1070: "2022) as well as a number of gridded data files submitted to the VERIFY project listed in Table C1. Access to the data files requires free registration to obtain a username and password. Alternatively, interested users are invited to contact the persons listed in Table C1 to request gridded datafiles directly from them. We do not provide access to data already made freely available elsewhere, as we prefer users use mechanisms put in place by the original providers so that they are able to ensure their continued funding for their work."

Some additional comments related to this are:

*[4. Data Availability] → Given that this is a data journal, primarily, I think pretty much everything used in this study needs to be made publicly available: gridded inversion fluxes, gridded BU model fluxes, the lateral fluxes and on and on. In fact best practice would also be to make codes for bookkeeping, inversion and bottom-up models freely available, as well as all of their inputs. I found it somewhat shocking actually that in a ~ 100 page paper for ESSD only six lines were devoted to data availability.*

*l1153 - [This excludes $CO_2$ fossil data for the IEA, which is subject to license restrictions.] → As alluded to in the general comments, the inclusion of the IEA fossil emissions certainly doesn't conform to the guidelines in Carlson and Oda. More generally, and especially for an open access data journal, I don't support the inclusion of paywalled data. It is my strong recommendation that these data be excluded from this analysis unless the authors can secure public access to this version of the IEA data (and not just coarsened summaries).*

We understand the reviewer's concerns about the availability of the IEA data, in particular for a paper published in ESSD. As this dataset is widely used, we feel that its inclusion helps place it in context for a wider community. We also note that none of our conclusions depend on the IEA data.

We now address general comments 1-5, including related specific comments in the text.

***General comment (1)*** *(including comments at lines 97, 433): In the presentation of an ensemble of fossil CO2estimates, I'm interested to know more about their independence. To what extent are the different estimates using similar activity data and/or emissions factors? This question bears directly on the use of an ensemble as a proxy for uncertainty estimation which is critical when trying to understand the extent to which bottom-up and top-down estimates are different. The same comment applies to inventories (and to a lesser extent models) of terrestrial biosphere fluxes. There is brief mention of this issue around line 1360 in the Appendix, but this needs to be elevated to the main text and expanded.*

*l97 - [Overall, for both CO2 fossil and net CO2 land fluxes, we find current independent approaches] → how independent are the fossil fluxes in terms of the source data that is used?*

*l433 - the independence of the different fossil CO2 products is not discussed or evaluated. If Andrew 2020 did this already, that's great, but please summarize their conclusions about independence here.*

*l627 - [Figure 5 presents $CO_2$ land estimates for total Forest land (both remain and convert classes, "FL"). For the total Forest land, the results were simulated with an ecosystem model (ORCHIDEE) and a global dataset (FAOSTAT) as it is not possible for these two approaches to separate out the "remain" and "convert" land use category.] → does FAOSTAT also use national forest inventory data for European countries? Either way, this would be good to note to establish the degree of independence with the other methods.*

*l1360 - [However, the use of common input data restricts the independence between the datasets and, from a verification perspective, may limit the conclusions drawn from the comparisons.] → This is a key issue to me and should be presented and further discussed in the main text.*

Following these reviewer comments, we have included a summary of the discussion from Andrew (2020) in a new subsection at Section 2.4, with one paragraph on fossil bottom-up sources, and one paragraph covering vegetation bottom-up inventories and models. Briefly, the energy datasets are not fully independent, as they largely rely on data reported by national agencies. We recognize that this does limit estimation of uncertainties and thus comparison with other datasets. We have re-iterated this caution at various places throughout the manuscript:

    L298: "The datasets are also not fully independent, as discussed in Sec. 2.4"

    L490: "As datasets are not fully independent, the uncertainties in Fig. 1 need to be interpreted with caution."

    L1086: ", although the datasets are not fully independent, which complicates uncertainty estimation."

The following text comprises the new Section 2.4.

[revised manuscript text omitted]

*General comment (2) (including comments at lines 193, 248, 509): The definition of managed land is very important for comparing top-down and bottom-up TBM fluxes with national inventories, as is the correction of top-down fluxes for lateral carbon transport. I think it would help to explain early on how both of these issues need to be accounted for when comparing TD and BU (e.g., TBM) estimates to national inventories. I realize that in the case of managed land definitions, these are not necessarily easy to obtain, but I would expect that these maps are available for most if not all of the countries in the present study. Moreover, in the case of the US, at least, > 99% of the area of the 48 contiguous states are considered managed, making this a minor issue. Even a rough initial description of how much a complication this is in TD/BU comparison would be helpful. Regarding lateral fluxes, in addition to early presentation of the necessity for correcting for these fluxes to arrive at stock/reservoir changes (at least in the case of using NEE estimates), discussion of the uncertainty of lateral flux uncertainties would be helpful.*

*l193 - I think since the concept of managed and unmanaged lands are so important to comparing (or not being able to compare) national inventory fluxes or stock changes with those from (e.g.) terrestrial biosphere models or inverse models, by this point in the paper, it would be useful to have discussed 'managed lands' by this point.*

*l248 - [For the land-based sector, the land management proxy] --> it might be good to explain this a bit and not just reference it.*

*l509 - [Additional CO2 fluxes occur on unmanaged land, but these fluxes are very small in Europe] --> is it that the fluxes (mole/m2/s) are small or that the area of unmanaged land is very small, or both? It would be good to have established the overall UK+EU27 managed v. unmanaged land area fractions well ahead of this instead of in the following sentence. To me this is important background information regarding the comparability of top-down v. bottom-up for the intro.*

We originally included a discussion of managed lands at l248 and l509. We have added new text at L158, before the reviewer's comment at l193, with a rewording of what is found at l248: "For the land-based sector, Member States are only required to report terrestrial biospheric fluxes from managed lands, instead of distinguishing between direct and indirect human-induced and natural effects on carbon fluxes for all ecosystems (Grassi et al., 2018a, 2021). This "managed land proxy" avoids having to quantify, for example, increased carbon uptake in remote Forest land due to reactive nitrogen emissions from both natural soils and human-applied synthetic fertilizers." As this hopefully clarifies the concept for the reader at an earlier stage, we have left the original text at l248, which now serves as a reminder. The paragraph became longer with this addition, so we have added a break. In addition, we have added "managed land proxy" to the glossary.

Spatially-explicit maps of managed lands are, unfortunately, not currently published by most countries, even within the European Union and United Kingdom. We have added the following text at L283: "Spatially-explicit maps of managed lands are not currently available, even for the relatively data-rich region of the European Union and United Kingdom. However, most of the European Union is classified by the Member States as managed land; current estimates from available country-aggregated data indicates only 5 % of land in the EU is unmanaged, including some Forest land, Grassland, and Wetlands." For l509, we have added the clarification, "but the fraction of unmanaged land in the European Union is only around 5 % and divided between Forest land, Grassland, and Wetlands." More details on the countries with unmanaged areas are included in Section 3.3 to which small modifications have been made consistent with our capitalization scheme (e.g., changing "forest areas" to "Forest land").

We originally included a discussion of the carbon emitted from lateral fluxes (crop and wood trade; river transport) at lines l933-943. Specific estimates of uncertainty are not yet available for the lateral flux emissions. The source of data comes from FAO and IEA, along with some conversion factors and hypotheses. Based on expert knowledge, we estimate 50% uncertainty (i.e., significant, but not 100%), which results in a spread of around 70 Tg C yr$^{-1}$. This is comparable to the spread of the regional inversions in Fig. 5, though still lower than that of the global inversions. We have added the following text at L925:

"Uncertainties for net emissions of $CO_2$ due to lateral transport of carbon are not yet available. However, FAO and IEA statistics form the basis of calculated fluxes due to wood and crop trade. FAO estimates an uncertainty of 50 % on carbon emissions and removals from forested land (Tubiello et al., 2021). Even if uncertainties in trade fluxes are not available, 50% therefore works as a first order approximation given the similarities between the two fluxes (i.e., a well-tracked value multiplied by an uncertain emission factor). Uncertainties in net carbon uptake by rivers and lakes are estimated to also be on the order of 50 % due to the fact that these fluxes can only be calculated based on budget closure including estimates of river exports to the coast, emissions of carbon from the water surface to the atmosphere, and burial of carbon in aquatic sediments (Battin et al., 2023). Combined, this results in an uncertainty of around 70 Tg C yr$^{-1}$ for the lateral fluxes, which is on the same order as the ensemble spread for the regional inversions as shown in Fig 5, though still lower than that of the global inversions."

We have added the following reference:

Battin, T. J., Lauerwald, R., Bernhardt, E. S., Bertuzzo, E., Gener, L. G., Hall Jr, R. O., Hotchkiss, E. R., Maavara, T., Pavelsky, T. M., Ran, L., Raymond, P., Rosentreter, J. A., and Regnier, P.: River ecosystem metabolism and carbon biogeochemistry in a changing world, Nature, 613, 449-459, https://doi.org/10.1038/s41586-022-05500-8, 2023.

***General comment (3)*** *(including comment at Section 3.3.5): Relatedly, there is no mention of masks of managed lands being applied to TD or BU fluxes when comparing to the inventories. Can you confirm this and also discuss the possible uncertainties associated with not doing this?*

*[3.3.5. Comparison of atmospheric inversions with NGHGI CO2 estimates] → Sorry, I may have missed this, but were managed land masks applied to the TD results before comparison with the inventories? Actually, the same question applies to the BU models.*

The reviewer is correct: we did not apply any managed-land mask to inventories, unlike, for example, the work of Deng et al. (2022). The reasons for this are primarily: 1) the uncertainty associated with these products, and 2) the fact that the European Union and United Kingdom are largely managed. For example, in the work of Deng et al. (2022), the (un)managed land map is created by looking solely at intact forest landscape maps. This may be an acceptable assumption at global scale, but as noted above, unmanaged land in the European Union is small, and divided between Forest land, Grassland, and Wetlands. Given the high uncertainties associated with terrestrial biospheric flux datasets in this work (Figs. 4-10); the overall small amount of unmanaged land in the European Union (around 5%); and the lack of data to estimate unmanaged Grassland and Wetlands, we did not attempt to separate managed and unmanaged land. We have added the following text at L901 to clarify this:

"Notice that unlike other works (e.g., Deng et al., 2022), we have not applied a managed-land mask to the inversions or bottom-up models in order to be compatible with the managed land proxy in the NGHGIs. The reasons for this are two-fold. One, most of the land in the European Union is managed, as noted above. Second, no such mask currently exists, even for the relatively data-rich EU. A managed land mask created solely based on non-intact forests (e.g., Deng et al., 2022) neglects that Grassland and Wetlands contribute significantly to unmanaged areas in the EU. Including fluxes from the 5 % of unmanaged land in the EU is unlikely to change any conclusions in this work given the uncertainties in the LULUCF methods presented here. As soon as a reasonably accurate managed land mask is available, however, it should be used."

***General comment (4)*** *(including comments at footnote 4): Use of the word "natural" (line 108 and elsewhere). There are several issues here. First, in the carbon cycle community more generally, "natural" often refers to emissions and removal by oceans and the terrestrial biosphere. In the context of national inventories, though, terrestrial ecosystem sources and sinks over managed land are counted as anthropogenic, so this usage needs be clarified. Second, and more generally, even in unmanaged lands (and the oceans), using "natural" in the post-industrial era is not appropriate. Given that we exist in a world strongly affected by anthropogenic impacts on CO2, temperature and precipitation (and also O3, N deposition, etc.), even so-called "natural" terrestrial biosphere and oceanic fluxes have a strong human component, and at this point we need to be using precise language in the scientific literature that reflects this. Finally, even if there were no anthropogenic component to NEE or net ocean fluxes, the typical dichotomy is that between "natural" and "anthropogenic" fluxes. But what does it say about our worldview? Is homo sapiens, and even fossil fuels, not part of the natural world? They're certainly not supernatural. One way to distinguish fluxes might be as terrestrial biospheric, oceanic, and fossil, with perhaps some distinction for "TB" fluxes over managed and unmanaged land.*

*Footnote 4: [As this comparison is central to our work, we decided that "land" as defined by the IPCC was a good compromise.] --> this is also not ideal because the vast majority of fossil emissions are land based.*

The reviewer raises a long-standing debate in the community around nomenclature, which we touched on briefly in footnote 4, explaining our compromise for using "land". As a solution, the reviewer suggests "terrestrial biospheric", and distinguishing between managed and unmanaged land. Indeed, the distinction between managed and unmanaged land is critically important in properly comparing inventories with other model results, but not straightforward, given the lack of accurate spatially-explicit maps noted in Question 3. We have replaced multiple instances of variations on the word "natural" in the text with "terrestrial biospheric" when referring to net carbon fluxes, distinguishing between "managed" and "unmanaged" whenever possible. On the other hand, we left instances of "natural disturbances", "natural vegetation", "natural processes", and similar phrases when it seems to still make sense. For example, at L641, we say "total net $CO_2$ land emissions...induced by both LULUCF and natural processes" to make it clear that all emissions are considered regardless of source. Otherwise, we fear that the reader may be too focused on either anthropogenic or terrestrial biospheric emissions and interpret "total" to mean simply the one they have in mind. In addition, some disturbances in the terrestrial biosphere (e.g., fire) can occur both with and without human intervention, and we think "natural" is still suitable in this case.

While we agree that humans are part of nature, identification of human influences is critical to GHG accounting under current international agreements, and thus the distinction can still be useful.

We have also added a sentence to footnote 4: "However, we avoid the word "natural" as much as possible, under the assumption that almost all terrestrial ecosystems are significantly impacted by humans in the current era."

***General comment (5)***: *Finally, I think a table or glossary of terms would useful: e.g., class, sector, category; acronyms/abbreviations like NBE, LUC, LUCC, LULUCF, etc.*

In response to this, and a comment from the anonymous reviewer, we have added a glossary in the Appendix and referenced it in the main text, harmonizing the use of these words throughout the text.

Other comments – major

Here we address other comments which we found to be "major" in terms of work and/or thought required. Note that some comments are no longer applicable as various text has been changed/removed in response to comments by other reviewers (e.g., l537, Figure 3).

*l100 - [We conclude that $CO_2$ emissions from fossil sources have decreased over the past 30 years in the EU27+UK, while large uncertainties on net uptake of $CO_2$ by the land surface prevent trend identification] → from an atmospheric perspective, anyway, uncertainties seem small. "prevent trend identification" can be reframed more meaningfully to say, "land fluxes are relatively stable: we can rule out positive or negative trends larger (smaller) than x (y)."*

We have run a Monte Carlo analysis on the NGHGI $CO_2$ LULUCF data, using simple linear regression and including the uncertainties of each point by using the provided 95% confidence interval to define a normal distribution (i.e., the 95% CI is equal to $2\sigma$, where $\sigma$ is the width of the distribution). Sampling 1000 realizations of the timeseries for the period 1990-2019 and fitting a normal distribution to the resulting slopes gives -0.27 +/- 0.34 Tg C $yr^{-2}$, where the uncertainty indicates the 95% confidence interval. We have thus changed the text at L104 to be: "while land fluxes are relatively stable: positive or negative trends larger (smaller) than 0.07 (-0.61) Tg C $yr^{-2}$ can be ruled out for the NGHGI".

*l330 - [The prior anthropogenic emissions provided for all regional inversions reported here (i.e., EUROCOM, EUROCOM drought 2018, VERIFY CSR, VERIFY CIF-CHIMERE, and VERIFY LUMIA) are all based on EDGAR v4.3, BP statistics, and TNO datasets by generating spatial and temporal distributions through the COFFEE approach (Steinbach et al., 2011). Small differences exist between exact versions used by the different groups] --> quantify this... 'differences of less than xx % (yy TgC/yr) exist based on [land masks, regridding, etc.]...'*

*l335 - [Overall, differences in the prior anthropogenic emissions are not expected to explain the large differences seen between the different regional biogenic inversions nor between the regional and global biogenic inversions, but efforts should be continued to harmonize them to the greatest extent possible in future intercomparisons.] --> this would be easier to understand if 'small differences' is defined above. For example if the range of differences was not larger than 10 TgC/yr for Europe, then (as a good first guess, assuming reasonable closed constraint of Europe's total atmospheric CO2 budget), the terrestrial biosphere differences would be of the same order, if not actually closer to 10 TgC/yr.*

Our sentence regarding the anthropogenic emissions used by the different regional inversions ("Small differences exist between exact versions used by the different groups") was misleading for two reasons:
    - differences occur for the years 2019 and 2020 only; they correspond to two versions of the anthropogenic emissions product: the second version was generated to better fit with the changes in the emission temporal profiles during the Covid-19 crisis and the anomalies between 2019 and 2020
    - these differences correspond to the temporal variations of the emissions within the year, but not to the annual budgets, which are identical at the mapping grid to country scales in both versions of the emission products

Furthermore, in theory it would be possible to do a rough analysis at the monthly to hourly scales of the priors and generate a statement like the reviewer suggests above. However, it's not clear that the end result would be that useful. In particular, one of the assumptions that the reviewer makes in l335 is that the carbon budget of Europe is closed in the regional inversions, which is not the case (regional budgets are not formally closed in global inversions, either, as the methods can "borrow" to some extent from other regions). Other factors are likely to play a bigger role in the results (e.g., transport models, boundary conditions, uncertainty settings). While a quantification of such impacts is certainly interesting, it is well beyond the scope of this paper. We have therefore added the following text to clarify this at L379: " Differences in fossil fuel emissions for the regional inversions only exist for the years 2019 and 2020, and they only concern the temporal variation within the year, not the annual totals per pixel (or country). Therefore, differences in the prior anthropogenic emissions are not expected..."

*l375 - [In 2019, the UNFCCC NGHGI (2021) net CO2 flux estimates for EU27+UK, accounted for 3.01 Gt CO2 from all sectors (including LULUCF) and 3.28 Gt CO2 excluding LULUCF (Fig. B1), corresponding to a net sink of LULUCF of -0.27 Å} 0.11 Gt CO2.] -> is this a std. dev. of several estimates meant to represent an uncertainty? Or something else? Either way, please define/explain. Sorry if I missed this earlier. Also, please report an analogous +/- for the total and/or fossil emissions.*

The uncertainty in the LULUCF values is the 95% confidence interval propagated from the uncertainties of the Member States using gap-filling methods as noted in Appendix A. We have added the following text, including a modification of units and paying attention to significant figures in response to other comments at L462: "In 2019, the UNFCCC NGHGI (2021) net $CO_2$ flux estimates for EU27+UK accounted for 820 Tg C from all sectors (including LULUCF) and $900 \pm 10$ Tg C excluding LULUCF (Fig. B1), corresponding to a net sink of LULUCF of $-74 \pm 30$ Tg C, where the uncertainties are 95 % CI calculated in accordance with the gap-filling methods of Appendix A2 and propagated to the sector level through Gaussian quadrature."

*l380 - [The NGHGI shows minimal inter-annual variability (largely due to methodology),] -> First, please clarify here if you are referring to LULUCF IAV. Lack of net/total flux IAV is more a function of the fact that fossil emissions have little IAV and dominate the total/net. Second, if talking about LULUCF IAV, please explain what aspect of the methodology results in this (in the US forest surveys at least, stock changes are smoothed due to infrequent plot revisits. maybe the same here?). At least include a citation or refer to the section below where this might be discussed.*

We have clarified in the text that we are referring to LULUCF IAV by rewriting L470 as follows: "The NGHGI shows minimal interannual variability in the LULUCF sector (Fig. B2), largely due to methodology. For example, emissions and removals from Forest land are typically based on forest statistics and surveys that are only completed every 5-10 years (see, for example, the national inventory reports and references cited therein of France, Germany, and Sweden). The largest contributors to IAV in the EU NGHGI forestry fluxes are fires and windstorms (EU NIR, 2021). Consequently, the 2019 values are indicative of longer-term averages."

*l392 - [Uncertainties are much larger in the LULUCF estimates than in the fossil CO2 estimates (both for total LULUCF and for individual components of FL, CL, and GL);] → were these uncertainties calculated in the same ways? i.e. one by ensemble spread and the other by 'internal' random error?*

This is an observation that holds regardless if one compares fossil and LULUCF fluxes from the NGHGI ('internal' random error) or fossil and LULUCF fluxes from bottom-up models (ensemble spread). We have clarified this by using the following text at L485 (overwriting a comment by another reviewer): "Uncertainties are much larger in the LULUCF estimates than in the fossil $CO_2$ estimates, regardless if one represents uncertainty by internal random error (i.e., the NGHGI totals in Fig. 1a, and the sub-sector LULUCF fluxes in Fig. 1d) or ensemble spread (i.e., bottom-up models in Fig. 1b, and the sub-sector LULUCF fluxes in Fig. 1e);".

*l399 - [Similarly, carbon uptake by the land surface has remained more or less stable over the past three decades, with the vast majority of that occurring in forests.] --> This is a much better way of describing the results than in the abstract. However, I still recommend a quick statistical analysis to quantify 'more or less stable' — e.g., something like 'At the 95% CI, we can rule out trends since 1990 greater than x TgC/yr or less than y TgC/yr.'*

Using the same results from the comment at l100, we have modified the text to be at L495: "By applying a Monte Carlo analysis and taking each point to be normally distributed around the mean with a width 2σ equal to the given 95 % CI, we realized 1000 linear regressions of the NGHGI across the 1990-2019 period. From this, we fit a normal distribution to the slopes, and can rule out trends greater than 0.07 or less than -0.61 Tg C yr$^{-2}$ with 95 % confidence. Therefore, any trend over these 30 years is likely less than 1 % of the net carbon uptake, with the vast majority of that occurring in forests." We note that both this comment and that at l100 technically referred to the whole group of approaches. We have signaled out the NGHGI as an illustrative example.

*l424 - [This was done to weigh more heavily the regional approaches under the assumption that higher resolution simulations and more region-specific input data will lead to more accurate results. While the latter assumption appears reasonable, the first assumption can be disputed.] --> What would be the basis of disputing this? Given the very high spatial heterogeneity of the European landscape, it seems reasonable to think that high resolution would lead to better fluxes.*

Higher resolution simulations will indeed produce more accurate results assuming accuracy of the data and model processes also increases with resolution, which is not necessarily the case: a model parameterized at coarse resolution will be exposed to forcing values outside the parameterization range as the resolution increases, which may result in undesirable results due to nonlinear behavior. A simple illustration of this comes from temporal resolution: the range of monthly mean temperatures for a pixel is much narrower than the range of hourly temperatures. In terms of processes, constant tree-mortality may be an appropriate assumption at the scale of thousands of square kilometers, while abrupt mortality may be necessary to describe dynamics at the stand level. We have added text to help clarify this at L528: "...can be disputed. Finer resolution leads to models being exposed to values of input variables (e.g., temperature, rainfall) outside the parameterization range, which may result in unexpected behavior. Process representation can also change with spatial scale. Constant tree

mortality, for example, is often used in models at coarse resolution, while abrupt tree mortality (stand-replacing disturbances) may better describe stand-level dynamics."

*l499 - [Forest aging affects the net sink both through the forest growth (net increment) - which tends to level off or decline after a certain age - and the harvest, because a greater area of forest reaches forest maturity] --> Sorry if this is naive, but does forest aging affect mortality in undisturbed forest stands as well? That is, I would suspect that as median stand age increases, mortality likelihood also increases.*

Mortality has been observed to increase in relatively undisturbed stands in the Pacific Northwest United States ("undisturbed" in Gray et al. is taken to mean "no disturbance between measurements"). On the other hand, no correlation with stand age was found with recent increases in canopy mortality in temperate forests in Europe (Senf et al.).

We have added the following sentence at the end of this paragraph (L615): " The impacts of aging on mortality, another process which affects the net sink through reduced production and increased respiration, are less clear (e.g., Gray et al., 2016; Senf et al., 2018)."

We have added both references to the Reference section.

Gray, A. N., Whittier, T. R., and Harmon, M. E.: Carbon stocks and accumulation rates in Pacific Northwest forests: role of stand age, plant community, and productivity, Ecosphere, 7, e01224, https://doi.org/10.1002/ecs2.1224, 2016.

Senf, C., Pflugmacher, D., Zhiqiang, Y., Sebald, J., Knorn, J., Neumann, M., Hostert, P., and Seidl, R.: Canopy mortality has doubled in Europe's temperate forests over the last three decades, *Nat Commun,* 9, 4978. https://doi.org/10.1038/s41467-018-07539-6, 2018.

*l512 - [Grassland]* ➔ *lower case*

Land-use categories are written here as they are typically written in the IPCC guidelines and the National Inventory reports submitted under the UNFCCC. To emphasize that we are talking about grasslands as defined and reported by the countries (which vary from country to country), and not some generic scientific definition of "grasslands", we have written this and other Land-use categories with a capital letter at the start. We have modified footnote 7 in order to clarify for the reader.

*l618 - [The UNFCCC NGHGI uncertainty of CO 2 estimates for FL-FL across the EU27+UK, computed with the error propagation method (95 % confidence interval) (IPCC, 2006), ranges between 34 % - 55 % when analyzed at the country level for all years, as it varies as a function of the component fluxes (EU NIR, 2019).]* → *recommend stating uncertainty in TgC/yr not %. For values that can easily cross zero this is much better. if the value for a year were -10 +/- 50 TgC what would the fractional uncertainty mean?*

*Figure 7 – [Grassland plot] Uncertainties for inventory here are suprising: a) why so large for a quantity so close to zero (which seems to be expected given the assumptions)?, b) why does the uncertainty shrink so much over time?, and c) why did the uncertainties change so much from the previous iteration? Some mention of this with a reference would be helpful.*

Using absolute error may have advantages, as indicated by the reviewer. As noted in l1443, we use errors provided by the European Union for Forest land, Grassland, and Cropland as percentages for a single year, which we apply directly as a first-order estimate for all years. If uncertainties cross zero, they are reported as larger than 100%. We have made the following modifications to the text:
    L680: "The UNFCCC NGHGI uncertainty of $CO_2$ estimates from Forest land across the EU27+UK, computed with the error propagation method (95 % confidence interval, see IPCC, 2006), is 13.5 % for the year 2019 (EU NIR, 2021). This percentage is applied across all years for both FL and FL-FL, and in year 2019 it translates into an uncertainty of 12 Tg C for FL-FL."

We have also added the following:

L354: "Uncertainties for FL, CL, and GL are reported as percentages by the European Union, and we use them directly. An uncertainty greater than 100 % implies that either a sink or a source is possible."

In addition, we have added the second sentence in the above to similar text in the new section in the Appendix on uncertainty in the NGHGI.

L1366: "Note that the above procedure does not apply to LULUCF categories (FL, CL, and GL). Estimates for these values were taken directly from the EU NIR (2021) without gap-filling or consideration of correlations. An uncertainty greater than 100 % implies that either a sink or a source is possible. As the values are given for only one single year, this value is applied uniformly across the whole timeseries."

Reasons for the large uncertainty in Grassland (353.8% in Table 6.41 of the 2021 EU NIR) are, unfortunately, not given by the National Inventory Report. We had included a paragraph of discussion on this issue at l771-781, but it does not completely address the reviewer's concerns. Unfortunately, little additional information is available. We have added the following at L798: "Uncertainty estimates for the EU27+UK come from a synthesis of estimates for each of the 28 Member States, and are applied to each year individually based on the data provided for a single year (2019). The apparent drastic change in uncertainty from 1990 to 2019 is due to the emissions getting much closer to zero (i.e., 7.8 Tg in 1990 compared to 0.5 Tg in 2019), which itself is due primarily to changes in the way Grassland is treated in the United Kingdom, Bulgaria, and Sweden (EU NIR, 2021)."

We incorrectly had 752% in the GL plot, due to mistakenly adding in the uncertainty for the UK. Due to previous agreements, the UK emissions are included in the EU estimates up to and including the 2022 version of the NGHGI, despite that the UK had exited the Union. The FL and CL plots were affected in a similar manner, but the differences are smaller (13.0 to 13.5% for FL, and 48.0 to 61.1% for CL). We have thus changed the uncertainties in the GL NGHGI timeseries to 353.8%; those in the FL timeseries to 13.5%; and those in the GL timeseries to 61.1%. The data provided in the Zenodo repository has been updated accordingly.

*l654 - [depending on their timing in relation to the growing season. This is in line with flux tower measurements that show significant year to year variability (Ciais et al. 2005).] → Please quantify this, even roughly. Do eddy-covariance NEE IAV show variability within 10% of the model, a factor of 2, etc.?*

During the 2003 drought in Europe, eddy covariance NEE for forested sites in Europe varied from -51 to +366 g C m$^{-2}$ yr$^{-1}$ compared to the previous year, according to Ciais et al. (2005). By using the Forest land remaining forest land area for the EU27+UK as reported in the EU NIR (2021) of 163567 kha, this results in a range of -83 to 600 Tg C yr$^{-1}$, which is quite significant given the overall Forest land sink of around -80 Tg C yr$^{-1}$. This is, obviously, only a very rough estimate of the variability, given that site-level extremes are likely cancelled out. Visual examination of Figure 1 in the same paper suggests that 50% of the NEE may be lost during an extreme drought from June to August, which suggests around 15% lost throughout the year. We have added the following text at L713:

"...in relation to the growing season. Flux tower measurements show that carbon sink strength in a European forest may weaken by 50 % during a summer drought, or a loss of 15 % of net carbon uptake over the course of the year (Ciais et al., 2005)."

*l662 - [A few reasons for differences between estimates seen in Fig. 4 and 5 can be readily identified. For this study, the ORCHIDEE model used the ESA-CCI LUH2v2 PFT distribution (a combination of the ESA-CCI land cover mapfor 2015 with the historical land cover reconstruction from LUH2 (Lurton et al., 2020)), and assumes that the shrub land cover classes are equivalent to forest. In terms of area, the original ESA-CCI product corresponding to the EU27+UK shows shrub land equal to about 50 % of the tree area in 2015. A similar analysis using the FAOSTAT domain Land Cover, which maps and disseminates the areas of MODIS and ESA-CCI land cover classes to the SEEA land cover categories 15 , shows that shrub-covered areas are around 20 % of that of forested areas for the EU27+UK. The impact of classifying shrubs as "forests" on the total carbon fluxes could therefore account for a significant percentage of the differences between*

*ORCHIDEE and other results in Fig. 5. In addition, CBM depends strongly on input data and related uncertainty. Historical data are retrieved from both country and EU statistics and usually refers to forest management units rather than individual inventory plots. Finally, trends in forest carbon strongly result from management, which are not represented in this version of ORCHIDEE but are included in CBM and EFISCEN-Space.] → This paragraph is a bit confusing to me. First, I thought the fundamental difference between Figs 4 and 5 was the quantity being evaluated:  FL vs. FL-FL.  Second, it appears you only discuss the differences between the TBM estimates, not the inventories; I would specify this in the first sentence.  Finally, instead of just listing "set up" differences and saying they could account for the flux differences, it would be helpful to the reader to say that a particular aspect of the set up would, e.g., induce more variability or account for either a positive or negative difference.*

The reviewer is correct: the difference between Figs. 4 and 5 is FL vs. FL-FL.  ORCHIDEE is thus not directly comparable with CBM or EFISCEN-Space.  This paragraph is indeed confusing.  We have re-written it at L676 as:

"The top and bottom panels in Fig. 3 are not directly comparable due to different quantities being displayed (FL-FL vs. FL).  For the NGHGI, the value in the bottom panel is simply the value from the top panel with the addition of emissions/removals on land converted to Forest land within the past 20 years.  The sink gets stronger by around 20 Tg C yr$^{-1}$ when considering FL, which is to be expected as abandonment of Cropland or Grassland and subsequent regrowth of forest results in a net uptake of carbon due to storage in woody biomass.  The UNFCCC NGHGI uncertainty of $CO_2$ estimates from Forest land across the EU27+UK, computed with the error propagation method (95 % confidence interval, see IPCC, 2006), is 13.5 % for the year 2019 (EU NIR, 2021).  This percentage is applied across all years for both FL and FL-FL, and in year 2019 it translates into an uncertainty of 12 Tg C for FL-FL.

Differences within the top panel of Fig. 3 are small, perhaps because all three approaches (NGHGI, CBM, EFISCEN-Space) rely heavily on forest inventory statistics.  The same can be said for FAOSTAT FL fluxes in the bottom panel of Fig. 3.  Among all the data plotted on the two graphs, ORCHIDEE stands out.  Despite site-level evaluation (e.g., Vuichard et al., 2019), the vegetation classes in ORCHIDEE are fairly broad (e.g., temperate needleleaf evergreen) and parametrized to reproduce global fluxes, which means ORCHIDEE may be less suitable for regional simulations without further adjustments.  As trends in forest carbon strongly result from management, the lack of explicit management in this version of ORCHIDEE also likely contributes, given the importance of management across Europe."

*Figure 4 (FL-FL) as with Fig. 3, I'd consider moving top panel to appendix or SI.  If it's kept here, I strongly recommend removing the transparency as it's a bit hard to see that way.*
*Figure 5 (FL) - given the similar between FL and Fl-FL, I wonder if Fig. 4 is necessary in the main text.  This paper is already massive, and finding ways to trim the main text, at least, will help readability.*
*Figure 6 --> Even here where the difference is substantial between top and bottom plots, I think the explanation in the text saying the Ecosse and Epic now agree better (see also Appendix Fig. X) would be sufficient.*

We struggled with what to do here.  On one hand, we understand the reviewer, and would like to improve readability.  On the other hand, readers find it quite informative to be able to quickly compare to the "old" results and see the differences illustrated.  However, the goal of this paper is to provide a synthesis on the current state-of-the-art.  This does not necessarily mean extensive comparisons to the previous "state-of-the-art" in the main text.  Additionally, several reviewers found the main text too long, and one reviewer stated it was unbalanced towards the land fluxes.  Therefore, we have removed the "old" plots on all graphs.  In addition, we have combined six CO2 land plots into three: 1) FL and FL-FL, 2) CL and GL, and 3) NGHGI versus bottom-up and top-down models.  One plot was placed into the Appendix to support a point given in the text.  We combined the sections with the plots and modified the text to improve readability, also attempting to remove superfluous text to reduce the overall length, as noted in the general response.

*L677 - [Cropland section ]→ for both this and the grassland section following, it would be helpful to gauge the reality of the assumptions made in the models and inventories by comparing the net uptake to measurements, either biometric or eddy flux.*

We have added the following text at L772: "ORCHIDEE, EPIC-IIASA, and ECOSSE have previously been compared to measurements of net carbon fluxes and soil organic carbon changes at the site level (e.g., Balkovič et al., 2020; Chen et al., 2019; Zhang et al., 2018; Vuichard et al., 2019). Further comparison is outside the scope of this work, given site heterogeneities and the challenges in upscaling such data to a regional level as presented here. We note that this version of ORCHIDEE only includes management implicitly, which makes direct comparison to specific sites less informative."

We have also added the following two references.

Chen, J., Wang, S., Kraxner, F., Balkovič, J., Xu, X., and Sun, L.: Spatial Analysis of the Soil Carbon Sequestration Potential of Crop-Residue Return in China Based on Model Simulation, J. Resour. Ecol., 10, 184-195, https://doi.org/10.5814/j.issn.1674-764x.2019.02.009, 2019.

Zhang, J., Balkovič, J., Azevedo, L.B., Skalský, R., Bouwman, A.F., Xu, G., Wang, J., Xu, M., and Yu, C.: Analyzing and modelling the effect of long-term fertilizer management on crop yield and soil organic carbon in China. Sci, Total Environ, 627, 361-372. https://doi.org/10.1016/j.scitotenv.2018.01.090, 2018.

*l703 - [In the NGHGI, the reported source for the EU27+UK is mostly attributed to emissions from cropland on organic soils in the northern part of Europe] → and otherwise, crops are considered to be a carbon neutral closed loop?*

Cropland is divided into annual crops and woody crops (e.g., apple and olive orchards). Living biomass from annual crops are generally assumed to be in balance, which means whatever is assimilated in a country during the year is also respired. Woody crops, however, can store carbon in woody biomass. Definitions of Cropland vary between countries. Austria, for example, includes Christmas tree plantations; Belgium includes all tillage land where the vegetation falls below the Forest thresholds; France includes vineyards; Sweden only includes regularly tilled agricultural land. For the European Union and United Kingdom, Romania serves as an outlier as it considers some forest plantations to be regeneration activities under the Kyoto Protocol and classifies them as Cropland, resulting in a strong sink (EU NIR, 2021). Carbon is also uptaken by CL-CL mineral soils in France and Spain. These exceptions are not enough to counterbalance the overall emissions from organic soils, in particular from Land converted to cropland (X-CL); although X-CL accounts for only 9% of the total Cropland area, it is responsible for 73% of emissions across the region. We have added the following text at L745 to clarify this:

"...of peat. In general, annual crops are assumed to be in carbon balance: any carbon assimilated during the year is respired in the same location. Woody crops (e.g., apple or olive orchards), however, are an exception, and Cropland on mineral soils uptake carbon in both France and Spain. Romania reports a strong sink on Cropland due to the inclusion of some forest plantations. Overall, emissions from organic soils on Land converted to cropland dominate, however; despite accounting for only 9 % of total Cropland area in the EU27+UK, they are responsible for 73 % of Cropland emissions (EU NIR, 2021)."

*l711 - [ORCHIDEE also shows a much larger year-to-year variation due to the response of vegetation and respiration fluxes to sub-daily meteorology. EPIC-IIASA and ECOSSE both operate on daily timescales (ECOSSE was updated to daily for this work, though the previous version was monthly).] → Can you rule out that the IAV wouldn't simply result from weather variations at the daily scale? In other words, could Orchidee just have significantly higher sensitivity to (e.g.) soil moisture or temperature than the other models? The explanation below seems more like a hypothesis rather than a conclusion based on numerical experiments with the model.*

The hypothesis is that more extreme values, and thus more extreme fluxes, will be seen with subdaily forcing data. However, EPIC-IIASA and ECOSSE both use minimum and maximum temperatures as daily input, not just the mean, and therefore in theory both models may see extremes similar to what ORCHIDEE experiences at the subdaily scale. We feel the reviewer is therefore correct, and we have replaced the start of this paragraph with the following (L756):

"ORCHIDEE also shows a much larger year-to-year variation than EPIC-IIASA and ECOSSE. This is unlikely to be caused by model timesteps (EPIC-IIASA and ECOSSE at daily, ORCHIDEE at half-hourly) as both EPIC-IIASA and ECOSSE use minimum and maximum temperatures during the course of the day as input, not simply the mean daily temperature. Therefore, all three models should see similar extremes, and crop vegetation may simply be more sensitive to meteorological forcing in ORCHIDEE. FAOSTAT and NGHGIs are mostly..."

Note that this paragraph has undergone additional revisions due to shortening efforts when merging Cropland and Grassland sections.

*l883 - [As mentioned above, taking the difference of the TRENDY S2 and S3 simulations provides an estimate of the net flux from land use change, but inconsistencies are introduced either way, and therefore further research is needed in order to establish which approach (S3-S2, or simply S3) leads to the most consistent comparison.] → it would help if this were explained, because the earlier argument that S2 vs. S3 was a good estimate of net. I thought that LULUCF would also be a net, but without a specific definition of LULUCF, I'm not sure…*

At l856, we mentioned that S3-S2 is used in the GCB, but that it does not fully resolve inconsistencies compared to using only the S3 simulations, pointing to four specific examples (net vs. gross LUC, managed vs. unmanaged land, wood harvest, and 'convert' emissions). This sentiment is re-iterated at the line of this comment (l883): ``inconsistencies are introduced either way." We have tried to make this clearer by introducing the following text, recognizing that some of the relevant discussion has been removed in response to other comments:

at L830: "However, even taking S3-S2 does not permit an apples-to-apples comparison between DGVMs, bottom-up inventories, and bookkeeping models. In particular, "

at L858: "taking the difference of the TRENDY S2 and S3 simulations does not permit a fully-consistent comparison between DGVMs, bottom-up inventories, and bookkeeping models for LULCUF fluxes, and for simplicity we simply report S3 NBP from DGVMs in Fig. 5. Further research is needed in order to establish..."

*l889 - [Bookkeeping models like BLUE and H&N do not include indirect effects on biomass growth due to factors such as $CO_2$ fertilization, nitrogen deposition, and climate change, while NGHGIs implicitly include these impacts on managed land through updated statistics]*

This line was not specifically tagged by the reviewer. However, the reviewer's comment at line 848, made moot by other changes, also applies here, and therefore we address it here. It's perhaps clearer to say that "Bookkeeping models always regrow biomass at the same rate." If the regrowth curves and associated target carbon densities are parameterized from conditions in the year 1750, the vegetation will always regrow at a pre-industrial rate. If they are parameterized for the year 2020, vegetation will always regrow under modern conditions, which does indeed implicitly include $CO_2$ fertilization, nutrient deposition, and climate change. In the bookkeeping approach as used in e.g., the Global Carbon Budget, regrowth curves are representative for present-day conditions and kept the same throughout history. The rationale behind this definition is that it excludes effects related to the loss of additional sink capacity, which occurs in estimates based on DGVMs. In DGVMs, growth rates continuously respond to environmental changes, such that natural and anthropogenic (land-use) drivers cannot be separated (Obermeier et al., 2021). We have modified the text to make this clearer:

L867: "Bookkeeping models like BLUE and H&N always regrow biomass at the same rate. In the bookkeeping approaches used here, regrowth curves are representative for present-day conditions and kept the same throughout history, which is the same approach used in the Global Carbon Budget. NGHGIs, on the other hand, include legacy effects from changing environmental conditions, in particular in soil pools."

*l907 - [CABLE-POP also used a higher resolution land use land cover change (LULCC) dataset for the results submitted to VERIFY (0.25° as opposed to 1.0°). The increased IAV from the high-resolution CABLE-POP compared to ORCHIDEE is suspected to have been introduced through the construction of the LULCC dataset as described in Appendix A2.] → wouldn't the more relevant comparison be between cable-pop veryify and cable-pop trendy? it's clear from fig. 8 that cable-pop verify exceeds its trendy variant because it often is beyond the 0-100th percentile bounds of the trendy ensemble.*

For comment at l907, the reviewer is correct: we can test our hypothesis that higher resolution forcing leads to increased IAV by looking at both ORCHIDEE and CABLE-POP compared to their TRENDY versions. This is not a perfect comparison, as the forcings (in particular, the metrological forcing) is not are not identical between TRENDY and this work. However, such a comparison at least provides a first-order insight. For ORCHIDEE, from 1990-2020, the values for TRENDY and this work are remarkably similar: -168 ± 75 Tg C yr$^{-1}$ and -167 ± 24 Tg C yr$^{-1}$, respectively, where the uncertainty represents one standard deviation of the annual flux values. For CABLE-POP, the results are much different: -40 ± 142 Tg C yr-1 and -92 ± 214 Tg C yr-1 for TRENDY and this work, respectively, showing more interannual variability for the higher resolution forcing version. We have reworded as follows at L883:

"ORCHIDEE and CABLE-POP provide a nice test case of the impact of high spatial resolution forcing on net carbon fluxes in the EU27+UK, as they are present in both the TRENDY ensemble (0.5 °) as well as the VERIFY results (0.125 °). Using one standard deviation of the mean annual net $CO_2$ flux as a measure of the IAV, CABLE-POP indeed shows a much higher IAV at high resolution (-40 ± 142 Tg C yr$^{-1}$ and -92 ± 214 Tg C yr$^{-1}$ for TRENDY and this work across 1990-2019), while the results for ORCHIDEE are almost identical between the two resolutions. More analysis is therefore required to confirm the relationship between spatial resolution and interannual variability in DGVMs for the EU27+UK."

Note that this text appears in a paragraph assembled from others as a result of attempts to shorten.

*l911 - [The differences between bookkeeping models and UNFCCC and FAOSTAT are discussed in detail elsewhere, and focus on the inclusion of unmanaged land in bookkeeping models but not FAOSTAT and UNFCCC methodologies (Petrescu et al., 2020; Grassi et al., 2018a, 2021). ORCHIDEE, CABLE-POP and the TRENDY v10 ensemble means show much higher inter-annual variability due to the sensitivity of the model fluxes to highly variable meteorological forcing at sub-daily time steps which allow for much more rapid responses to changing conditions, as already discussed in the previous sections.] → same comment as earlier — i.e. I don't think you've demonstrated that the large IAV might even be generated by daily variability in met forcing.*

While writing, our hypothesis was that higher resolution leads to more extreme input values, which, in non-linear systems and assuming similar model formulations, can lead to more variability. The reviewer correctly points out that we did not mention "assuming similar model formulations", nor did we support this assumption. Even with this assumption, sensitivities may be controlled by a few parameters, so re-parameterization may account for larger IAV. A much deeper analysis will be needed to determine the true causes of high IAV. We have replaced the second sentence (and the following sentence) by:

L892: "ORCHIDEE, CABLE-POP and the TRENDY v10 ensemble means show much higher inter-annual variability as they simulate sub-annual responses of carbon fluxes to climate, while the climate responses of inventories and bookkeeping models are averaged over multiple years."

*l958 - [As pixels reported for the high- resolution simulations here are around 10 km wide, this implicitly assumes that HWP never travel more than 10 km from the harvest site (this becomes 50 km in coaster resolution simulations like TRENDY). Therefore, removing emissions from lateral carbon transport makes inversions more comparable not only to NFGHGIs but also to DGVMs.]→ I'm a bit confused here. I understand that to maximize compatability between inversely-derived fluxes and inventories, you'd want to remove the spatial pattern of emissions arising from lateral fluxes that occur in reality but are assumed to happen at the site of photoysnthesis in the inventories. However, wouldn't you also want to adjust the inversion at the locations of assimilated C that is then laterally displaced? For example in the US, there is a large annual net sink in the Midwest resulting from crop GPP that is laterally displaced and respired to the east and west. From the atmospheric perspective, this sink is real, but from the inventory perspective it is 'fake' and needs to be corrected by considering how much grain was laterally displaced.*

The correction that we apply includes both of these terms: emission and assimilation. Our terminology is perhaps confusing. We have changed "emissions due to lateral transport of carbon" and related phrases throughout the text to be "net carbon flux due to lateral transport", in addition to adding, "Net carbon flux due to

lateral transport includes both carbon imported into a country/pixel and respired and carbon assimilated in a country/pixel and then transported to a different country/pixel before respiration." at L1885.

*l962 - [The mean of the EUROCOM ensemble of European inversions shows good agreement with UNFCCC NGHGI data, but with a huge spread of annual model results that extends from significant sources into large sinks. This large spread can be linked to uncertainty in atmospheric transport modeling, inversion methods and assumptions, and to limitations of the observation system.] → When you say 'large spread', it's not clear if you are referring to the mean IAV of a given inversion relative to the inventory time series (which can probably be explained as mentioned above by the low inventory repeat frequency), or if you are referring to the inter-model spread in a given year, which then is in fact related to inversion uncertainties you mention. I recommend calculating some basic statistics that allow you to differentiate mean inverse model IAV and inter-model spread.*

We have clarified the text.

L935: " Flux estimates from inversion methods for CO2 land show much more variability than the NGHGI, both on the interannual scale as well as for any given year (Fig. 5, bottom). The mean values from 2010-2018 show good agreement but with an order of magnitude more variability in the inversions: $-88 \pm 60$ Tg C yr$^{-1}$ for EUROCOM and $-80 \pm 6$ Tg C yr$^{-1}$ for the NGHGI, where the uncertainty here is the standard deviation of the annual mean values for each. For any given year, the spread between the inversions is also much greater ($170 \pm 70$ Tg C yr$^{-1}$ for EUROCOM versus $63 \pm 3$ Tg C yr$^{-1}$ for the NGHGI, which represents the mean and standard deviation of the 0-100th percentiles for the inversions and the 95% CI for the NGHGI). This large spread per year..."

*l967 - [The annual mean (overlapping period 2010-2018) of the EUROCOM v2021 inversions (-80 [-175,-4] Tg C yr ) is the closest inversion estimate to the timeseries mean of the NGHGI estimates (-88 ± 31 Tg C yr -1 ), where the error bars for the inversion indicated the [0th,100th] percentiles due to the small size of the ensembles.] → Considering the spread of the inverse model results as a proxy for their structural uncertainties (method, transport, etc.) I think it's important to point out that the ensemble of inversions is consistent with the inventory estimates. Of course it would be good if the single-year inversion spread was much smaller, but this spread is probably a reasonable assessment of the the present uncertainty. Another thing I would find useful is to know how much of an impact the lateral transport correction is making. You could provide an average value and perhaps refer to an additional figure with more detail in the appendix.*

We have added the following text at L948:

"...small size of the ensembles. The ensemble of all regional inversions is consistent with the NGHGI estimates, assuming the spread of the inverse model results is an accurate proxy of the structural uncertainties. The impact of the net emissions of lateral fluxes due to wood trade, crop trade, and rivers is clear: without factoring in their contribution of the approximately $-140$ Tg C yr$^{-1}$, the sink from regional inversions, in particular, would be much stronger than even the strongest estimate of the NGHGI (i.e., the lower boundary on the green bar in Fig. 5)."

*l1002 - [The new GCP2021 inversions show a clear trend towards decreasing the CO 2 sink strength of the land surface after 2017, contrary to the NGHGI estimates which are relatively stable (Fig. 9, bottom).] → This is not at all clear to me given the huge spread in any single year for the models. It could be that the ordering of the GCP ensemble is consistent year to year so that there is in fact a trend, but there's no graphical indication or statistical anlaysis presented that makes me see a meaningful trend.*

The reviewer has a good point here; we were overzealous in searching for a trend in the new GCP results. We have run a Monte Carlo analysis on the GCP data, using simple linear regression and including the uncertainties of each point, assuming that the error bars indicate the 68, 95, or 99.7% confidence interval (i.e., one, two, or three sigma for a normal distribution). There is no period of four years (e.g., 2015-2018, 2017-2020) where zero falls outside the 95% confidence interval of the slope distribution of 1000 linear regressions, although 2015-2018 is close when the uncertainties on each point are assumed to represent the 99.7% CI (i.e., three sigma): $26 \pm 28$ Tg C yr$^{-2}$. We have therefore changed to text to be (L970):

"Despite an apparent trend in the mean of the new GCP2021 inversions towards a source near 2017, the spread of the models precludes significance; following 1000 realizations of a Monte Carlo analysis assuming the min-max ensemble spread represents $3\sigma$ in a normal distribution, the only period of at least four consecutive years for which the 95% confidence interval comes close to excluding zero is 2015-2018 ($26 \pm 28$ Tg C yr$^{-2}$)."

*l1016 - [As shown in Fig. 8, the mean 1990-2019 value of the mean of the 15 TRENDY DGVM simulations is -81.9 Tg C yr-1] → here and elsewhere, is a third sig. fig. Justified?*

Following the rule that significant figures are determined by uncertainties, and that having more than two significant figures in uncertainties is not informative, we cannot justify three significant figures for many of these numbers. We have changed this throughout the text. We keep three significant figures in some situations. For example, when a value is given with asymmetric error bars (e.g., -73 [-135,45] Tg C yr$^{-1}$) as in this case using two significant figures for the lower limit results in only a single significant figure for the mean value. While perhaps not completely warranted, we find keeping three significant figures for one of the limits improves readability. We have added some text to explain our position at L265: "We make every effort to limit the number of significant figures as a function of the error bars. In some cases (e.g., asymmetric error bars which overlap zero), we retain an extra significant figure to improve readability."

*l1539 - [Section on NGHGI for LULUCF] Is this necessary? Wasn't most of this said above? Could you have a common general intro before discussing land and fossil separately?*

The paragraphs from l1539-52 give an overview about the NGHGI. This text is also found at l1372-85. The text at l1539-52 has therefore been removed. In addition, most of the text under the $CO_2$ fossil NGHGI in Section A1 is applicable to both fossil and LULUCF fluxes, and it has thus been moved to a new section (section A2 in Appendix A).

*lL885 - [(BLUE vGCP (-61 Tg C yr -1 )] → would it be possible to stick with TgC (and not Tg CO2) throughout the paper?*

We have changed all units to Tg C, including eliminating Tg $CO_2$, Gt $CO_2$, and Mt $CO_2$. This involved changing the units in two figures (Figs. 2 and A1). The fossil datasets in the Zenodo repository have also been updated.

*l1105 - [Total CO 2 land fluxes from EU27+UK and five main regions in Europe are presented, divided into top-down (top panel) and bottom-up (bottom panel) approaches for clarity.] → Given that the subject of the paper is EU27+UK I found it surprising and somewhat confusing to have the final figure deal not just with these 28 countries but with a much larger set. In particular, my eye was drawn (and you mention in the text) the surprising Eastern European results, but in fact these don't contribute to the EU27+UK values. I recommend that you limit Fig. 11 to EU27+UK only for symmetry with the rest of the paper (including the title) and then present the TD and BU results for other ways of considering Europe in a table or an Appendix figure.*

*Figure 11 (regional map)→ It would be helpful if the legend didn't obscure the map. Especially in the case of Eastern Europe which shows such a large sinks (esp. for TD) it would be useful to see what this part of Europe consists of.*

In response to the reviewer's comments, and in order to shorten the paper in response to a general request by multiple reviewers, we have removed this figure and all associated discussion.

*l472 - [The inversion reports total fossil CO2 emissions calculated from NOx combustion emissions.] → or NO2? This is a bit confusing. Satellites measure only NO2, not NO, so NO2 would be the data source.*

It's true: satellites measure $NO_2$ concentrations. However, $NO_x$ emissions at the source are what is needed to constrain $CO_2$ fluxes when exploiting the co-emission of these different species by fossil fuel combustion. The emission of NO and $NO_2$ during the combustion processes generate complex and varying NO/$NO_2$ ratios over short spatial and temporal scales, and deriving sensible NO2-to-ffCO2 emission ratios at the source of emissions and a sensible link between the $NO_2$ emissions at the source and the $NO_2$ concentrations observed by satellites is therefore very challenging. The $NO_x$ inversions with CIF-CHIMERE account for the atmospheric chemistry

(CHIMERE and its adjoint code have a full chemistry scheme) and thus can link NOx emissions and satellite $NO_2$ observations, thus permitting constraint of fossil $CO_2$ emissions.

We have clarified this in the text below.

*Earlier you said that this inversion used CO2/NO2 ratios, but now you are referring to CO2/NOx ratios.*

The text was not rigorous enough regarding the use of $NO_2$ vs $NO_x$ depending on whether we speak about the concentrations or the emissions. Each occurrence of $NO_2$ has now been checked carefully and corrected when needed.

*l281 [Here, this conversion relies heavily on the emission ratios per country, month and large sector of activity from the TNO-GHGco-v3 inventory (Dellaert et al., 2021)] → if the NO2/CO2_ff ratios are coming from inventories anyway (from E_NO2_ff and E_CO2_ff), what benefit is there to adding NO2 to the mix? Why not just use the CO2_ff? This approach would just seem to add uncertainty unless there are reasons to think that the emissions ratio benefits from some kind of error cancellation and is more accurate than the inventory-based CO2_ff.*

Uncertainties in the fossil fuel emissions can arise from uncertainties in 1) the amount of activity regarding fossil fuel combustion and 2) the emission factors. For fossil fuel $CO_2$ (CO2_ff) emissions, (2) is generally well known, which means the primary potential uncertainty is in (1). In some countries, the uncertainty in (1) can be significant, and we feel that atmospheric inversions can help reduce this uncertainty, as well as provide independent and more timely data. Here, we provide an assessment of the current potential for this, even though our analysis focuses on countries for which (1) is well known and thus for which the inventory-based CO2_ff emission estimates should be accurate.

There is a link between (1) for co-emitted species (e.g., NOx_ff and CO2_ff), while (2) is species-specific. The concept here is thus to derive CO2_ff emissions by exploiting the link in (1) using $NO_2$ satellite data (currently, we lack suitable CO2 observation networks data to derive CO2_ff emissions using atmospheric $CO_2$ mole fraction directly). We acknowledge that the use of NOx/CO2_ff ratios from the inventory is currently a limitation in the process of inferring CO2_ff emissions from atmospheric data since (2) bears large uncertainties for NOx_ff, but this method still allows us to get CO2_ff emission estimates driven by atmospheric data.

We have added clarifications in the text below.

*l472 - [However, in principle, the derivation of CO2 emissions from the NOx inversions should be restricted to fossil fuel CO2 emissions based on the fossil fuel CO2/NOx ratio from the TNO, as there is a better-established relationship between CO2 and NOx from combustion of fossil fuels.] → better than what?*

*l476 - [Finally, it's important to note that the inversion results are not fully independent of the bottom-up methods, as the prior estimates are based on TNO gridded products.] → As mentioned above, it's not just the prior. As big of a problem is that the CO2/NO2 emission ratios (or CO2/NOx?, please clarify) are coming from an inventory, which means, why not just use the CO2ff inventory. It seems to me that the addition of NO2 data can only add uncertainty at this point. The CO2fos inversion has value as a proof of concept, but in the context of the present study, I don't think it's credible enough to include as an independent source of information.*

*l1511 - [For the first time, to our knowledge, variational regional inversions have been performed to estimate the European CO2 fossil emissions using NOx emissions from OMI satellite observations.] → this a bit confusing because OMI is sensitive to mole fractions/concentrations, not fluxes, per se, and moreover OMI measures NO2, not NOx. Since there is already this discussion in the appendix, it would be helpful for the reader not to have to go to another paper to understand a bit about the chemistry (or assumptions) that allow for conversion between NOx and NO2. They are being used interchangably in this paper, which is not necessarily a good assumption.*

We have clarified all of the above comments around the fossil fuel CO2 inversions in the text below.

*l475 - [fuels. Future inversions co-assimilating CO2 data will make a clearer distinction in the processing of fossil-fuel and other anthropogenic emissions.] → Given the fact that CO2 data are strongly affected by terrestrial biosphere fluxes, it's hard to see how co-assimilating CO2 data (without additional measurements such as 14CO2, and other obs perhaps) would actually make things 'clearer'.*

This point touches on a general debate around atmospheric inversions. It's true that $CO_2$ data are strongly affected by terrestrial biosphere fluxes. However, in principle, differences in spatial and temporal differences between the fossil and terrestrial biosphere sources and sinks can be exploited with a good knowledge of the atmospheric transport. In practice, of course, this is challenging. And, actually, we feel that the potential of the incorporation of data on co-emitted species to help distinguish between the fluxes (similarly to what the reviewer expects from $^{14}CO_2$ data) is high.

However, in this case, we were referring to the configuration of the inversions, and not about the theoretical capabilities. We have revised the sentence to make this clearer.

Modified text in response to all of the reviewer's above comments:

L302: " In this analysis, the inventory-based bottom-up $CO_2$ fossil emissions estimates are separated and presented per fuel type and reported for the last year when all data products are available (2017). This updates Andrew (2020) and Petrescu et al. (2021b) which both report the year 2014. In order to provide a quasi-independent estimate of fossil emissions assimilating satellite observations of the atmosphere subject to current capabilities of atmospheric inversions, the CIF-CHIMERE model was used to produce a fossil fuel $CO_2$ emission estimate for the year 2017. CIF-CHIMERE is a coupling between the variational mode of the Community Inversion Framework (CIF) platform developed in the VERIFY project (Berchet et al., 2021), the CHIMERE chemistry transport model (Menut et al., 2013) and the adjoint of this model (Fortems-Cheiney et al., 2021a). To overcome the lack of $CO_2$ observation networks suitable for the monitoring of fossil $CO_2$ emissions at national scale, this inversion is based on the assimilation of satellite $NO_2$ data, which are representative of $NO_x$ emissions, as $NO_x$ is co-emitted with $CO_2$ during fossil fuel combustion. The uncertainties in the anthropogenic activities underlying the fossil fuel combustion are shared by both $CO_2$ and co-emitted species. Therefore, in principle, information from co-emitted species such as $NO_x$ and CO can be used to decrease the uncertainties in fossil fuel $CO_2$ emissions. Recent top-down inversions of anthropogenic $CO_2$ emissions from Europe indicate that uncertainties using satellite measurements of $NO_2$ are much lower than for co-emitted CO when deriving fossil $CO_2$ emissions (Konovalov et al., 2016). Therefore, results shown below only incorporate $NO_2$ and not CO observations. The CHIMERE model includes a full chemistry scheme to enable linkage of observations of atmospheric $NO_2$ mole fractions to surface $NO_x$ emissions. While the spatial and temporal coverage of the $NO_2$ observations is large, there are many factors contribute to uncertainty in fossil fuel emission activity data, including the uncertainties in $NO_x$ emission factors and thus the ratio of $NO_x$ to $CO_2$ emissions. Therefore, the influence of using $NO_2$ observations in determining fossil $CO_2$ emissions is subject to uncertainties which have not been characterized appropriately yet in the framework of VERIFY. Here, this conversion relies heavily on the emission ratios per country, month and large sector of activity from the TNO-GHGco-v3 inventory (Dellaert et al., 2021), which has been partly developed in VERIFY, and which is based on the most recent UNECE-CLRTAP[1] and UNFCCC official country reporting respectively for air pollutants and greenhouse gasses. The detailed descriptions of each of the data products are found in Appendix A3."

L572: "The sole available inversion for $CO_2$ fossil fluxes is produced by the CIF-CHIMERE model, shown in Fig. 1c and Fig. B3 (for a single year). The inversion yields plausible fossil emission estimates, although it is below NGHGI estimates including both Energy and IPPU (Figs. 1a,c,B3) as well as the ensemble of nine bottom-up inventories. Uncertainties of CIF-CHIMERE inversion estimate have not yet been quantified, however they are likely largely driven by large uncertainties in the input data. The satellite observations of $NO_2$ have large uncertainties, which partly explains the small departure from the prior fluxes during the optimization. Emission ratios between $NO_x$ and $CO_2$ are also uncertain (those from the prior are currently used). The
* * *
[1] UNECE Convention on Long-Range Transboundary Air Pollution. https://unece.org/environment-policy/air

atmospheric chemistry surrounding both production and destruction of $NO_2$ is another major source of uncertainty. The inversion reports total fossil $CO_2$ emissions calculated from $NO_x$ fossil fuel combustion emissions. However, in principle, the derivation of $CO_2$ emissions from the $NO_x$ inversions should be restricted to derivation of fossil fuel $CO_2$ emissions based on the fossil fuel $CO_2/NO_x$ ratio from the TNO inventory, since there is no process linking the other fossil $CO_2$ emissions to the $NO_x$ fossil fuel emissions. Future inversions co-assimilating $CO_2$ data will have to make a clearer distinction in the processing of fossil-fuel and other anthropogenic emissions in order to exploit the joint fossil fuel signals in $CO_2$ and $NO_2$ observations. Finally, it is important to note that the inversion results are not fully independent of the bottom-up methods, as the prior estimates and $CO_2/NO_x$ emission ratios are based on TNO gridded products. However, part of the lack of departure from the prior can also be attributed to the general consistency between the prior and the observations, which raise optimistic perspectives for the co-assimilation of co-emitted species with the data from future $CO_2$ networks dedicated to anthropogenic emissions."

*l1524 - [Considering the short NO 2 lifetime, we do not consider its import from outside the domain: its boundary conditions are set to zero. Nevertheless, we take into account peroxyacetyl nitrate (PAN) and the associated NOx reservoir for the large-scale transport of Nox. ] → PAN is a component of NOy, not NOx…. sorry if I'm missing something obvious here.*

PAN has a longer lifetime than $NO_x$. In our model and in the atmosphere, PAN modifies the atmospheric $NO_2$ concentrations, which are observed by the satellite, due to the atmospheric chemistry. This link between the PAN and the $NO_2$ concentrations impacts our inversion of the $NO_x$ emissions. We have modified the text:

L1490: "Considering the short $NO_2$ lifetime, we do not consider its import from outside the domain: its boundary conditions are set to zero. Nevertheless, we take into account peroxyacetyl nitrate (PAN) for the large-scale transport of $NO_x$. Due to atmospheric chemistry, it represents an important NOx reservoir and has a significant impact on the regional $NO_2$ tropospheric columns observed by OMI."

*l1554 - [Each country uses its own country specific method which takes into account specific national circumstances (as long as they are in accordance with the 2006 IPCC guidelines), as well as IPCC default values, which are usually more conservative and result in higher uncertainties.] → what does [conservative] mean here? resulting in smaller sinks? please clarify. also, while it makes sense in a general way, that using default instead of country-specific values would be more uncertain, from the IPCC guidline angle on calculating formal national inventory uncertainties, I don't see how this follows.*

By "conservative", we mean that the default values are designed for the widest possible application within relatively broad classifications. For example, an emission factor for "temperature deciduous forest" is designed to apply to such forests in temperate zones across the world, not taking into account variations which arise due to particulars in regional species composition which may result in significantly higher or lower estimates.

For the uncertainty, we derive our interpretation from Volume 1, Chapter 3 of the 2006 IPCC Guidelines: "Default methods represent a compromise between the level of detail that would be needed to create the most accurate estimates for each country, and the input data likely to be available or readily obtainable in most countries. Default methods are often simplifications, and may introduce large uncertainties into a national estimate."

We have made the following modifications to the text (1512): "...as well as IPCC default values, which are a "compromise between the level of detail that would be needed to create the most accurate estimates for each country, and the input data likely to be available or readily obtainable in most countries" (Volume 1, Chapter 3 of IPCC, 2006). They may, therefore, result in higher uncertainties."

*Table A2 → This table is pretty hard to read in its current format. But I don't have any good suggestions…. There is a lot of white space. Is there a way to reformat, perhaps as three separate tables?*

We take note of the concern over the white space and the presentation of the information in this table. The goal of this table is to summarize differences between three publications. We have thus pulled out what we consider to be the biggest changes, using columns to indicate the publication and rows to indicate the datasets, separating between bottom-up, top-down, fossil, and land fluxes. We find this to be much more readable.

**Minor – textual modifications of a sentence or less, without an explicit proposition by the reviewer**

*l66 - [This study provides a consolidated synthesis of fossil sources (CO 2 fossil) and natural sources ]→ do these include formally "managed" ecosystems, or just the non-managed fraction of the terrestrial biosphere.*

We have added the text, "(including formally managed ecosystems)".

*l70 - [this study aims to answer essential questions identified in the previous syntheses and understand the differences between datasets, particularly for poorly characterized fluxes from managed ecosystems.] → likewise, are these considered 'natural'?*

We have added the text, "and unmanaged".

*l88 - [The sole top-down inversion of fossil emissions currently available accounts for 3800 Tg CO 2 yr -1 (1038 Tg C yr -1 ), a value close to that] → doesn't seem that close.*
*l466 - [The inversion yields plausible and consistent fossil emission estimates compared to nine bottom-up estimates from BU datasets with global coverage] --> in what sense? Earlier, it was reported that fossil emissions from the inversion were 3800 Tg CO2/yr, which seems far outside the span of the BU estimates.*

An incorrect data value was taken for the CIF-Chimere fossil inversion. The value for the year 2017, which is the most recent year shared among all the bottom-up datasets, is 875 Tg C. We have modified the text in the abstract to read (L91): "The sole top-down inversion of fossil emissions currently available accounts for 875 Tg C in this same year, a value outside the uncertainty of both the NGHGI and bottom-up ensemble estimates and for which uncertainty estimates are not currently available."

We have modified the text at L573 to read: "The inversion yields plausible fossil emission estimates, although it is below NGHGI estimates including both Energy and IPPU (Figs. 1a,c,B3) as well as the ensemble of nine bottom-up inventories."

*l94 - [and was calculated after removing land- atmosphere CO 2 fluxes caused by lateral transport of carbon (crops, wood trade and inland waters)] → for countries with ocean borders, shouldn't this also include terrestrial C export to the oceans?*

We have clarified in the text that these fluxes include river transport to oceans by rephrasing, "crop trade, wood trade, river transport, and net uptake from inland water bodies". We have also removed a redundant "the" from that sentence.

*l103 - [However, uncertainties in top-down approaches to estimate CO 2 fossil emissions remain uncharacterized and are likely substantial.] → I don't doubt this. However, uncertainties in TD land fluxes could also be underestimated because of not accounting for transport uncertainties, e.g.*

We have added the text, ", in addition to known uncertainties in top-down estimates of the land fluxes".

*l116 - [In contrast, global net CO2 emissions from land use and land use change (LULUC, primarily deforestation) estimated from bookkeeping models and dynamic global vegetation models (DGVMs)] -> isn't this usually LULUCF in UNFCCC nomenclature?*
*l153 - [AFOLU] - It would be useful to note how this relates to LULUCF.*

*l867 - has [LULUCF] been defined yet? I guess that this is the sum of the forest, grassland, and cropland categories.*
*l1553 - [For the biogenic CO 2 emissions from sector 4 LULUCF, methods for the estimation] → has 'sector 4' been defined?*

We have addressed the above three comments by adding: "; see glossary at Table A1 for more details" at L121, and adding the terms to the Glossary. "Sector 4" is the official number of the sector in the NGHGI reporting. We have clarified the text ("LULUCF (Sector 4 in the terminology of the NGHGIs)") and added it to the LULUCF definition in the glossary. Footnote 3 also previously referred to AFOLU. We have added text in that footnote pointing the reader to the glossary.

*l117 - [change (LULUC, primarily deforestation) estimated from bookkeeping models and dynamic global vegetation models (DGVMs) were estimated to have a small decreasing trend over the past two decades] -- > also quantify 2020 levels to compare with FF.*

We have added the text ", and a value in the year 2020 of $900 \pm 700$ Tg C yr$^{-1}$".

*l162 - to me 'build mutual trust' pre-supposes a level of compatability or accuracy between national inventories and other approaches that is not yet warranted at this point in the analysis. What if detailed scientific analysis demonstrates fundamental issues with national inventories and/or other TD or BU approaches. I would suggest simply saying that the goal is to learn about both approaches by comparing them.*

We have modified the sentence to read, "to advance a trust-building process by mutual understanding developed through comparison of both approaches", as well feel that part of the trust-building process is mutually understanding the approach (our co-authors include folks from both communities).

*Table 2 - my understanding is that the inventory LULUCF values represent annual managed terrestrial biosphere C reservoir (stock) changes, not fluxes, whereas the model values of 'net land flux' are large-scale NEE.*

Under the IPCC guidelines, Member States can choose either a stock-change or a gain-loss reporting method. The latter are flux based. In the EU, countries use a mix of approaches, with 16 out of 27 countries implementing "Gain-loss" (Table 6.12 in the EU NIR, 2021). We have added this both as a footnote to Table 2 and as part of our "Net flux" definition in the Glossary.

*Table 2 (NBP in ORHCIDEE) - does this include biomass burning (wildfire plus ag and other non-energy combustion)? as mentioned in general comments a glossary or table of definitions would be helpful.*
*l652 - [ORCHIDEE results indicate that climatic perturbations and extreme events (multi-month droughts, in particular) can have significant impacts on the net carbon fluxes depending on their timing in relation to the growing season.] → It would be helpful, here or earlier, to know if the version of Orchidee used here includes fire. If the IAV simulated by Orchidee does not include fire, it's likely that a similar model simulation that did include fire would produce larger IAV.*
*Table 2 - [(CSR)] --> "Total CO2 inverse flux" is going to depend on the setup of the inversion; it's not necessarily NBP (which needs to be defined either in table footnotes or in the main text prior to the introduction of the table.)*
*l321 - [The two updated inverse model ensembles presented are the GCP2021 for the period 2010-2020 (Friedlingstein et al., 2022) and EUROCOM for the period 2009-2018 (Monteil et al., 2020; Thompson et al., 2020).] –>fixed/prior fossil emissions are described here. what about biomass burning?*

We have added three footnotes to Table 2 discussing the net carbon flux:

[a] Member States use a mix of gain-loss and stock-change reporting methods (Table 6.12 in EU NIR, 2021). The net flux from a given country can thus be based on either stock changes or flux changes.

[b] The definition of NBP various from model to model. Most models include harvest, but not necessarily other disturbances. Please refer to Table C2 for more details.

[c] The net carbon flux from regional inversions over land is the residual after fixing fossil $CO_2$ emissions and $CO_2$ fluxes from biomass burning. In other words, any flux not included in those two categories is reflected in the net flux from the inversions. Biomass burning is prescribed in two of the EUROCOM models (LUMIA and FLEXINVERT+; see Monteil et al., 2020, and Thompson et al., 2020) and ignored (i.e., assumed negligible in Europe) for the others.

In addition, we have added a term "net carbon flux" to the Glossary, with similar details.

*Table 2 – [1Y] → one-yearly resolution? this isn't clear. maybe define in table footnotes.*
*Table 2 – [3H] → assume this means three-hourly, but this should be noted somewhere. can't expect readers to know this.*

We have added a sentence to the caption of the table: "The timesteps 1Y, 1M, 1W, and 3H refer to the availability of the data: "one year", "one month", "one week", and "three hours", respectively."

*Table 2 – [Annual uncertainty gap-filling for total LULUCF by Environment Agency Austria (EAA).] → reference for this?*

This work has been the result of close collaboration between the VERIFY project and the EAA, and is described in more detail in Appendix A. There is currently no published reference. This text has been replaced by text referring to the EU GHG inventory team. Similar text has been added in several locations (Table 1, Table 2, Table C1), as well as a sentence in the acknowledgements.

*l227 - [We have taken efforts to exclude outliers from the datasets used to construct ensembles,] → this seems rather important as the definition of an outlier is subjective. Is there a discussion of this later on or in another article?*
*l228 - [distributed. For this reason, we display the mean for all ensembles.] --> mean is of course fine in this case, but for random, normal distributions, the mean and median are the the same.*

Upon re-examination, we found the use of the term "outlier" misleading. We removed only three total datasets from the three ensembles (GCP, TRENDY, and EUROCOM), which were detailed in both the Appendix and Table 2. One was removed for purely technical reasons, while the other two were removed after visual inspection (Figure A5). We have changed the main text to not refer to "outliers" (L257): "While the mean and median become identical in the case of independent randomly-distributed data, the median downplays the skewness of the data." We removed the sentence highlighted by the referee above ("We have taken efforts..."). We have also added the reason why we did not include ISAM (several outlier years upon visual inspection) in the TRENDY ensemble in the appendix, as we had previously mentioned it was not included without elaboration.

L1865: "In the case of VERIFY, 15 of the 16 models for TRENDY v10 (except for ISAM, which after visual inspection showed several outlier years) were used."

*l348 - [The LUMIA inversion system submitted four simulation results to the VERIFY project, based on the 2018 Drought Task Force project (labeled here as EUROCOM, Thompson et al., 2020). The primary difference is that the years 2019-2020 were added based on boundary conditions using TM5 and ERA5 meteorological data. -> a bit confusing because TM5 nearly always derrives its airmass fluxes (and some additional meteo) from ERA5 meteo. How are TM5 and ERA5 different here?*

We have clarified the text as (L393): "The LUMIA inversion system submitted four simulation results to the VERIFY project, based on the set-up developed for the 2018 Drought Task Force project (labeled here as EUROCOM, Thompson et al., 2020), but with a refined definition of both prior and observation uncertainties. Also, for the years 2019-2020, the transport models (FLEXPART and TM5) were driven by ERA5 meteorological data, whereas for previous years, ERA-Interim data were used."

*l385 - [The Industrial Process and Product Use (IPPU) sector contributes 7.6 % or 0.2 Gt CO2 yr-1. CO2 emissions reported as part of the agriculture sector cover only liming and urea application,] -> my guess is that the vast majority of this is cement. Please include this fraction, perhaps parenthetically.*

The fraction of cement has been included parenthetically, and the units changed in accordance with another comment: "68 Tg C (21 Tg C of which is cement production)".

*l404 - [Eight bottom-up models produce a 25-75 % percentile which is almost invisible on the scale of the graph (center-top, gray shading).] --> because this is hard to see on the graph, it would be nice to quantify in words, e.g. 'the 25th - 75th percentile spread among eight BU fossil models/data products represents just X PgC/yr, or Y %'.*

The following text has been added (L503): "Eight bottom-up models produce a mean 25-75% percentile spread of 24 Tg C yr$^{-1}$ across the overlapping timeseries (center-top, gray shading)."

*l429 - not at all clear what 'Remain' means and how it relates to 'Total'. Sorry if I missed the discussion of this earlier.*

This was briefly touched upon at l300-301 and footnote 8. We have added some additional text at the site of the reviewer's comment to remind readers and hopefully clarify for them what the total is (L534): "between those models estimating "Remain" and "Total" fluxes, where Total indicates all land of a particular type (e.g., Forest land) regardless of the length of time it has been this type, i.e., Total is the sum of all Remain and Convert."

*l440 - [if IEA doesn't include any carbonates, then carbonates are removed from all emissions datasets that report these separately.] → if IEA is, for example, the only one missing carbonates, why not adjust IEA in some reasonable way to make it more realistic?*

The reviewer has pointed out an alternative way to harmonize data. We preferred to not modify the original data, and instead compare just those values reported.

*Also, the way this is written is as a hypothetical statment. Is that all it is, or did you mean to say 'IEA doesn't include carbonates, therefore carbonates were subtracted from all other products…'?*

This is correct. We have re-phrased the statement to not be hypothetical (L545): "For example, IEA doesn't include any carbonates, and thus carbonates were removed from all emissions datasets that include them."

*l451, Figure 2 - two things don't appear consistent with Fig. 1b. a) units CO2 v. C., and b) the spread. Even considering the 25-75th percentile, the spread in Fig. 2 is ~ 150 Mt CO2/yr or ~40 TgC/yr, which seems a bit bigger than the gray banding in 1b.*

a) We have modified the units, as noted in another comment.

b) We appreciate the reviewer's keen eye to have noticed this. The spread in Figures 1b and 2 are both 100 Tg C per year (when the 0th-100th percentile is taken for Figure 1b). As the datasets are grouped together more closely, the 25th-75th percentile comes out to be much narrower, only 24 Tg C, as noted in another comment. We have added a sentence to clarify this at L570: "Across the overlapping timeseries, the mean value of the 25th-75th percentile is 24 Tg C yr$^{-1}$, with a 0th-100th percentile of 100 Tg C yr$^{-1}$."

*l489 - [However, several gaps still exist, mainly in non-forest lands and non-biomass pools (e.g., EU NIR, 2022).] → what is an example of this?*

We have modified the sentence at L600: "However, several gaps still exist, mainly in non-forest lands and non-biomass pools (e.g., soil carbon in Forest land mineral soils, dead organic matter on Cropland and Grassland; for more details, see Table 6.6 in EU NIR, 2021)."

*l508 - [As discussed by Petrescu et al. (2020), the fluxes reported in NGHGIs relate to emissions and removals from direct LULUCF activities (clearing of vegetation for agricultural purposes, regrowth after agricultural abandonment, wood harvesting and recovery after harvest and management) but also indirect CO2 fluxes due to processes such as responses to environmental drivers on managed land] --> recommend being explicit here. maybe parenthetically mention (CO2, temperature, moisture).*

Added, "(e.g., long-term changes in $CO_2$, air temperature, and water availability)", as short-term changes (e.g., daily) are not reflected.

*l698 - [carbon is assimilated into biomass growth during the growing season, after which the biomass dies, is partitioned between litter and harvest (50 % to each), and either decays or vaporizes, respectively.]→ this usage here and above is a bit odd. I think respired would be clearer and more correct. Moreover, I don't understand the difference between decay and 'vaporization'. Assimilated carbon, once dead, decays and can move back to the atmosphere as CO2 via heterotrophic respiration (I suppose very small fractions could move to gas phase via some reduced compounds too).*

In model terms, harvest biomass that is "vaporized" enters the atmosphere immediately. Anything not vaporized enters the litter pool, where it may respire over the course of days to years, or be transferred to the soil pool, where it may also respire over longer periods of time. "Decay" and "respire" are synonymous as they both imply a process that occurs over time, while "volatilize" is more similar to instantaneous and complete combustion. We have replaced "decay" with "respire" in the text where appropriate, in addition to adding these terms to the glossary. In some places, we use "gradual respiration" to emphasize the time component.

*l699 - [NGHGIs assume that all aboveground biomass of non-woody crops re-enters the atmosphere at harvest.] → so the national inventories, at least those used here, do not account for any later use (or horizontal translocation) of grains?*

Correct. This is the reason why emissions of carbon from lateral transfer of crop products need to be factored into account when comparing inventories to inversions. We have modified the sentence in order to clarify: "The NGHGIs of Member States in the EU assume that biomass lost to harvest (i.e., all biomass in annual crops, but only part of the total biomass for perennial woody crops) enter the atmosphere in the same MS during the same year (EU NIR, 2021, and IPCC, 2006). As such, no displacement of carbon fluxes due to trade is taken into account."

Update: this text was removed during shortening efforts, as it was repeated in several places in the combined Grassland and Cropland section.

*l759 - [This category also includes all grassland from wild lands to recreational areas as well as agricultural and silvo-pastoral systems, subdivided into managed and unmanaged, consistent with national definitions (Petrescu et al., 2021b).] → Maybe this is a very minor land type in Europe (not so in the US), but does this also include urban and suburban turf grass? (sports fields, lawns, golf courses etc.)*

The Grassland category is unlikely to include these types of grasses. Bulgaria includes "permanent lawns and grassland which are not used for production purposes" under Grassland, but no other Member State explicitly mentions something similar. Instead, "gardens", "green urban areas", "city parks", and "courtyards" are mentioned in the Settlements category by multiple Member States. We have added text about this (L793): "Urban green spaces, on the other hand, are often included in the Settlements category (EU NIR, 2021), which is not explicitly simulated by any bottom-up model reported here."

*l928 - [In these inversions, all components of the carbon cycle that contribute to the observed atmospheric $CO_2$ gradients between stations are implicitly included as the inversions incorporate observed atmospheric concentrations of $CO_2$.] → was mole fraction used earlier? if so, be consistent*

Done. "mole fraction" was also added to the glossary to help readers understand the difference between this and "concentration".

*l934 - [Bottom-up methods (including all the NGHGIs for European countries) do not consider emissions and removal of atmospheric $CO_2$ due to lateral transport of carbon, while observations assimilated into top-down inversions record all $CO_2$ fluxes without separating their components.] → I'm not sure I agree with this, but I am not an inventory expert so I may be mistaken. However, if inventories (e.g.) track terrestrial biosphere stock changes — say the soil pool in a forest plot between five-year measurement intervals — then any lateral transport of SOC (e.g.) by rivers will have reduced the stock increment. In this sense, wouldn't the inventories implicitly account for at least some lateral transport? Or is this statement true only in regard to lateral crop and wood transport where the inventory approaches assume local respiration of harvest (wood and grain)?*

We have clarified that this is primarily carbon from biomass, and added some text about SOC. The sentence now reads (including text from another comment; L911): "Bottom-up methods (including all the NGHGIs for European countries) do not consider emissions and removal of atmospheric $CO_2$ due to lateral transport of biomass carbon, while inversions calculate geographically resolved net land-atmosphere $CO_2$ fluxes without regard to the original location of photosynthetic assimilation. Some lateral transport of soil organic carbon may be taken into account by measuring stock changes, but given the mix of stock-change and gain-loss methods

used in NGHGIs in the EU, and the presence of methods ranging from Tier 1 to Tier 3, exactly how much is far from trivial to determine."

*l1017 - [TRENDY DGVM simulations...(with a a range of [-285,118] Tg C yr -1 )] → sorry, I forget your criteria for 25/75 vs. 0/100 but given 'range' I take this to mean 0/100 and wonder if given 15 members, if 25/75 would be reasonable?*

We have changed the text to use the 25-75$^{th}$ percentile (-172, -20), removing a superfluous "a". We have also replaced the word "range" with "25-75$^{th}$ percentile uncertainty range".

*l1023 - [Note that these categories are used as input to top-down approaches, and therefore cannot be disaggregated into results after the simulation.] → It's not clear what 'category' means in this case. Do you mean these models? I would expect that the TD approaches would simply aggregate BU NEE (or NBP) from all categories as a single temporally and spatially resolved prior, erasing the category definitions (about which the atmospheric CO2 obs couldn't discern).*

In this case, "categories" refers to Land types. We have clarified the text by modifying l1019 ("In addition, the number of Land type categories included"), l1020 ("NGHGI (e.g., Forest land, Wetlands)"), and l2023 ("Note that Land types are aggregated as prior fluxes in the top-down approaches,").

Update: all this text was removed during efforts to shorten.

*l1035 - [Posterior uncertainties in top-down (TD) estimates mostly come from: 1) errors in the modeled atmospheric transport; 2) aggregation errors, i.e., errors arising from the way the flux variables are discretized in space and time and error correlations in time; 3) errors in the background mole fractions; and 4) incomplete information from the observations and hence the dependence on the prior fluxes. → Figure 10 and this discussion are, as far as I can tell, only dealing with structural/systematic errors and not random errors (although (4) is related to random error). Although systematic errors, such as those arrising from simulated vertical tracer movement/distribution, probably dominate total uncertainty, random errors can still be important. The ensemble approach of Chevalier et al referred to below does allow calculation of random error in a single inversion, but it appears that the information in the Chevalier ensemble approach is being used in a somewhat limited way. More importantly, I think it would be useful to mention that the multi-model ensemble approach is being used as a proxy for estimation of systematic error and that calculation of random error is generally difficult when using the most common inverse model flux optimization approaches.] → this is generally confined to regional inversions. it is true for global inversions as well, implicitly, in the sense that the inflow of CO2 gradients to a region of interest (Europe here) needs to be accurate, but I suspect that this point refers to the explicit case of regional inverisons.*

1) We have added two sentences about the multi-model ensemble at the end of the paragraph (L994): "The multi-model ensemble approach is being used as a proxy for estimation of systematic error. Calculation of random error is generally difficult when using the most common inverse model flux optimization approaches."
2) We have modified the text at (3): "errors in the background mole fractions, in particular for regional inversions;"

*l1170 - [Fossil CO 2 emissions are more straightforward to estimate than ecosystem fluxes due to combustion being easier to model and parameterize at large scales.] → generally this is true, but more specifically the issue is that fuel production, trade, including end-point sales (and implicitly consumption) are tracked very well by statistics. The use of 'model and parameterize' to implies the use of a process/mechanistic-based approach, which I don't think is the case.*

We have modified the text to read (L1088): "Fossil $CO_2$ emissions are more straightforward to estimate than ecosystem fluxes due to extensive data collection around fuel production and trade, assuming that fuel statistics and accurate emission factors are available." (combining with another reviewer's comment)

*l1176 - [However, this initial TD inversion is not yet capable of distinguishing the minor differences between the various BU estimates and does not yet quantify uncertainties.] → unlike, e.g., Basu et al, 2020.*

We have modified the text (L1094): "However, this initial TD inversion is not yet capable of distinguishing the minor differences between the various BU estimates and does not yet quantify uncertainties, unlike, for example, Basu et al. (2020), which presents fossil fuel combustion and cement production emission including uncertainty estimates for the United States."

The Basu reference has been added to the References: "Basu, S., Lehman, S. J., Miller, J. B., Andrews, A. E., Sweeney, C., Gurney, K. R., Xu, X., Southon, J., and Tans, P. P.: Estimating US fossil fuel $CO_2$ emissions from measurements of $^{14}C$ in atmospheric $CO_2$, Proc. Nat. Acad. Sci., 117, 13300-13307, https://doi.org/10.1073/pnas.1919032117, 2020."

*l1184 - [The IPCC guidelines prescribe methodologies that are used to estimate the CO 2 fluxes in the NGHGI, but grant countries significant freedom to adopt methods appropriate to their national circumstances.] → Globally this is certainly true. But how important is this in the EU27+UK?*

This is still true in the EU27+UK to a large extent. A more complete discussion of this is given in the EU National Inventory Report, which shows the countries using Tier 1, Tier 2, and Tier 3 approaches to various parts of LULUCF fluxes. Even among Tier 3 approaches, there are no standardized models used across the countries. We have added the text (L1105): "Even in the European Union, Member States use a wide variety of stock-change and gain-loss methods ranging from Tier 1 to Tier 3 depending on the specific LULUCF flux being estimated (EU NIR, 2021)." to clarify.

*l1205 - [For example, indirect fluxes on managed land are included in NGHGI and FAOSTAT,] → specify here 'indirect fluxes resulting from x,y,z…'*

L1127: We have added, "indirect fluxes resulting from long-term changes in growing conditions, such as $CO_2$, air temperature, and water availability".

*l1206 - [while additional sink capacity (e.g., Petrescu et al., 2021b) is included in estimates from process-based models (e.g., ORCHIDEE or TRENDY DGVMs).] → this is not clear. what do you mean by 'capacity'? The future or unrealized potential to be a sink or do you just mean to say additional sink?*

We have added "additional sink capacity" to the glossary, in addition to making the following modification to that sentence (L1129): "managed land are included in NGHGI and FAOSTAT. Additional sink capacity compared to pre-industrial conditions (also called the "amplification effect", e.g., Gasser and Ciais, 2013) occurs on Forest land in process-based models (e.g., ORCHIDEE or TRENDY DGVMs) due to improved growing conditions resulting from $CO_2$ fertilization, climate change, and anthropogenic nitrogen deposition, while this is not included in bookkeeping models which use the same regrowth curves for pre-industrial and modern times. The use of gross land…"

The following reference has been added to the References:

Gasser, T. and Ciais, P.: A theoretical framework for the net land-to-atmosphere $CO_2$ flux and its implications in the definition of "emissions from land-use change", Earth Syst. Dynam., 4, 171–186, https://doi.org/10.5194/esd-4-171-2013, 2013.

*l1219 - TD estimates also show very large inter-annual variability (Fig. 9). → And (I think) inter-model spread which as mentioned above is really the indicator of uncertainty, not IAV.]*

We have modified the text, "very large inter-annual variability and uncertainty (Fig. 5)".

*l1222 - [Currently, regional inversions (LUMIA, CSR and EUROCOM) are still under development and face different challenges from the coarser resolution global systems used here to represent regional results (GCP).] → this is all true, but at least according to Fig. 10, the regional inversions seem to do significantly better in mean and spread compared to the global inversions. I think it's probably worth mentioning this at this point.*

We have added the text (L1148): "As seen in Fig. 6, the mean of the regional inversions appears to agree better with the NGHGI than that of the global inversions, after the net carbon fluxes from lateral transfers are taken into account. In addition, the inter-model spread of the regional inversions is smaller."

*l1272 - [Similar to Petrescu et al. (2021b), this updated study concludes that a complete, ready-for-purpose monitoring system providing annual carbon fluxes across Europe is still under development, but data sources are beginning to show improved agreement compared to previous estimates. Therefore, significant effort must still be undertaken to reduce the uncertainty across all potential methods (i.e., structural uncertainty in the models as well as the input data supplied to the models or inventory approaches) used in such a system (e.g. Maenhout et al., 2020).] → Convergence of TD and BU is not necessarily a sign of progress (but it may be). The real goal should be, imo, to be able to robustly quantify and then reduce uncertainties in the various methods precisely so that differences in the central values can be identified and understood.*

We feel the reviewer agrees with us on this point. We have tried to clarify by modifying the following text (L1196): "Significant effort must still be undertaken to robustly quantify and then reduce uncertainties (both in the models themselves as well as in their input data) used in such a system so that differences in the central values can be identified and understood (e.g., Maenhout et al., 2020)."

*l1282 - [Forest management changes forest composition and structure, which affects the exchange of energy with the atmosphere (Naudts et al., 2016),] → see also https://www.science.org/doi/full/10.1126/science.abm9684. This may not be hugely relevant to Europe but vis a vis the statement above re forest-based CO2 mitigation approaches being 'required', it's important to recognize the limited capacity for trees, globally, to deal with the mess we've made and continue to make.*

As the reviewer points out, this reference is not hugely relevant to Europe, while all the other references are focused on the continent. However, we have taken his point, and changed the "requires" in L1201 to "requires consideration of" in order to avoid implying that any single solution is a magic bullet.

*l1297 - [VERIFY's primary aim is to develop scientifically robust methods to assess the accuracy and potential biases in national inventories reported by the parties through an independent pre-operational framework.] → unclear [pre-operational]*

We have added the following sentence (L1224): ' "Pre-operational" seeks to bridge the gap between pure research efforts and those aiming to provide regular (e.g., annual) updates of a product.'

*l1307 - [bottom-up activity data (e.g., fuel use and emission factors) and ecosystem measurements (bottom-up models).] → as mentioned above, bottom up models seem very different to me than ecosystem measurements.*

We have rephrased this as "bottom-up activity data (e.g., fuel use and emission factors, as represented in inventories) and ecosystem measurements (e.g., aboveground biomass and net ecosystem fluxes, as assimilated into bottom-up and top-down models)."

*l1451 - [EDGAR v6.0 Uncertainties:]→ it would be much more helpful in this section to get insight into the basis of AD and EF uncertainties more than the math involved in propagating those uncertainties.*
*l1453 - [the methodology devised by IPCC (2006, Chapter 3) is adopted, that is the overall uncertainty is the square root of the sum of squares of the uncertainty of the EF and AD (uncertainty of the product of two variables).] → this sounds like the error formula for the addition, not multiplication, of two variables. For products, the relative uncertainty is the square root of the sum of the squares of the relative uncertainties of each factor.*

According to Solazzo et al. (2021), the uncertainties are taken from the default uncertainties of the IPCC guidelines, with few exceptions. Thus, their work addresses mostly the uncertainty aggregation across sectors and countries.

The equation for uncertainty propagation comes directly from Solazzo et al. (2021), Eq. 1, and, indeed, Eq. 3.1 in the IPCC 2006 guidelines. As the reviewer points out, the relative uncertainty of a product is the square root of the sum of the squares of the relative uncertainties of each factor. We have clarified both of these comments by making the following modifications in the text (L1396): "(stemming from lack of knowledge of the true value of the EF and AD), the methodology devised by IPCC (2006, Chapter 3) is adopted (Solazzo et al., 2021), including the use of default uncertainties. The overall relative uncertainty in emissions is thus given by simple error propagation for the product of two variables, where the overall relative uncertainty is the square root of the sum of squares of the relative uncertainties of the EF and AD."

*l1467 - [[CDIAC] includes emissions from fossil fuels and cement production from 1751 to year n-3.] → and (maybe not important in Europe?) gas flaring.*

We have clarified that gas flaring is included under fossil fuels: "emissions from fossil fuels (including gas flaring) and cement production".

*l1470 - [EIA. The US Energy Information Administration publishes international energy statistics and from these derives estimates of energy combustion CO 2 emissions.] → based on production, consumption,…? some additional information would be useful here.*

EIA's method is not completely documented. We have modified the text at L1421 to be: "The US Energy Information Administration publishes international energy statistics and from these derives estimates of $CO_2$ emissions from energy combustion based on energy consumption."

*l1473 - [The International Energy Agency publishes international energy statistics and from these derives estimates of energy combustion CO 2 emissions including from the use of coal in the iron and steel industry.] → do the other BU products not include this? Why are you mentioning this specifically?*

IEA focuses on energy combustion emissions, but in addition adds iron and steel estimates for the IPPU sector. We have rephrased this to be more clear (L1427): "The International Energy Agency publishes international energy statistics and from these derives estimates of $CO_2$ emissions from energy combustion. In addition, IEA also estimates emissions from the use of coal in the iron and steel industry, while not providing any other IPPU estimates."

*l1476 - [IEA. Estimates are available by sector for a fee.] → Are annual national totals free? This wasn't clear from your data availability statement.*

We have modified the Data Availability statement by adding the text (L1066): "Most sector-level data from IEA are available for a fee, although some high-level emissions data can be accessed free-of-charge." We have modified the sentence highlighted by the reviewer in the same fashion (L1431).

*l1477 - [GCP] → To what extent are these values different than CDIAC. Maybe I'm out of date, but I had thought CDIAC annual national (and global) totals were the foundation of the GCP values.*

It is correct that the GCP's fossil $CO_2$ dataset was once entirely derived solely from CDIAC's dataset, with some extension using BP data, but this has since changed as described in Andrew and Peters (2022). We have added the sentence (L1437): "GCP's fossil $CO_2$ dataset was once entirely derived solely from CDIAC's dataset, with some extension using BP data, but this has since changed as described in Andrew and Peters (2022)."

We have added the following reference for the dataset:

Andrew, R. M., Peters, G.: The Global Carbon Project's fossil $CO_2$ emissions dataset. Global Carbon Project. Dataset. https://doi.org/10.5281/zenodo.5569234. Accessed 01 Oct 2021.

**Trivial – changing a couple words based on the reviewer's suggestion**

*l97 - [for a DGVM are mixed] → the DGVMs, assuming there is more than one*

Done.

*l112 - clarify that this is shorthand for mole fraction, which is better nomenclature and used in the previous line.*

We have added the text, "(referred to as "concentrations")".

*Footnote 4: Define SRCCL*

Done. "Special Report on Climate Change and Land".

*l471 - [The atmospheric residence time of NO2 is another major source of uncertainty.] ☝☝☺ I don't think 'residence time' accurately reflects the complexity here. This is not like Radon; instead there is a lot of complex chemistry involved in both the production and destruction of NO2 that is not appropriately described by an e-folding residence time.*

The text has been changed to "The atmospheric chemistry surrounding both production and destruction of $NO_2$".

*l264 - [The main reason is that the types of atmospheric monitoring instruments suitable for fossil CO2 atmospheric inversions have not yet been widely deployed (Ciais et al., 2015).] -> better to say 'atmospheric measurements' because the main issue is not deployment of instruments but rather the number of 14C measurements at centralized laboratory locations (mainly AMS)*

Done.

*l504 - [Carbon uptake as seen by the atmosphere may occur on either managed or unmanaged land, and results from processes such as photosynthesis, respiration, and disturbances (e.g., fire, pests, harvest).] → Consider rewording along lines of 'Net carbon uptake… results from the balance of …'. Strictly speaking, carbon uptake can't result from respiration, e.g.*

Done.

*l585 - remove "as".*

Done.

*l650 - [ORCHIDEE shows a high inter-annual variability in carbon fluxes for forests in Fig. 5 because it incorporates meteorological data at sub-monthly timescales, while methods based on forest inventories are generally updated only every few years (e.g, five years for FRA), which results in a more climatological perspective.] → maybe say 'higher than inventory' or similar. full range variability of 0.080 PgC/yr is not very large from large scale perspectives.*

Done.

*l658 - [while the NBP reported in figures in this paper include fluxes from litter and soil respiration.] → maybe rephrase as 'respiratory fluxes from litter and soil'.*

Done.

*l678 - [Cropland (CL, represented in the UNFCCC NGHGI 2021 as UNFCCC category 4B) includes net CO2 emissions and removals from soil organic carbon (SOC)] → I suspect you don't mean 'removals from' — e.g. leaching — but 'rather emissions from or removals by'. Please revise if so.*
*l756 - [Grassland (GL, UNFCCC category 4C) includes net CO 2 emissions and removals from soil organic carbon] → same comment as above…*

Done.

*l682 - [Note that the FAOSTAT value only includes the carbon flux from organic soils drained for agriculture, while ECOSSE, EPIC-IIASA, and ORCHIDEE include biomass volatilized immediately upon harvest; biomass left on site to decay as litter; and soil organic carbon.] → presumably you mean something like 'respiration from SOC'.*

We have replaced the text by "carbon fluxes from biomass volatilized immediately upon harvest; biomass left on site to decay as litter; and respiration from soil organic carbon".

*l685 - [The previous synthesis of Petrescu et al. (2021b) (Fig. 6, top) compared models against results for GL-GL from the NGHGI. For the current work, we compare against the total Grassland values (GL). The reason for this is that FAOSTAT, ECOSSE, EPIC-IIASA,] → Grasslands are in the next section. It's confusing that you are mentioning this here? Or did you mean to write CL-CL and Cropland?*

We have corrected the error, changing GL-GL to CL-CL and Grassland to Cropland.

*l695 - remove "data"*

Done.

*l800 - [A Tier 1 IPCC approach assumes no changes in either living or dead biomass pools on grasslands; only considers organic soils which have been drained for grazing; and only considers mineral soils which have undergone a change in management. ] → 'it only considers'. Eliminate next semi-colon and make a separate sentence. → it only*

We have broken this sentence up into two.

*l818 - [States reports] → make possessive*
*l835 - [model structural uncertainty (i.e., processes not included)] → change to 'e.g.' because there are more kinds of structural uncertainties besides whether processes are included or not. For example, the choice of parameterizations, among other things.*
*l841 - [In addition, DGVMs often do not distinguish between managed and unmanaged land, while NGHGIs report results only from managed land.] → do any? or are managed/non-managed land distinctions applied ex post facto.*
*l848 - [In addition, bookkeeping models do not account for fluxes arising from "indirect" anthropogenic influences such as $CO_2$ fertilization or climate change.] → I can see how bookkeeping models do not \*explicitly\* account for climate and CO2 effects, but in the example of forest regrowth following pasture conversion, surely CO2 and climate have an impact on this regrowth, so these effects would seem to be accounted for, just implicitly. I may be misunderstanding the bookkeeping process though.*

The text covering these comments was removed due to length considerations following a comment by another reviewer, and therefore these points are moot. However, the comment at l848 was found to be relevant elsewhere, and thus addressed there.

*l871 - [The gray bars represent the individual model data] → the spread of the*

Done, changing "represent" to "represents" afterwards for subject-verb agreement.

*l888 - [fall within the 95 % confidence interval of the UNFCCC NGHGI estimate of -86 ± 33 Tg C yr -1 . Only H&N rests apart with a stronger sink (-142 Tg C yr -1 ).] → although without uncertainty on this value, it's hard to really say how different it is than the national inventories.*

We have added ", although it's difficult to say how different it is from NGHGI without uncertainty estimates."

*l897 - [However, all of these estimates fall inside the NGHGI uncertainty range in Fig. 8. This suggests that indirect effects are small in the EU27+UK.] → I'm not sure I agree with this logic. The indirect effect here appears to be 55 TgC/yr which is substantially larger than the 33 TgC/yr uncertainty (one sigma?) of the inventories.*

We have removed the sentence "This suggests that indirect effects are small in the EU27+UK."

*l908 - [Gross fluxes are, by definition, larger than net fluxes, and consequently a method which incorporates gross fluxes (like CABLE-POP) can be expected to undergo larger changes than a method incorporating net fluxes (like ORCHIDEE).] → I don't follow the logic here re 'incorporation' of gross fluxes. For example if you had two TBMs predicting NEE, one which used a statistical model for NEE direcctly and the other which used process-based (or other) formulations for GPP and respiration, I don't think the 'direct' NEE model would be guaranteed to have less IAV in NEE than the sum of GPP and Resp from the other model. IAV would depend on the nature of the formulations.*

This text has been eliminated due to efforts to shorten, and therefore this question is moot.

*l916 - [the fact that DGVM models] → DGVMs -- M and model is redundant.*

Done.

*l928 - [In these inversions, all components of the carbon cycle that contribute to the observed atmospheric CO 2 gradients between stations are implicitly included as the inversions incorporate observed atmospheric concentrations of CO 2 .] → globally, at least, not many locations where CO2 mole fraction is measured can justifiably be referred to as 'stations'; it's more accurate to refer to all as 'observation sites'.*

Done.

*l934 - [Bottom-up methods (including all the NGHGIs for European countries) do not consider emissions and removal of atmospheric CO 2 due to lateral transport of carbon, while observations assimilated into top-down inversions record all CO 2 fluxes without separating their components.] → this is true, but 'separating their components' is a bit vague and maybe not quite the point you want to be making. I think the issue with lateral transport and inversions is something along the lines of 'inversions calculate geographically resolved net land-atmosphere CO2 fluxes without regard to the original location of photosynthetic assimilation.'*

We have changed the text according to the reviewer's suggestion.

*l955 - [Inversions, on the other hand, see the HWPs where they transform into CO 2 , either through decomposition or incineration.] → combustion is probably a better word choice here as it is describing the relevant chemical process; incineration is more particular.*

We changed "incineration" to "combustion".

*l959 – remove "F" in "NFGHGIs".*

Done.

*l959 - here and many other places be consistent about pluralization of NGHGI — in many cases it seems that the "s" is appropriate but missing, as at the end of the next line.*

We have gone through and made an effort to correct this. It's sometimes confusing, as the EU NGHGI is a collection of all the Member States' NGHGIs. In addition, sometimes the whole compilation is referred to as a single inventory, while other times a sector or subsector is referred to as an inventory. We have tried to use NGHGI when referring to a specific timeseries presented here for the EU, and NGHGIs when referring to general inventories.

*l1007 - [Out of the GCP2021 models, CAMS was the model responsible for the lower sinks (data not shown), which may be due partly to changes in the stations assimilated.] → it sounds from this construction that there were multiple variants of CAMS used (i.e., 'lower sink\*s\*')? If so, maybe rephrase to refer to CAMS variants and not just CAMS.*

"lower sinks" here refers to individual years. We have rephrased to be, "CAMS was the model responsible for the strongest sink in the ensemble during most years." to also avoid a potentially confusing "lower".

*l1008 - [Table B2 summarizes the processes included in the CO 2 land models presented in this work, as these processes are seen for the moment as the main cause of discrepancies between estimates shown in all the previous figures.] → this statement seems overly general. what about the differences between inverse flux spread and/or IAV and the inventories? This has nothing to do with the land models. Please refine this statement.*

We have changed the text to be, "Table B2 summarizes the processes included in the various approaches to estimating CO2 land fluxes presented in this work".

*l1109 - [GST] → defined yet?*

We have defined it here and removed the acronym, as it was not used anywhere else.

*l1193 - [should contribute to achieving Climate neutrality within the Union by 2050 (EU, 2021b).] → lower case. also do you mean 'carbon neutrality'?*

We have rewritten "climate" with a lowercase "c". "Climate neutrality" is the phrase used by the European Union in the cited document.

*l1241 - [This likely reflects relatively the use of data-driven models] → remove 'relatively'*

Done.

*l1362 - [On the other hand, TD estimates are constrained by independent atmospheric observations and can serve as an additional, nearly independent quality check for NGHGIs.]→ This really depends on the inversion, especially what it uses as a prior (hopefully not the same inventory it's attempting to evaluate) and the prior uncertainty, among other things. You could say 'potentially independent'.*

Done.

*l1372 - [Under the UNFCCC convention and its Kyoto Protocol, national greenhouse gas (GHG) inventories] → already defined I think…*

We have replaced "greenhouse gas (GHG)" by "GHG", adding it to the glossary. We have also changed one other occurrence, making sure GHG was defined at the first usage.

*l1595 [LUH2v2-ESACCI: "We desc] → remove quotation marks?*

This whole paragraph is a quotation. We have added the citation at the beginning instead of the end in order to make this clearer.